# Learning In-context $n$-grams with Transformers: Sub-$n$-grams Are Near-stationary Points

**Aditya Varre** [* 1]   **Gizem Yüce** [* 1]   **Nicolas Flammarion** [1]

## Abstract

Motivated by empirical observations of prolonged plateaus and stage-wise progression during training, we investigate the loss landscape of transformer models trained on in-context next-token prediction tasks. In particular, we focus on learning in-context $n$-gram language models under cross-entropy loss, and establish a sufficient condition for parameter configurations to be stationary points. We then construct a set of parameter configurations for a simplified transformer model that represent $k$-gram estimators (for $k \leqslant n$), and show that the gradient of the population loss at these solutions vanishes in the limit of infinite sequence length and parameter norm. This reveals a key property of the loss landscape: sub-$n$-grams are near-stationary points of the population cross-entropy loss, offering theoretical insight into widely observed phenomena such as stage-wise learning dynamics and emergent phase transitions. These insights are further supported by numerical experiments that illustrate the learning dynamics of $n$-grams, characterized by discrete transitions between near-stationary solutions.

## 1. Introduction

Transformers (Vaswani et al., 2017) have become central to modern machine learning, due to their capabilities such as in-context learning (ICL) (Brown, 2020)—the ability of models to perform new tasks by leveraging a few examples provided within the context, without the need for parameter updates or retraining.

Recent empirical studies have revealed that the dynamics that result in in-context learning abilities often deviate from a simple monotonic decrease in loss, exhibiting complex

behaviors with plateaus such as grokking (Power et al., 2022) and stage-wise transitions (Olsson et al., 2022) where the training process frequently lingers in regions of slow progress before going through a sudden phase transition after which ICL abilities are acquired. Mechanistic interpretability studies suggest that after these plateaus, specific circuits, such as induction heads (Olsson et al., 2022), or syntactic structures (Chen et al., 2024a), are gradually learned. This raises a natural question:

*Why does training linger at plateaus before developing such abilities?*

The setting of Edelman et al. (2024) provides an avenue to take a closer look at this question with the specialized task of learning in-context $n$-grams (Shannon, 1948; Chomsky, 1956; Brown et al., 1992). Here, the training unfolds in a hierarchical, phase-wise manner. It begins with the transformer making uniform predictions, progresses through unigram and bigram predictions, and potentially generalizes to higher-order $n$-grams. Figure 1 illustrates how these training phases overlap with the losses of the sub-$n$-gram estimators.

Building on this observation, we provide a theoretical foundation for why training lingers at long plateaus—the loss landscape of transformers trained on in-context sequential data exhibit stationary points aligned with sub-hierarchical or sub-syntactic solutions. These points act as intermediate solutions where gradients vanish, causing training to stagnate before transitioning to the next hierarchical level.

Concretely, this paper explores the loss landscape of transformer models trained on in-context $n$-gram language models to predict the next token. We show a sufficient stationarity condition for the cross-entropy loss that the sub-$n$-gram constructions satisfy, shedding light on the incremental and phase-wise learning phenomena observed during training for in-context $n$-grams. Our main contributions can be summarized as follows:

- In Section 3, we provide a sufficient condition for the solutions to be the stationary points of the cross-entropy loss in the next-token prediction task when model's score function depends solely on a sub-sequence of the input history. This characterization provides a powerful tool for

---
*Equal contribution   [1]Theory of Machine Learning Lab, EPFL, Switzerland.   Correspondence to:   Aditya Varre <aditya.varre@epfl.ch>, Gizem Yüce <gizem.yuce@epfl.ch>.

*Proceedings of the $42^{nd}$ International Conference on Machine Learning*, Vancouver, Canada. PMLR 267, 2025. Copyright 2025 by the author(s).

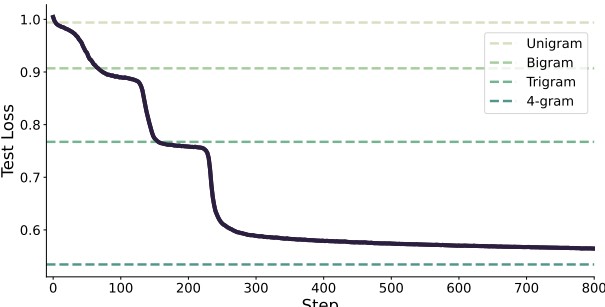

Figure 1: The stage-wise behavior of the test loss during training for the $4$-gram language model. The dashed lines represent the cross entropy loss of the $k$-gram estimators for $k = 1, 2, 3, 4$. The plateaus of the test loss overlap with the losses of $k$-gram estimators.

analyzing the structure of the loss landscape for various tasks, including but not limited to $n$-grams.

- In Section 4, we demonstrate that a set of solutions representing sub-$n$-gram estimators are near-stationary points, by verifying that they satisfy the conditions sufficient for the gradient of the population loss to converge to zero in the infinite weight and sequence length limit.

- In Section 5, we present empirical evidence that illustrates the structural evolution of these models during training and how transitions between training phases are in alignment with the predictions of our theory.

## 1.1. Related Work

**In-context Learning.** To understand in-context learning (ICL) (Brown, 2020), previous works have explored various approaches. One approach is mechanistic interpretability, which has revealed the emergence of circuits called induction heads during ICL (Olsson et al., 2022). Another approach involves studying ICL on specific hypothesis classes to understand how transformers solve these tasks in context. A common feature of the training dynamics in these studies is the presence of plateaus (Chen et al., 2024a; Kim et al., 2024), after which the models acquire certain capabilities. Examples include studying ICL on regression tasks (Garg et al., 2022; Von Oswald et al., 2023; Ahn et al., 2024), boolean functions (Bhattamishra et al., 2023), regular languages (Akyürek et al., 2024), and $n$-grams.

**(In-context) Learning $n$-gram language models.** $n$-gram language models, or higher-order Markov chains, are effective mathematical models for generating sequential data and capturing certain aspects of natural language (Shannon, 1948; Jelinek, 1998; Jurafsky & Martin, 2009). As a result, many studies have focused on analyzing transformers through the lens of Markov sequences. Svete & Cotterell (2024) examines the representational limits of transformers on $n$-grams, while Rajaraman et al. (2024) focuses on the in-context counterpart. Makkuva et al. (2024) explores

the landscape of transformers on data from binary first-order Markov chains (bigrams) but does not consider the in-context setting. Bietti et al. (2024) investigates the formation of induction heads needed to learn bigrams with specific trigger tokens. Additionally, Nichani et al. (2024) studies the formation of induction heads through gradient descent in learning causal structures. A closely related work to ours is Edelman et al. (2024). They report the stage-wise dynamics of transformers while learning in-context $n$-grams, and identify that these stages correspond to sub-$n$-grams. However, their theoretical analysis is limited to one step of gradient descent on binary bigrams, while we characterize the sub-$n$-grams as near-stationary points for unrestricted $n$ and vocabulary size, explaining the long plateaus once the training reaches a state expressing these solutions. Chen et al. (2024b) study the same in-context $n$-gram prediction task, but with an architecture that involves a feed-forward network layer that does the token selection. However, they have a threee-stage training procedure and an initialization scheme that ensures different heads attend to different tokens from the start, eliminating the stage-wise learning dynamics we study.

**Sequential learning.** Fukumizu & Amari (2000) analyzes the plateaus in the loss curve and their relationship to critical points for supervised learning with neural networks. The characterization of dynamics, including jumps between these stationary points, has also been studied for simpler models such as matrix and tensor factorization (Razin et al., 2021; Jiang et al., 2022), matrix sensing (Arora et al., 2019; Li et al., 2021; Jin et al., 2023), diagonal networks (Gissin et al., 2020; Berthier, 2022; Pesme & Flammarion, 2023), linear networks (Saxe et al., 2019; Gidel et al., 2019; Jacot et al., 2021; Varre et al., 2023), ReLU networks (Boursier et al., 2022; Abbe et al., 2023) and transformers with diagonal weight matrices (Boix-Adsera et al., 2023).

**Large Language Models(LLMs) as $n$-grams** Nguyen (2024) investigates whether LLM predictions can be approximated by simpler, interpretable statistical rules based on $n$-gram frequencies, and shows that LLMs exhibit curriculum learning during training—starting with simpler $n$-gram patterns and progressively capturing more complex ones. In a related vein, Zekri et al. (2025) leverages the equivalence between auto-regressive models and $n$-gram models with long contexts to derive generalization bounds.

## 2. Problem Setting

In this section, we define the in-context next token prediction loss, the landscape of which we aim to understand for transformers. Next, we introduce the $n$-gram language models and formally describe the disentangled attention-only transformer architecture.

**Notation.** Let $[\mathcal{S}] = \{1, 2, \ldots \mathcal{S}\}$ be a finite alphabet for $\mathcal{S} \in \mathbb{N}$. The Kleene star $[\mathcal{S}]^*$ denotes the set of all finite-length sequences whose elements are in $[\mathcal{S}]$. For $l \leqslant k$, $x_l^k = \{x_l, x_{l+1} \ldots x_{k-1}, x_k\} \in [\mathcal{S}]^{k-l+1}$ denotes a subsequence, and when not specified, $l = 1$. For an element $x_t$ in a sequence, its $l$-history refers to $x_{t-l}^{t-1}$ and its $(-l)$-token refers to $x_{t-l}$. $\Delta^{k-1}$ denotes the probability simplex over $\mathbb{R}^k$. A language model $p$ is a distribution over $[\mathcal{S}]^*$. We use $\mathrm{e}_i^d$ to denote the $i^{\text{th}}$ elementary basis vector of $\mathbb{R}^d$, simplified to $e_i$ when $d = \mathcal{S}$. For a matrix $r \in \mathbb{R}^{T \times d}$, $r[t] \in \mathbb{R}^d$ denotes the vector representation of it's $t^{\text{th}}$ row.

## 2.1. In-context Next Token Predictions

Consider a language model $p$ and a parametric model $p(\theta, .) : [\mathcal{S}]^* \to \Delta^{\mathcal{S}-1}$, which predicts the probability of the next token. Given a sequence $x^T \in [\mathcal{S}]^T$, the performance of the model in predicting the next token is evaluated using the cross-entropy loss (CE):

$$\ell\left(\theta, x^T\right) = \sum_{s \in \mathcal{S}} p(x_{T+1}{=}s | x^T) \log\left(p(\theta, x^T)[s]\right).$$

In the in-context setting, we assume that the ground truth language model is a mixture of multiple language models. Specifically, for every sequence, a language model $p_\tau$ is sampled from a prior distribution $\mathcal{P}$. Then, the sequence $x^T$ is sampled from $p_\tau$.

The in-context population loss is defined as

$$\mathcal{L}(\theta) \coloneqq \mathop{\mathbb{E}}_{p_\tau \sim \mathcal{P}} \mathop{\mathbb{E}}_{x^T \sim p_\tau} \ell\left(\theta, x^T\right). \tag{1}$$

We focus on the task of learning in-context $n$-grams where $p_\tau$'s are modeled as $n$-gram language models.

## 2.2. In-context $n$-grams task

We start with the definition of $n$-gram language model and discuss some estimators for this setting.

**The $n$-gram Language Model.** A language model $p_\tau$ is an $n$-gram language model if it satisfies the two key assumptions:

(a) Markov property: The conditional probability of a token depends only on the $(n-1)$-history rather than the entire history, i.e.,

$$p_\tau\left(x_l | x^{l-1}\right) = p_\tau\left(x_l | x_{l-n+1}^{l-1}\right), \text{ for } l \geqslant n.$$

(b) Time Homogeneity: The transition probabilities are independent of the position in the sequence. Formally, for all $t, l \in [T]$ and sequences $s^n \in [\mathcal{S}]^n$,

$$p_\tau\left(x_l{=}s_n | x_{l-n+1}^{l-1}{=}s^{n-1}\right) = p_\tau\left(x_t{=}s_n | x_{t-n+1}^{t-1}{=}s^{n-1}\right).$$

The assumptions above distinguish $n$-grams from general language models by limiting the dependency range and assuming uniform transition dynamics over time. Although

these assumptions restrict their expressive power, they retain certain key characteristics of natural language, such as causal dependence on the prior context. Under these assumptions, the probability of the sequence $x^T$ can be expressed using the chain rule as

$$p_\tau(x^T) = \prod_{l \in [T]} p_\tau(x_l | x^{l-1}) = \prod_{l \in [T]} p_\tau\left(x_l | x_{l-n+1}^{l-1}\right).$$

**Estimators.** For the next token prediction task with the in-context $n$-gram language model, the $k$-gram estimator is of particular interest.

**Definition 2.1** ($k$-gram estimator). Given a sequence $x^T$ and $i \in [\mathcal{S}]$, the $k$-gram estimator is defined as

$$\widehat{p}_k\left(x_{T+1}{=}i \big| x^T\right) = \frac{\sum\limits_{l=k}^{T} \mathbb{1}\{x_{l-k+1}^{l-1}{=}x_{T-k+2}^T\} \mathbb{1}\{x_l{=}i\}}{\sum\limits_{l=k}^{T} \mathbb{1}\{x_{l-k+1}^{l-1}{=}x_{T-k+2}^T\}}.$$

Intuitively, this estimator checks whether the $(k{-}1)$-histories match and counts the tokens that follow. For $k = 1$ (unigram), it computes the empirical frequency of each token in the sequence. For $k = 2$ (bigram), the estimator computes the empirical frequency of tokens that follow those matching the $T^{\text{th}}$ token in the sequence. The $n$-gram estimator is the "in-context" maximum likelihood estimator (MLE) for the $n$-gram language model. Moreover, Han et al. (2021) shows that the smoothed version of this estimator achieves the minimax optimal rate for the next-token probability estimation. We include all the $k$-gram estimators where $k < n$, in the definition of sub-$n$-grams.

We note that our choice of the $n$-gram language model and $k$-gram estimator is primarily for ease of presentation. The results in this paper naturally extend to more general time-homogeneous causal dependencies in the sequence beyond $n$-grams as in Nichani et al. (2024). For example, our results also apply to causal graphs where a token $x_t$ depends only on specific parent tokens, such as $x_{t-2}$ and $x_{t-4}$. Similarly, while the $k$-gram estimator matches a contiguous $(k{-}1)$-history by definition, a similar estimator can match non-contiguous histories and count what follows. For instance, let the $(\{1, 3\})$-history of a token $x_t$ refer to $(x_{t-1}, x_{t-3})$; an estimator could leverage this pattern to predict subsequent tokens in the same way the 3-gram estimator does for the contiguous 2-history. We use the term sub-$n$-grams also to include such cases, which we discuss further in Appendix F.2. Next, we introduce the attention-only transformer architecture used for this in-context learning task.

## 2.3. Disentangled Transformer Architecture

Given an input sequence $x^T \in [\mathcal{S}]^*$, the transformer first maps each token to a $d$-dimensional embedding using token-wise semantic and positional embeddings, defined as $E : [\mathcal{S}] \to \mathbb{R}^d$ and $P : [T] \to \mathbb{R}^d$. The core of the transformer is the self-attention mechanism, which allows the model to weigh different parts of the input sequence based on learned similarity scores. Given a sequence embedding $r \in \mathbb{R}^{T \times d}$, self-attention computes a weighted sum of token embeddings as

$$\text{SA}_{(Q,K,V)}(r) = \boldsymbol{\sigma}\left(\left(r\mathbf{Q}^\top \mathbf{K} r^\top\right)\right) r\mathbf{V},$$

where $(\mathbf{Q}, \mathbf{K}, \mathbf{V})$, are the query, key and value matrices of the attention head, and the masked soft-max $\boldsymbol{\sigma}(\cdot)$ is defined as

$$\boldsymbol{\sigma}(x)[i][j] = \begin{cases} \frac{exp(x_{ij})}{\sum_{k \leqslant i} exp(x_{ik})}, & j \leqslant i \\ 0, & otherwise \end{cases}.$$

Traditionally, the outputs of self-attention layers are added to the residual stream. For ease of interpretation and analysis, we instead consider a disentangled architecture in which the outputs of individual heads and layers are concatenated rather than summed. This framework was proposed by Friedman et al. (2023) and formalized by Nichani et al. (2024). The disentangled architecture retains the same representational power as the standard formulation (see Theorem 3 of Nichani et al. (2024)) and is formally defined below.

**Definition 2.2 (Disentangled Attention-only Transformer).** Let $L$ be the depth, $\{h_\ell\}_{\ell \in [L]}$ be the number of heads per layer, $d$ be the embedding dimension, $d_\ell$ be the dimension of layer $\ell$, $d_h$ be the hidden dimension of the model parameters, and $d_{out}$ be the output dimension. For the $h^{\text{th}}$ head in the $\ell^{\text{th}}$ layer, let $\mathbf{Q}_\ell^{(h)}, \mathbf{K}_\ell^{(h)} \in \mathbb{R}^{d_h \times d_\ell}, \mathbf{V}_\ell^{(h)} \in \mathbb{R}^{d_\ell \times d_\ell}$, be the query, key, and value matrices of the respective head and layer, let $\text{SA}_\ell^{(h)}(\cdot) = \text{SA}_{\{\mathbf{Q}_\ell^{(h)}, \mathbf{K}_\ell^{(h)}, \mathbf{V}_\ell^{(h)}\}}(\cdot)$ and let $\mathbf{U} \in \mathbb{R}^{d_{out} \times d_L}$ be the unembedding matrix. Given an input sequence $x^T$, the disentangled transformer outputs the logits $\text{TF}(\theta)$ for $\theta = \left\{ \{\mathbf{Q}_\ell^{(h)}, \mathbf{K}_\ell^{(h)}, \mathbf{V}_\ell^{(h)}\}_{\ell \in [L], h \in [h_\ell]} \cup \mathbf{U} \right\}$, given by,

$$r_0 = [E(x^T), P(x^T)] \in \mathbb{R}^{T \times d},$$
$$r_\ell = [r_{\ell-1}, \text{SA}_\ell^{(1)}(r_{\ell-1}), \dots, \text{SA}_\ell^{(h)}(r_{\ell-1})]$$
$$\text{TF}(\theta) = r_L \mathbf{U}^\top.$$

To simplify the presentation of our theoretical results, we consider a two-layer simplified disentangled transformer with specific design choices. These modifications, detailed below, include orthogonal token embeddings and a fixed value matrix in the second layer.

**Embeddings.** The token embedding $S : [\mathcal{S}] \to \mathbb{R}^d$ is orthogonal, which means that the set $(s_i)_{i \in [\mathcal{S}]}$ forms an orthogonal family in $\mathbb{R}^d$. Here, $s_i$ denotes the embedding of the token $i$. For such an embedding to exist, $d \geqslant \mathcal{S}$. For any sequence $x^T$, the input is encoded as,

$$r_0 = \begin{bmatrix} s_{x_1} & s_{x_2} & \dots & s_{x_T} \end{bmatrix}^\top \in \mathbb{R}^{T \times d}.$$

No explicit positional embeddings are used[1].

**First Attention Layer.** The first attention layer contains $m$ attention heads, and each attention matrix is parameterized by a single learned matrix $\mathbf{A}_1^{(h)} \in \mathbb{R}^{T \times T}$ and is independent of the input embeddings. The output of each head is therefore given by

$$r_1^{(h)} = \boldsymbol{\sigma}\left(\mathbf{A}_1^{(h)}\right) r_0 \left(\mathbf{V}_1^{(h)}\right)^\top \in \mathbb{R}^{T \times d}. \quad (2)$$

This construction is equivalent to a standard attention head where token embeddings are concatenated with one-hot positional embeddings and the attention only relies on the latter. Thus, the output of the first layer is given by

$$r_1 = \begin{bmatrix} r_1^{(0)} & r_1^{(1)} & \dots & r_1^{(m)} \end{bmatrix} \in \mathbb{R}^{T \times (m+1)d}, \quad (3)$$

where $r_1^{(0)} = r_0$ is the skip connection.

**Second Attention Layer.** The second attention layer contains a single attention head, where the value matrix is fixed to $\mathbf{V}_2^{(1)} = [I_d; 0_d; \dots; 0_d]_{(m+1)d \times d}$ that reads the first block. Consequently , $r_1 \mathbf{V}_2^{(1)} = r_1^{(1)} = r_0$ and the output of the second layer is given by

$$r_2 = \boldsymbol{\sigma}\left(r_1 \mathbf{Q}_2^\top \mathbf{K}_2 r_1^\top\right) r_0 \in \mathbb{R}^{T \times d}. \quad (4)$$

We note that there is no concatenation to the residual stream in the second layer, which corresponds to not using a residual connection in the standard transformer.

Finally, the unembedding matrix is given by

$$U = \sum_{j=1}^{\mathcal{S}} e_j s_j^\top, \quad (5)$$

where $(e_j)$'s are canonical basis of $\mathbb{R}^{\mathcal{S}}$. Note that if the token embedding $S$ is the one-hot encoding, then $U$ is the identity matrix. Finally, given a sequence $x^T$, the probability of the next token estimated by the model, denoted by $p_\theta(x^T)$ is

$$p_\theta(x^T) = U r_2[T].$$

$\theta = \left\{\mathbf{A}_1^{(h)}, \mathbf{V}_1^{(h)}\right\}_{h=1}^{n-1} \cup \left\{\mathbf{K}_2, \mathbf{Q}_2\right\}$ denotes the set of parameters of our simplified disentangled model.

---

[1]Positional information is implicitly used through the attention matrix in the first layer.

# 3. A Sufficient Stationary Condition for Population CE on Sequences

In this section, we analyze the gradient of the population next-token cross-entropy loss for sequential data. This analysis provides fundamental insights into the training dynamics. We begin by presenting a lemma that provides a closed-form expression for the gradient.

**Lemma 3.1.** *Consider any parametric model $p_\theta(.)$ : $[\mathcal{S}]^T \to \Delta^{\mathcal{S}-1}$ that maps a sequence of states to a probability vector on the states. The derivative of the population cross-entropy loss $\mathcal{L}(\theta)$ with respect to a parameter $\theta_i \in \mathbb{R}$ is*

$$\partial_{\theta_i} \mathcal{L}(\theta) = \mathop{\mathbb{E}}_{p_\tau \sim \mathcal{P}} \mathop{\mathbb{E}}_{x^T \sim p_\tau} \left\langle p_\theta(x^T) - p_\tau(.\,|\,x^T), \partial_{\theta_i} \log p_\theta(x^T) \right\rangle,$$

*where $\partial_{\theta_i}$ is the partial derivative with respect to $\theta_i$ and $\log(\cdot)$ denotes component-wise logarithm.*

The lemma 3.1, presented for a fixed length $T$ for notational simplicity, generalizes to sequences of arbitrary length. This result extends prior analyses such as Bietti et al. (2024, Lemma 1) and Makkuva et al. (2024).

The gradient expression in Lemma 3.1 is the expectation of an inner product between two key terms: (a) the *prediction residual*, $p_\theta(x^T) - p_\tau(.\,|\,x^T)$, representing the error between the model's estimate and the true next-token distribution; and (b) the *score function*, which is the gradient of the model's logits. In the following proposition, we demonstrate that when the score function's structure depends only on a specific sub-sequence of the input, the population loss gradient simplifies significantly.

**Proposition 3.2.** *For any $\theta_* \in \mathbb{R}^p$ such that $\partial_{\theta=\theta_*} \log p_\theta(x^T) = g(p_\tau, x_t^T)$, i.e., the score is solely a function of the context $p_\tau$ and the last $T-t+1$ elements of the sequence $x^T$, the gradient of the population loss $\mathcal{L}$ can be written as*

$$\nabla \mathcal{L}(\theta_*) = \mathop{\mathbb{E}}_{p_\tau \sim \mathcal{P}} \mathop{\mathbb{E}}_{x^T \sim p_\tau} \left\langle p_{\theta_*}(x^T) - p_\tau(.\,|\,x_t^T), g(p_\tau, x_t^T) \right\rangle.$$

*Furthermore, if for such $\theta_* \in \mathbb{R}^p$, the model estimates the conditional probability of the next token $p_\tau(.\,|\,x_t^T)$, i.e., $p_{\theta_*}(x^T) = p_\tau(.\,|\,x_t^T)$ almost surely for $p_\tau \sim \mathcal{P}$, then $\theta_*$ is a stationary point.*

Proposition 3.2 provides a sufficient condition for stationarity. It reveals that if the score function $g$ depends only on the context and a suffix $x_t^T$, then the gradient is no longer driven by the error against the true probability conditioned on the full sequence, $p_\tau(.|x^T)$, but rather by the error against the probability conditioned on the shorter suffix, $p_\tau(.|x_t^T)$. Therefore, the population loss gradient vanishes when the model correctly learns to estimate the latter.

We leverage this proposition to formally show that the population loss gradient vanishes when the model correctly learns the true $k$-gram conditional probabilities. However, the proposition is not specific to the setting of $k$-grams and holds generally for cross-entropy loss in next-token prediction tasks. A generalization of this result, which applies to models that depend on any arbitrary subset of tokens (not just a contiguous suffix), is discussed in Appendix C.2.

# 4. Theoretical Insights into Stage-wise Dynamics Through the Loss Landscape

In this section, we first provide the representations of the sub-$n$-grams with the simplified disentangled transformer architecture discussed in Subsection 2.3. We then prove that these sub-$n$-gram constructions are near-stationary points of the loss.

## 4.1. Representing Sub-n-grams with Simplified Transformer

We now construct a disentangled transformer to represent the $n$-gram estimator in Definition 2.1. We then extend this construction to represent any $k$-gram estimator for $k \in [n]$ by deactivating specific heads. Parameters are often expressed as sums of outer products of orthogonal vectors, emphasizing their role as associative memories in the spirit of Bietti et al. (2024).

**Transformer representing $n$-gram.** Consider the simplified transformer model presented in Section 2.3 with $(n-1)$ attention heads.[2] For the $h^{\text{th}}$ head in the first layer, we set the parameters as follows:

$$\mathbf{A}_1^{(h)} = c \sum_{l=h}^{T-1} \mathbf{e}_l^T \left(\mathbf{e}_{l-h}^T\right)^\top + c \sum_{l=0}^{h-1} \mathbf{e}_l^T \left(\mathbf{e}_0^T\right)^\top,$$

$$\mathbf{V}_1^{(h)} = \sum_{j=1}^{\mathcal{S}} s_j s_j^\top.$$

The second layer query and key matrices are assigned such that

$$(\mathbf{Q}_2)^\top \mathbf{K}_2 = c \sum_{j=1}^{\mathcal{S}} \sum_{h=1}^{n-1} s_j^{h-1} (s_j^h)^\top, \qquad (6)$$

where $c > 0$ is a constant scaling factor and $s_n^h \in \mathbb{R}^{nd}$ is a block vector defined as $(s_n^h)^\top = \big[\underbrace{0_d^\top \ 0_d^\top \ \cdots}_{h \text{ times}} \ s_n^\top \ \underbrace{0_d^\top \ \cdots}_{n-h-1 \text{ times}}\big]$.

Intuitively, the $h^{\text{th}}$ head in the first layer is constructed to attend to the token at relative position $-h$. The value matrix

---

[2]The heads can be more than $n-1$.

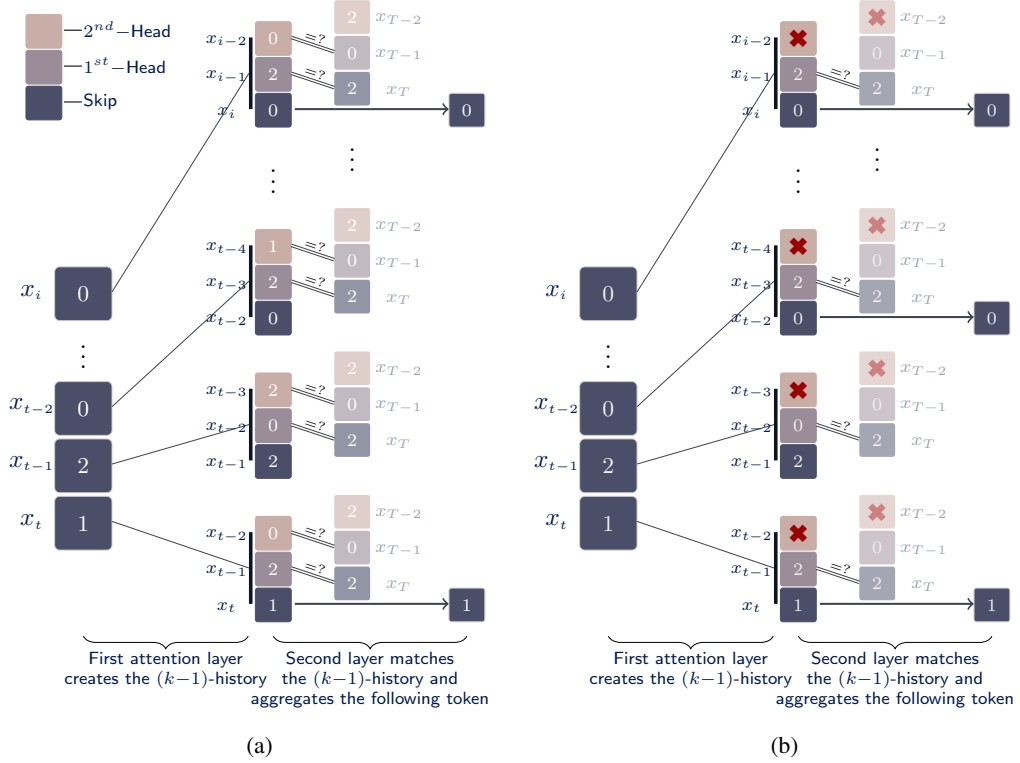

(a)             (b)

Figure 2: **Transformer representing a $n$-gram and $k$-gram estimator.** (a) The task we consider is learning in-context tri-grams ($n{=}3$). Here, we illustrate how the transformer given by $\boldsymbol{\theta}_*^n$ constructed in section 4.1 operates on the sequence $(\ldots, 2, x_i{=}0, \ldots, 2, 0, 2, x_t{=}1, \ldots 0, x_T{=}2)$ and computes a tri-gram($k{=}3$) estimator. Skip, $1^{st}$-Head, $2^{nd}$-Head denotes the outputs of the skip connection, $1^{st}$ and the $2^{nd}$ head. The first layer creates the $(2)-$history and the second layer compares it with the $(2)-$history of token $T{+}1$, i.e, $x_T = 2$. The second layer compares these histories and attends to tokens at $t, i$ and averages the tokens at that position. (b) ✖ represents that $2^{nd}$ head is deactivated. Hence, the first layer creates the $(1)-$history and the second layer matches with the $(1)-$history of token $T{+}1$, i.e, $x_T{=}2$ and attends and averages the tokens at $t, t{-}2, i$.

acts as an identity map, copying the embedding of this $(-h)$-token into the $h^{\text{th}}$ block of the first layer's output, $r_1^{(h)}$. As illustrated in Fig. 4, for any position $t$, the first layer's output aggregates the embeddings of the previous $n{-}1$ tokens (in the limit $c \to \infty$):

$$r_1[t] = \begin{bmatrix} s_{x_t}^\top & s_{x_{t-1}}^\top & \cdots & s_{x_{t-n+1}}^\top \end{bmatrix}^\top \in \mathbb{R}^{nd}. \quad (7)$$

The second attention layer then compares these histories using the specific structure of the query and key matrices defined in Equation (6). The pre-softmax attention score between the first-layer embeddings of the $i^{\text{th}}$ and $j^{\text{th}}$ tokens $r_1[i]$ and $r_1[j]$ is computed as:

$$\langle \mathbf{K}_2 r_1[j], (\mathbf{Q}_2) r_1[i] \rangle = c \sum_{l=1}^{n-1} \mathbb{1}\{x_{j-l}{=}x_{i+1-l}\}.$$

In the limit as $c \to \infty$, where the softmax converges to a hardmax, this construction ensures that the attention for predicting the token at $T + 1$ (using query from position

$T$) exclusively selects past positions whose $(n - 1)$-history matches the history at position $T$. The model's output is then the average of the tokens following these matches, thereby computing the $n$-gram MLE estimator (see Appendix B.1 for a formal proof).

**Transformer representing sub-$n$-gram.** The $n$-gram construction can be directly adapted to represent a $k$-gram for any $k < n$. The key insight is that the $h^{\text{th}}$ head is responsible for retrieving the $(-h)$-token. Hence, to compute a $k$-gram counting estimator, it suffices to deactivate the heads responsible for history elements beyond $k - 1$, as shown in Figure 4(b). The heads in the first layer are thus divided into two groups:

a) **Activated Heads.** These heads compute the $(-h)$-token for $h \in [k - 1]$.

b) **Deactivated Heads.** These heads output a zero vector, effectively ignoring history beyond $k - 1$ tokens.

The second layer structure is similar to $n$-grams case but with components related to the deactivated heads set to zero.

This is implemented by modifying the parameters as follows:

$$\mathbf{A}_1^{(h)} = \begin{cases} c \sum_{l=h}^{T-1} \mathbf{e}_l^T \left(\mathbf{e}_{l-h}^T\right)^\top + c \sum_{l=0}^{h-1} \mathbf{e}_l^T \left(\mathbf{e}_0^T\right)^\top, & \text{for } h \in [k-1], \\ \text{arbitrary} & \text{otherwise.} \end{cases}$$
(8a)

$$\mathbf{V}_1^{(h)} = \begin{cases} \sum_{j=1}^{\mathcal{S}} s_j s_j^\top \text{ for } h \in [k-1], \\ 0 \quad \text{otherwise,} \end{cases}$$
(8b)

$$(\mathbf{Q}_2)^\top \mathbf{K}_2 = c \sum_{j=1}^{\mathcal{S}} \sum_{h=1}^{k-1} s_j^{h-1} (s_j^h)^\top.$$
(8c)

This construction ensures that the second layer only compares the $(k-1)$-histories. The model attends uniformly to all past positions that share the relevant $k$-history, thereby computing the $k$-gram MLE estimator.

We denote the point given by Equations (8a), (8b), (8c) by $\boldsymbol{\theta}_*^k = \left(\{\mathbf{A}_1^{(h)}, \mathbf{V}_1^{(h)}\}_{h=1}^{n-1} \cup \mathbf{K}_2 \cup \mathbf{Q}_2\right)$ defined precisely in App. (10). In Lemma B.1, we provide formal results showing that $\boldsymbol{\theta}_*^k$ implements the $k$-gram estimator in Definition 2.1 in the limit $c \to \infty$.

**Other possible constructions.** The specific construction described above is not unique. For instance, multiple heads could be assigned to compute the same $(-l)$-token for $l < k$, and the result would hold as long as no head outputs the history for $l \geqslant k$. See Appendix F for details.

## 4.2. Sub-$n$-grams Are Near-Stationary Points

Having constructed the points $\boldsymbol{\theta}_*^k$ that implement $k$-gram estimators, we now show that they are first-order stationary points of the population cross-entropy loss in the large context and large norm asymptotic. This result follows from the general characterization of stationary points for the cross-entropy loss, provided in Section 3.

**Theorem 4.1.** For the simplified disentangled transformer $p_\theta$, the following on holds on the norm of the gradient at $\boldsymbol{\theta}_*^k$

$$\|\partial_{\theta=\boldsymbol{\theta}_*^k} L(\theta)\|$$
$$= \sqrt{c} \mathop{\mathbb{E}}_{p_\tau \sim \mathcal{P}} \mathop{\mathbb{E}}_{x^T} \mathcal{O}\left(\left\|p_\tau\left(.\big| x_{T-k+2}^T\right) - \widehat{p}_k\right\|^2\right) + \mathcal{O}(T\sqrt{c}e^{-c}).$$

This theorem quantifies the gradient norm at the $\boldsymbol{\theta}_*^k$ construction. The first term is driven by the statistical error of the $k$-gram MLE estimator and decays as $e^{-\Theta(T)}$ (Penev, 1991). Hence, in the limit of $c = \Theta(T) \to \infty$, the gradient vanishes. As the stationarity holds asymptotically as $T, c \to \infty$, we term these points *near*-stationary.

This result reveals a remarkable feature of the landscape of learning in-context $n$-gram language models with trans-

formers: points in parameter space that implement simpler solutions, $k$-gram estimators for $k < n$, are near-stationary points. This provides a formal basis for understanding stage-wise learning as a process of moving between these near-stationary points.

Note that our result holds for any sequence length $T$. It therefore also applies to the commonly used loss function, which averages the loss over sequences of varying lengths, i.e., $\mathcal{L}_*(\theta) = \sum_{t=t'}^T \mathcal{L}_t(\theta)$ where, with a slight abuse of notation, $\mathcal{L}_t$ refers to $\mathcal{L}$ in Equation (1) for sequences of length $t$. In the averaged loss, the gradients from the longer sequences suffer from the vanishing gradient problem as a result of the above theorem.

## 4.3. Proof Sketch

Our proof hinges on applying the stationarity result from Proposition 3.2. This requires establishing two key conditions for the parameter configuration: (a) demonstrating that model's prediction $p_{\theta_*^k}(x^T)$ converges to the true conditional probability $p_\tau(\cdot \mid x_{T-k+2}^T)$ and (b) proving that the score function depends solely on the $(k-1)$-history.

It is easy to see that the first condition holds asymptotically. As $c \to \infty$, the model $p_{\theta_*^k}(x^T)$ implements the $k$-gram estimator (see Lemma B.1), and as $T \to \infty$, the $k$-gram estimator converges to the conditional probability (see Lemma H.6 in the Appendix for the formal argument). Finally, to obtain non-asymptotic bounds on the gradient, we use these asymptotic computations while carefully controlling for the perturbation caused by non-asymptotic conditions.

The main technical challenge is proving the second condition: that the score function has a restricted dependency structure. Our argument proceeds by analyzing the structure of the model's output derivatives. The model's prediction is a weighted sum of one-hot vectors:

$$p_{\boldsymbol{\theta}}(x^T) = \sum_{i=1}^T a_{(i,T)}^{(2)} e_{x_i},$$
(9)

where $a_{(i,T)}^{(2)}$ are the attention scores in the second layer for key $i$ and query $T$ (see the Appendix B for details). The derivative with respect to any parameter $\theta_{(1)}, \theta_{(2)}$ in the first or second layer involves the derivatives of these attention scores:

$$\frac{\partial p_{\boldsymbol{\theta}}(x^T)}{\partial \theta_{(1)}}, \frac{\partial p_{\boldsymbol{\theta}}(x^T)}{\partial \theta_{(2)}} = \sum_{i=1}^T \frac{\partial a_{(i,T)}^{(2)}}{\partial r_1[i]} \frac{\partial r_1[i]}{\partial \theta_{(1)}} e_{x_i}, \sum_{i=1}^T \frac{\partial a_{(i,T)}^{(2)}}{\partial \theta_{(2)}} e_{x_i}.$$

A crucial observation is that the softmax function has a self-bounding property: the magnitude of the derivative of a softmax output is bounded by (and scales with) the output value itself. Consequently, when an attention score is close to zero, its gradient will also be close to zero. This property will play a key role in our subsequent analysis.

At the $k$-gram estimator parameterized by $\boldsymbol{\theta}_*^k$, the attention scores are highly sparse. Let $\mathsf{M}_T^k \subseteq [T]$, be the set of past positions whose $k$-history matches that of the $(T+1)^{\text{th}}$ token. Formally defined as $\mathsf{M}_T^k = \left\{ i \in [T] : \mathbb{1}\left(x_{i-k+1}^{i-1} = x_{T-k+2}^T\right) \right\}$. By construction, the attention scores for $\boldsymbol{\theta}_*^k$ are given by:

$$
a_{(i,T)}^{(2)} \approx \begin{cases} \frac{1}{|\mathsf{M}_T^k|}, & \text{for } i \in \mathsf{M}_T^k, \\ 0 & \text{o.w.} \end{cases}.
$$

Due to the self-bounding property, the summation in the gradient expression collapses, receiving significant contributions only from tokens within the matching set $\mathsf{M}_T^k$:

$$
\frac{\partial p_{\boldsymbol{\theta}}(x^T)}{\partial \theta_{(2)}}, \frac{\partial p_{\boldsymbol{\theta}}(x^T)}{\partial \theta_{(1)}}\bigg|_{\boldsymbol{\theta}=\boldsymbol{\theta}_*^k} \approx
$$
$$
\sum_{i \in \mathsf{M}_T^k} \frac{\partial a_{(i,T)}^{(2)}}{\partial \theta_{(2)}} e_{x_i}, \sum_{i \in \mathsf{M}_T^k} \frac{\partial a_{(i,T)}^{(2)}}{\partial r_1[i]} \cdots
$$

A key result we establish next is that the derivatives of $a_{(i,T)}^{(2)}, r_1[i]$ depend exclusively on the $(k-1)$-history of token $i$. Several reinforcing factors contribute to this result. Due to the specific structure of the *key* and *value* matrices in the second layer at $\boldsymbol{\theta}_*^k$, the gradients with respect to the parameters of the deactivated heads vanish. The deactivation, in turn, ensures that the embeddings after the first layer only contain the embeddings of the $(k-1)$-history. Finally, for $i \in \mathsf{M}_T^k$, its $(k-1)$-history is identical to that of token $T+1$. Together, these factors imply that the derivatives of the second layer are solely a function of $(k-1)$-history of token $T+1$. The complete proof is in Appendix B.2.

### 4.4. Extensions and Perspectives

**Beyond contiguous history.** Our main analysis focuses on contiguous $k$-gram histories (suffixes). However, as discussed previously in section 2.2, the dependency on the last $k$ tokens are not more important than the other $n-k$ for the $n$-gram data generating process under a uniform prior on the transition matrices. Our framework can be extended to show that estimators matching *arbitrary* subsets of the $(n-1)$-history (e.g., matching only the $(-i)$-token for some $i > 1$ as a counterpart of the bigram estimator) also correspond to stationary points under certain assumptions. See Appendix F.2 for details.

**Towards general transformer architecture.** Our parameter construction, techniques and methodology extend naturally to a general transformer architecture. First, positional encoding, which was not explicitly used in the simplified model, can be incorporated using one-hot positional encoding. The concatenation of the attention head output and residual connections can be replaced with the addition. Additionally, the value matrix in the second layer can be

incorporated in the architecture. However, in this case, the transformer's output would no longer be in $\Delta^{\mathcal{S}-1}$, requiring normalization via softmax at the end. This restriction on the value matrix can also be alleviated by using an MLP layer to approximate the logarithm (see Appendix F.1).

**On the Emergence of Syntactic Structure.** Consider the behavior of a gradient-based method at a sub-n-gram, say $\boldsymbol{\theta}_*^k$. As it is a stationary point, training remains at this point for a prolonged period, leading to a plateau in the training curve. However, as training progresses, the model eventually escapes due to landscape curvature or stochastic noise, allowing it to learn a new syntactic structure— attending to $(-k)$-token—before reaching the next stationary point. This phenomenon is general and has been empirically reported— emergence of a syntactic structure (Chen et al., 2024a; Wei et al., 2022), phase transitions (Olsson et al., 2022; Edelman et al., 2024). By carefully analyzing the loss landscape of a relevant yet simple in-context task, we demonstrate that the stationary points correspond to underlying syntactic structures. Consequently, our work provides insights into why these syntactic structures emerge following extended plateaus.

**Limitations.** Our theoretical results hold in the asymptotic limit of infinite sequence length ($T \to \infty$) and parameter scale ($c \to \infty$). The dependence on $c$ is mild, as the gradient decays exponentially for finite $c$. However, there is an inherent difficulty in moving beyond the assumption of infinite sequence length. Existing works on theoretical analysis of learning in-context $n$-gram like sequences with transformers, such as Rajaraman et al. (2024) and Nichani et al. (2024), also rely on this assumption. It is even unclear which estimator transformers learn at finite sequence lengths, even at the end of training. This makes it particularly challenging to determine what transformers exactly compute during intermediate stages of training.

## 5. Experimental Evaluation

In this section, we perform experiments on the disentangled transformer introduced in the previous section to examine the stage-wise learning behavior and analyze the different solutions the transformer learns during different stages of training. The code is available at `https://github.com/tml-epfl/sub-n-grams-are-stationary`.

**Experimental Setup.** We train our model on data generated from a trigram ($n = 3$) language model over a vocabulary of size $\mathcal{S} = 5$. Each in-context sequence has a length of $T = 32$, and the transition probabilities are drawn from a uniform Dirichlet prior, $\text{Dir}(\alpha\mathbf{1})$, with $\alpha = 0.5$. The model is the two-layer simplified Transformer analyzed in our theory, with two heads in the first layer and an embedding dimension of $d = 5$. We use one-hot token embeddings

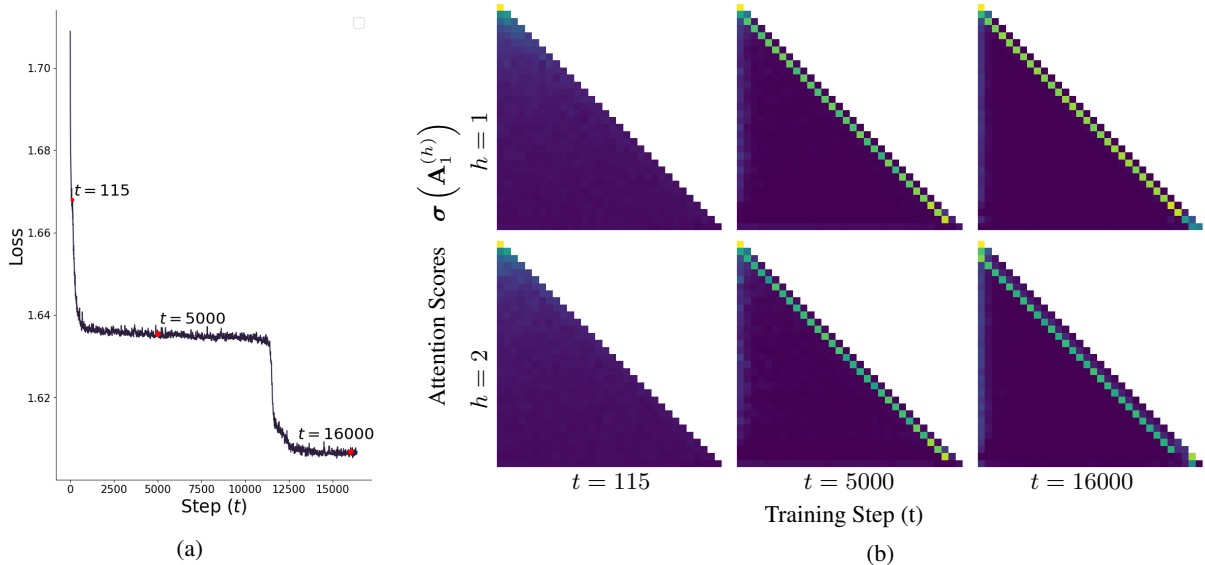

(a)                                                                 (b)

Figure 3: The evolution of the attention heads in the first layer during training. (a) Progression of the test loss during training. The highlighted points are the iterations on the plateaus for which we demonstrate the attention matrices. (b) The evolution of attention scores of the heads of the simplified transformer architecture during training representing the tokens it is attending. First, both of the attention heads attend to all the previous tokens uniformly. At the second plateau, they both attend to the previous token. Finally, as the model escapes this plateau, the second attention head learns to attend to $(-2)$-token at the end of training.

and train for $2^{14}$ iterations using the Adam optimizer with a constant learning rate of $0.01$ and a batch size of $128$. No weight decay is used. The test loss is evaluated on a separate set of $2^{16}$ sequences.

**Discussion.** To accurately predict the next token, the model needs to attend to the previous $n - 1 = 2$ tokens in the sequence. Figure 3a shows that learning occurs in distinct phases, where the model remains in a plateau for an extended period before quickly jumping to the next one. In Figure 3b, we illustrate the evolution of the attention maps from both heads in the first layer at various plateaus during training, providing a fine-grained view of the structure of the model at these stages. Initially, the attention maps are uniform, and the model does not consider token history in its predictions. Upon reaching the first plateau, both attention heads focus on the previous token (the $(-1)$-token), and the model behaves like a bigram estimator at this stationary point. In the later stages of training, the second attention head learns to shift its focus to the $(-2)$-token, transforming the model into a trigram estimator. The same phenomenon holds for the general attention-only transformers, see Fig 5.

## 6. Conclusion

In this work, we investigate the problem of learning in-context $n$-grams with transformers, specifically focusing on a simplified yet insightful setting to gain a deeper understanding of the dynamics of complex large language models. We first constructed specific parameter configurations

within a simplified Transformer architecture that provably implement sub-$n$-gram estimators. Our main theoretical result then establishes that these configurations correspond to near-stationary points of the population cross-entropy loss. This finding creates a mechanistic link between the geometry of the loss landscape and the empirical observation of training plateaus corresponding to sub-$n$-grams: the model's learning trajectory naturally pauses at these regions of vanishingly small gradient, which represent mastery of simpler, sub-syntactic skills, before eventually escaping to learn more complex dependencies.

Our analysis provides a foundational step, and several exciting avenues for future research remain. The theoretical results are derived under idealized conditions, namely for the population loss, in the limit of infinite sequence length, and without regularization. Extending these findings to the finite-sample regime and incorporating practical considerations like finite context windows and the effect of weight decay is an important direction. Furthermore, our work focuses on the static properties of the loss landscape. A complementary and crucial line of inquiry involves analyzing the training dynamics themselves. Characterizing how stochastic gradient-based methods navigate this structured landscape to escape plateaus and transition between solutions, as explored by Nichani et al. (2024), presents a key challenge for future research.

## Impact Statement

This paper presents work whose goal is to advance the field of Machine Learning. There are many potential societal consequences of our work, none of which we feel must be specifically highlighted here.

## Acknowledgements

This work was supported by the Swiss National Science Foundation (Grant No. 212111) and an unrestricted gift from Google. A.V. acknowledges funding from a Swiss Data Science Center Fellowship. The authors extend their gratitude to Adway Girish for his valuable feedback on the manuscript and to anonymous reviewers for their insightful comments, which significantly improved the final version.

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

# A. Organization of the Supplementary Material

## A.1. Links to Materials Referenced in The Main Text.

First, we present an index of the supporting material referenced in the main text.

- The proofs of Lemma 3.1 and Proposition 3.2 are provided in Section C and the discussion on the stationarity condition for subsequences beyond suffixes is provided at Remark C.2.

- The proof of the construction of $k$-grams is given in Subsection B.1 and the proof of Theorem 4.1 is provided in Subsection B.2.

- The discussion on the possible alternate representation for the stationary distribution is provided in Section F, and the extension for a general transformer architecture is given in Subsection F.1. Stationary points conditioned on subsequences that are not suffixes for a fixed $T$ are further discussed in the Subsection F.2

## A.2. Outline of the Supplementary Material.

- Section B, provides the construction of the $k$-grams and the proof of the stationarity of the construction (in Subsection B.1 and Subsection B.2 respectively).

- Section C provides the proofs of the results on the gradient of the cross-entropy and sufficient stationary conditions, i.e., of Lemma 3.1 and Proposition 3.2.

- Section D provides a technical lemma related to computing the gradient for two-layer simplified transformer.

- In Section E, a technical lemma related to the $k$-gram representations is provided.

- In Section F, we discuss the extensions of the results to a general transformer architecture in Subsection F.1 and the stationary points conditioned on subsequences that are not suffixes in Subsection F.2.

- In Section G, we present the derivatives of a single layer self-attention map.

## A.3. Notation and Definitions

**Notations.** We use $\otimes$ to denote the Kronecker product. We use $\mathrm{vec}$ operator for flattening the matrix to a vector. We use $\mathsf{e}_i^d$ to denote the $i^{\text{th}}$ elementary basis vector of $\mathbb{R}^d$. We drop the superscript when $d = \mathcal{S}$ and $e_i$ denotes the $i^{\text{th}}$ elementary basis vector of $\mathbb{R}^\mathcal{S}$.

**Definition A.1** (Jacobian of a function). Let $f : \mathbb{R}^{m \times n} \to \mathbb{R}^p$ be a $C_1$-function defined on a variable $X$. $\frac{\partial f}{\partial X}$ denotes the Jacobian which is a function from $\mathbb{R}^{m \times n} \to \mathbb{R}^{p \times mn}$.

**Definition A.2.** Define $\boldsymbol{\theta}_*^k = \left\{ \mathbf{A}_1^{(h)}, \mathbf{V}_1^{(h)} \right\}_{h \in [n-1]} \cup \{\mathbf{K}_2, \mathbf{Q}_2\}$ as the set of parameters given by the following expressions

$$(\mathbf{Q}_2)^\top = \sqrt{c} \sum_{j=1}^{\mathcal{S}} \sum_{h=1}^{k-1} s_j^{h-1}(s_j^h)^\top, \tag{10a}$$

$$\mathbf{K}_2 = \sqrt{c} \sum_{j=1}^{\mathcal{S}} \sum_{h=1}^{k-1} s_j^h(s_j^h)^\top, \tag{10b}$$

$$\mathbf{A}_1^{(h)} = \begin{cases} c\left( \sum_{l=h}^{T-1} \mathsf{e}_l^T \left(\mathsf{e}_{l-h}^T\right)^\top + \sum_{l=0}^{h-1} \mathsf{e}_l^T \left(\mathsf{e}_0^T\right)^\top \right), & \text{for } h \in [k-1], \\ \quad\quad\quad\text{arbitrary} & \text{otherwise}, \end{cases} \tag{10c}$$

$$\mathbf{V}_1^{(h)} = \begin{cases} \sum_{j=1}^{\mathcal{S}} s_j s_j^\top \text{ for } h \in [k-1], \\ 0 \quad \text{o.w.} \end{cases} . \tag{10d}$$

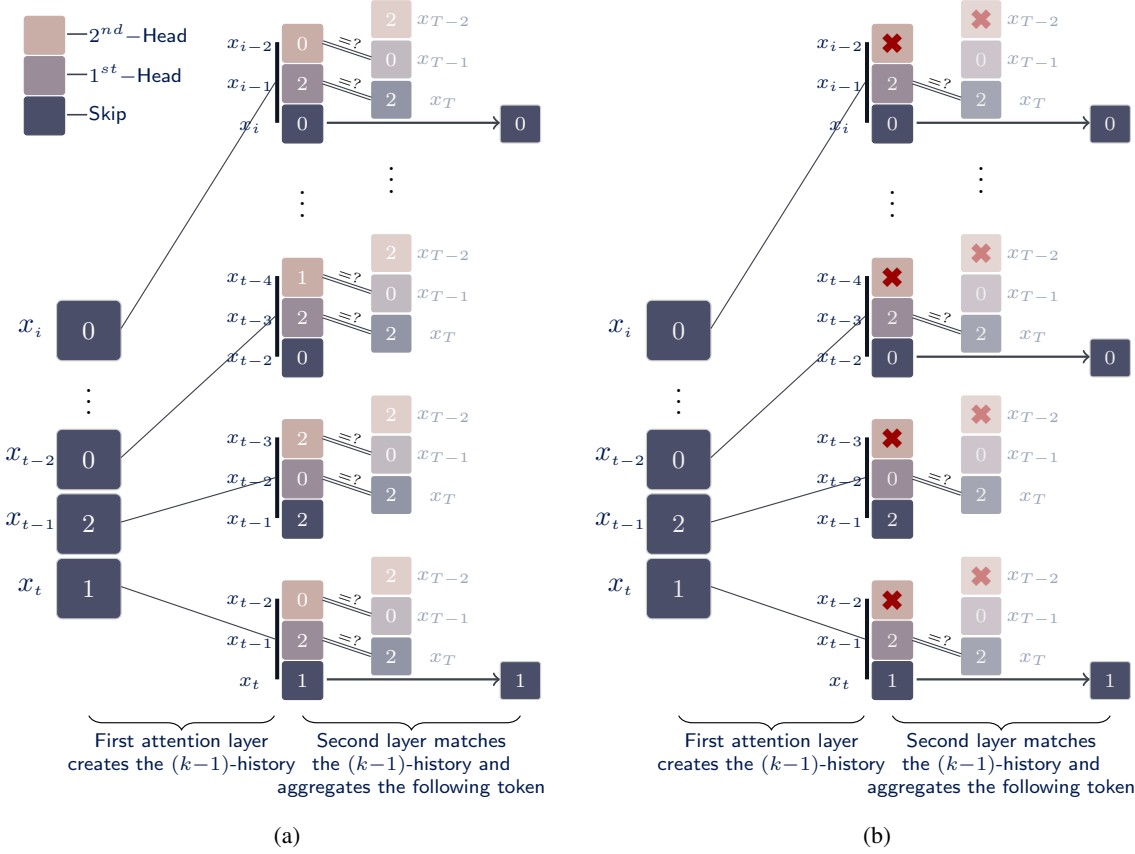

(a)                                                  (b)

Figure 4: **Transformer representing a $n$-gram and $k$-gram estimator.** (a) The task we consider task is learning in-context tri-grams ($n=3$). Here, we illustrate how the transformer given by $\boldsymbol{\theta}_*^n$ constructed in section 4.1 operates on the sequence $(\ldots, 2, x_i=0, \ldots, 2, 0, 2, x_t=1, \ldots 0, x_T=2)$ and computes a tri-gram($k=3$) estimator. Skip, $1^{st}$-Head, $2^{nd}$-Head denotes the outputs of the skip connection, $1^{st}$ and the $2^{nd}$ head. The first layer creates the $(2)-$history and the second layer compares it with the $(2)-$history of token $T+1$, i.e, $x_T = 2$. The second layer compares these histories and attends to tokens at $t, i$ and averages the tokens at that position. (b) ✖ represents that $2^{nd}$ head is deactivated. Hence, the first layer creates the $(1)-$history and the second layer matches with the $(1)-$history of token $T+1$, i.e, $x_T=2$ and attends and averages the tokens at $t, t-2, i$.

## B. Construction and the Stationarity of the $k$-gram solutions

In this section, we present the proofs of the technical lemmas and theorems presented in the main text. First, we present the parameter configurations representing $k$-grams and then prove the stationarity of these configurations.

### B.1. Representing $k$-grams with Simplified Transformer

**Forward pass of the transformer.** Before presenting the proofs, we give an alternate form of the forward pass of the simplified transformer. First, we express the embeddings after the first layer as follows:

$$r_1[i] = \mathcal{W}_o^{(0)} r_0[i] + \sum_{h=1}^{n-1} \mathcal{W}_o^{(h)} \sum_{j=1}^{i} a_{(j,i)}^{(1,h)} \mathbf{V}_1^{(h)} r_0[j],$$

where $a_{(j,i)}^{(1,h)}$ denotes the attention scores for key $j$ and query $i$ of the attention head $h$ in the $1^{\text{st}}$ layer. The matrices $\mathcal{W}_o^{(h)}$ are used for concatenation. Formally, the attention scores and the matrices are given as follows:

$$a_{(j,i)}^{(1,h)} = \frac{\exp\{\mathbf{A}_1^{(h)}[i,j]\}}{\sum_{l=1}^{i} \exp\{\mathbf{A}_1^{(h)}[i,l]\}},$$

$$\mathcal{W}_o^{(h)} = \sum_{j=1}^{\mathcal{S}} s_j^h s_j^\top.$$

For the $t^{\text{th}}$ token, the output embedding of the second layer writes

$$r_2[t] = \sum_{i=1}^{t} a_{(i,t)}^{(2)} r_0[i],$$

where $a_{(i,t)}^{(2)}$ denotes the attention scores in the second layer for key $i$ and query $t$, and are given by

$$a_{(i,t)}^{(2)} = \frac{\exp\langle \mathbf{K}_2 r_1[i], \mathbf{Q}_2 r_1[t]\rangle}{\sum_{j=1}^{t} \exp\langle \mathbf{K}_2 r_1[j], \mathbf{Q}_2 r_1[t]\rangle}. \tag{11}$$

The final output probabilities of the token $t$ after the unembedding are given by

$$p_{\boldsymbol{\theta}}(x^t) = U r_2[t] = \sum_{i=1}^{t} a_{(i,t)}^{(2)} U r_0[i] = \sum_{i=1}^{t} a_{(i,t)}^{(2)} e_{x_i}.$$

$\boldsymbol{\theta}_k^*$ **represents a $k$-gram.** Moving forward we provide the proof for the $k$-gram MLE constructions. To support the proofs, we define a subset of tokens, $\mathsf{M}_t \subseteq [t]$, which consists of tokens whose $k$-history matches the $k$-history of the $(t+1)^{\text{th}}$ token. Formally, it is defined as follows:

$$\mathsf{M}_t^k = \left\{ i : \mathbb{1}\left(x_{i-k+1}^{i-1} = x_{t-k+2}^t\right) \right\}.$$

We begin with a formal lemma to show that $\boldsymbol{\theta}_k^*$ represents a $k$-gram in the asymptotic limit where $c \to \infty$. To make the lemma easier to read on its own, we have restated the standard notations.

**Lemma B.1.** *Let $a_{(j,i)}^{(1,h)}$ denote the attention score of the key and query element $(i,j)$ in the $h^{th}$ head of the first layer and let $a_{(i,t)}^{(2)}$ denote the attention score between elements $i,t$ in the second layer. Let $\mathsf{M}_t^k = \left\{i \in [k,t] : \left(x_{i-k+1}^{i-1} = x_{t-k+2}^t\right)\right\}$ be the set of tokens which match the $k$-history of the $(t+1)^{th}$ token.*

*For the parameters $\boldsymbol{\theta}_*^k$ defined in Eq. (8), in the limit $c \to \infty$, the attention scores of the activated heads in the first layer, i.e., for heads $h$ where $h \leqslant k-1$, are given by,*

$$a_{(j,i)}^{(1,h)} = \begin{cases} 1 & \text{when } i > h \text{ and } j = i - h, \\ 1 & \text{when } i \leqslant h \text{ and } j = 1, \\ 0 & \text{otherwise} \end{cases}, \tag{12}$$

*and the attention scores in the second layer are given by*

$$a_{(i,t)}^2 = \begin{cases} \frac{1}{|\mathsf{M}_t^k|} & \text{for } i \in \mathsf{M}_t^k, \\ 0 & \text{otherwise} \end{cases}. \tag{13}$$

In order to prove the above lemma, we provide two lemmas one for each layer of the transformer, which detail the attention scores the respective layers compute. The lemmas below are non-asymptotic in $c$ and provides the bounds on the pertubation in the case of finite $c$. Intuitively, the attention scores of the $h^{\text{th}}$ head in the first layer are computed such that the each token attends to its corresponding $(-h)$-th token, The attention scores of the second layer ensures that $t^{\text{th}}$ token attends tokens whose $k$-history matches the $k$-history of the $(t+1)^{\text{th}}$ token. The $\mathcal{O}$ notation hides the terms polynomial in $c$.

**Lemma B.1.** *[First-Layer]  With the set of parameter $\boldsymbol{\theta}_k^*$ given in Def. A.2,*

*(a) The attention score of head $h$ in layer 1 of key $i$ and query $j$ denoted by $a_{(j,i)}^{(1,h)}$ is*

$$a_{(j,i)}^{(1,h)} = \begin{cases} 1 - \mathcal{O}(ie^{-c}) \text{ when } i \geqslant h \text{ and } j = i - h, \\ 1 - \mathcal{O}(ie^{-c}) \text{ when } i < h \text{ and } j = 0, \\ \mathcal{O}(e^{-c}) \text{ o.w..} \end{cases} \quad . \tag{14}$$

*(b) The first layer outputs the embeddings:*

$$r_1[i] = \begin{cases} s_{x_i}^0 + \sum_{h=1}^{k-1} s_{x_{i-h}}^h + \mathcal{O}(ie^{-c}) \cdot \mathbf{1} \quad \text{for } i \geqslant k-1, \\ s_{x_i}^0 + \sum_{h=1}^{i} s_{x_{i-h}}^h + \sum_{h=i+1}^{k-1} s_{x_0}^h + \mathcal{O}(ie^{-c}) \cdot \mathbf{1} \quad \text{for } i < k-1. \end{cases} \quad . \tag{15}$$

**Lemma B.2.** *[Second-Layer]  With the set of parameter $\boldsymbol{\theta}_k^*$ given in Def. A.2, the attention scores of the second layer are*

$$a_{(i,t)}^{(2)} = \begin{cases} \frac{1}{|\mathsf{M}_t^k|} - \frac{\mathcal{O}(te^{-c})}{|\mathsf{M}_t^k|^2} \text{ for } i \in \mathsf{M}_t, \\ \frac{\mathcal{O}(e^{-c})}{|\mathsf{M}_t^k|} \text{ o.w.} \end{cases} \tag{16}$$

.

For the proof of the above lemmas, see Section E.

**Proof of the construction for $k$-gram MLE in the limit $c \to \infty$.**   We give the proof of construction for any $k$ and when $k = n$ we get the $n$-gram estimator. The final output probabilities writes

$$p_{\boldsymbol{\theta}}(x^t) = \sum_{i=1}^{t} a_{(i,t)}^{(2)} e_{x_i}.$$

In the limit $c \to \infty$, the attention scores from the second layer from Lemma B.2 is given by,

$$a_{(i,t)}^{(2)} = \begin{cases} \frac{1}{|\mathsf{M}_t^k|}, \quad \text{for } i \in \mathsf{M}_t^k, \\ 0 \quad \text{o.w.} \end{cases} \quad .$$

Using this the output probabilities are only supported by the tokens in $\mathsf{M}_t^k$ and is given by,

$$p_{\boldsymbol{\theta}}(x^t) = \frac{1}{|\mathsf{M}_t^k|} \sum_{i \in \mathsf{M}_t^k} e_{x_i}.$$

Now

$$p_{\boldsymbol{\theta}}(x^t)[s] = \frac{1}{|\mathsf{M}_t^k|} \sum_i \mathbb{1}\{i \in \mathsf{M}_t^k\} \mathbb{1}\{x_i = s\}.$$

which exactly matches the $k$-gram MLE estimator in Definition 2.1.

### B.2. Proof of Stationary Points with Simplified Transformer

In this subsection, we will show that the set of parameters given by $\boldsymbol{\theta}_k^*$ are indeed near-stationary by providing a bound on the norm of the gradient.

**Theorem 4.1.**  For the simplified disentangled transformer $p_{\theta}$, the following on holds on the norm of the gradient at $\boldsymbol{\theta}_*^k$

$$\|\partial_{\theta = \boldsymbol{\theta}_*^k} L(\theta)\|$$
$$= \sqrt{c} \, \mathop{\mathbb{E}}_{p_\tau \sim \mathcal{P}} \mathop{\mathbb{E}}_{x^T} \mathcal{O}(\|p_\tau(.\,|x_{T-k+2}^T) - \widehat{p}_k\|^2) + \mathcal{O}(T\sqrt{c}e^{-c}).$$

**Proof** The proof is an application of Lemma D.1 for the transformer parameters that compute the $k$-gram MLE estimator $\boldsymbol{\theta}_*^k$ given in Def. A.2.

**Parameters of the second layer.** From Lemma D.1, the derivatives of $p_{\boldsymbol{\theta}}$ with respect to the second layer are

$$\frac{\partial p_{\boldsymbol{\theta}}}{\partial \mathbf{K}_2} = \sum_{i=1}^{t} a_{(i,t)}^{(2)} \, (e_{x_i}) \otimes \text{vec}\left(\mathbf{Q}_2 r_1[t](r_1[i] - \bar{r}_t^{(1)})^{\top}\right)^{\top},$$

$$\frac{\partial p_{\boldsymbol{\theta}}}{\partial \mathbf{Q}_2} = \sum_{i=1}^{t} a_{(i,t)}^{(2)} \, (e_{x_i}) \otimes \text{vec}\left(\mathbf{K}_2 (r_1[i] - \bar{r}_t^{(1)})(r_1[t])^{\top}\right)^{\top}.$$

Before computing these quantities at $\boldsymbol{\theta}_*^k$, we gather the attention scores and weighted embeddings from Lemma B.1, B.2.

$$r_1[i] = \begin{cases} s_{x_i}^0 + \sum_{h=1}^{k-1} s_{x_{i-h}}^h + \mathcal{O}(ie^{-c}) \cdot \mathbf{1} & \text{for } i \geqslant k-1, \\ s_{x_i}^0 + \sum_{h=1}^{i} s_{x_{i-h}}^h + \sum_{h=i+1}^{k-1} s_{x_0}^h + \mathcal{O}(ie^{-c}) \cdot \mathbf{1} & \text{for } i < k-1. \end{cases} \tag{17}$$

$$a_{(i,t)}^{(2)} = \begin{cases} \frac{1}{|\mathsf{M}_t^k|} - \frac{\mathcal{O}(te^{-c})}{|\mathsf{M}_t^k|^2} & \text{for } i \in \mathsf{M}_t^k, \\ \frac{\mathcal{O}(e^{-c})}{|\mathsf{M}_t^k|} & \text{o.w.} \end{cases} \tag{18}$$

The average embedding,

$$\bar{r}_t^{(1)} = \sum_{i=1}^{t} a_{(i,t)}^{(2)} \, r_1[i] = \sum_{i \in \mathsf{M}_t^k} a_{(i,t)}^{(2)} r_1[i] + \sum_{i \notin \mathsf{M}_t^k} a_{(i,t)}^{(2)} r_1,$$

The deviation due to finite weights can be controlled as the following,

$$\left\| \sum_{i \notin \mathsf{M}_t^k} a_{(i,t)}^{(2)} r_1[i] \right\|_{\infty} = \sum_{i \notin \mathsf{M}_t^k} a_{(i,t)}^{(2)} \sup_{i \in [T]} \|r_1[i]\|_{\infty} = \frac{\mathcal{O}(te^{-c})}{|\mathsf{M}_t^k|},$$

as $\|r_1[i]\|_{\infty} \leqslant 1$ for all $i$ and $a_{(i,t)}^{(2)}$ from Eq. (18) for $i \notin \mathsf{M}_t^k$. Now, we consider the summation $\sum_{i \in \mathsf{M}_t^k} a_{(i,t)}^{(2)} r_1[i]$.

$$\sum_{i \in \mathsf{M}_t^k} a_{(i,t)}^{(2)} r_1[i] = \sum_{i \in \mathsf{M}_t^k} \left[ \frac{1}{\mathsf{M}_t^k} - \frac{\mathcal{O}(te^{-c})}{|\mathsf{M}_t^k|^2} \right] r_1[i] = \frac{1}{|\mathsf{M}_t^k|} \sum_{i \in \mathsf{M}_t^k} r_1[i] - \frac{\mathcal{O}(te^{-c})}{|\mathsf{M}_t^k|} \mathbf{1}.$$

Recall that for $i \in \mathsf{M}_t^k$, $r_1[i]$ from Eq. (17) gives

$$r_1[i] = s_{x_i}^0 + \sum_{h=1}^{k-1} s_{x_{i-h}}^h + \mathcal{O}(ie^{-c}) \cdot \mathbf{1}.$$

Now we use the definition of the set $\mathsf{M}_t^k$ to simplify the above expressions of $r_1[i]$ and $\bar{r}_t^{(1)}$. Note that $x_{i-h} = x_{t+1-h}$ for $h \in [k-1], i \in \mathsf{M}_t^k$. Using this, we have,

$$r_1[i] = s_{x_i}^0 + \sum_{h=1}^{k-1} s_{x_{t+1-h}}^h + \mathcal{O}(ie^{-c}) \cdot \mathbf{1},$$

$$\frac{1}{|\mathsf{M}_t^k|} \sum_{i \in \mathsf{M}_t^k} r_1[i] = \frac{1}{|\mathsf{M}_t^k|} \sum_{i \in \mathsf{M}_t^k} s_{x_i}^0 + \sum_{h=1}^{k-1} s_{x_{t+1-h}}^h + \mathcal{O}(te^{-c}) \cdot \mathbf{1}.$$

Combining them, we get

$$\left\| \sum_{i \in \mathsf{M}_t^k} a_{(i,t)}^{(2)} r_1[i] - \frac{1}{|\mathsf{M}_t^k|} \sum_{i \in \mathsf{M}_t^k} s_{x_i}^0 - \sum_{h=1}^{k-1} s_{x_{t+1-k}}^h \right\|_{\infty} = \mathcal{O}(te^{-c}),$$

$$\bar{r}_t^{(1)} = \frac{1}{|\mathsf{M}_t^k|} \sum_{i \in \mathsf{M}_t^k} s_{x_i}^0 + \sum_{h=1}^{k-1} s_{x_{t+1-k}}^h + \mathcal{O}(te^{-c}) \cdot \mathbf{1}$$

For $i \in \mathsf{M}_t^k$,

$$r_1[i] - \bar{r}_t^{(1)} = s_{x_i}^0 - \frac{\sum_{i \in \mathsf{M}_t^k} s_{x_i}^0}{\mathsf{M}_t^k} + \mathcal{O}(te^{-c}) \cdot \mathbf{1},$$

$$\mathbf{K}_2(r_1[i] - \bar{r}_t^{(1)}) = \mathcal{O}(t\sqrt{c}e^{-c}) \cdot \mathbf{1},$$

$$\mathbf{Q}_2(r_1[t]) = \sqrt{c} \sum_{h=1}^{k-1} s_{x_{t+1-h}}^h + \mathcal{O}(t\sqrt{c}e^{-c}) \cdot \mathbf{1}.$$

For all $i$,

$$\left\| \mathbf{Q}_2 r_1[t] (r_1[i] - \bar{r}_t^{(1)})^\top \right\|_\infty \leqslant \sqrt{c},$$

$$\left\| \mathbf{K}_2(r_1[i] - \bar{r}_t^{(1)})(r_1[t])^\top \right\|_\infty \leqslant \sqrt{c}.$$

Recalling the gradients,

$$\frac{\partial p_{\boldsymbol{\theta}}}{\partial \mathbf{K}_2} = \sum_{i=1}^{t} a_{(i,t)}^{(2)} \left( e_{x_i} \right) \otimes \mathrm{vec} \left( \mathbf{Q}_2 r_1[t](r_1[i] - \bar{r}_t^{(1)})^\top \right)^\top,$$

$$\frac{\partial p_{\boldsymbol{\theta}}}{\partial \mathbf{Q}_2} = \sum_{i=1}^{t} a_{(i,t)}^{(2)} \left( e_{x_i} \right) \otimes \mathrm{vec} \left( \mathbf{K}_2(r_1[i] - \bar{r}_t^{(1)})(r_1[t])^\top \right)^\top.$$

We split the gradients supported on $\mathsf{M}_t^k$ and its complement,

$$\frac{\partial p_{\boldsymbol{\theta}}}{\partial \mathbf{Q}_2} = \sum_{i \in \mathsf{M}_t^k} a_{(i,t)}^{(2)} \left( e_{x_i} \right) \otimes \mathrm{vec} \left( \mathbf{K}_2(r_1[i] - \bar{r}_t^{(1)})(r_1[t])^\top \right)^\top + \sum_{i \notin \mathsf{M}_t^k} a_{(i,t)}^{(2)} \left( e_{x_i} \right) \otimes \mathrm{vec} \left( \mathbf{K}_2(r_1[i] - \bar{r}_t^{(1)})(r_1[t])^\top \right)^\top,$$

$$= \mathcal{O}(t\sqrt{c}e^{-c}) \cdot \mathbf{1} + \frac{\mathcal{O}(t\sqrt{c}e^{-c}) \cdot \mathbf{1}}{|\mathsf{M}_t^k|}.$$

The final gradient,

$$\frac{\partial p_{\boldsymbol{\theta}}}{\partial \mathbf{Q}_2} = \mathcal{O}(t\sqrt{c}e^{-c}) \cdot \mathbf{1} + \frac{\mathcal{O}(t\sqrt{c}e^{-c}) \cdot \mathbf{1}}{|\mathsf{M}_t^k|}. \tag{19}$$

On the similar lines,

$$\frac{\partial p_{\boldsymbol{\theta}}}{\partial \mathbf{K}_2} = \sqrt{c} \frac{1}{|\mathsf{M}_t^k|} \sum_{i \in \mathsf{M}_t^k} e_{x_i} \otimes \mathrm{vec} \left( \left[ \sum_{h=1}^{k-1} s_{x_{t+1-h}}^h \right] \left[ s_{x_i}^0 - \frac{\sum_{i \in \mathsf{M}_t^k} s_{x_i}^0}{|\mathsf{M}_t^k|} \right]^\top \right) + \mathcal{O}(t\sqrt{c}e^{-c}) \cdot \mathbf{1}.$$

Denote

$$\phi(x_{t-k+2}^t) := \sum_{h=1}^{k-1} s_{x_{t+1-h}}^h,$$

$$\bar{s}_t^0 := \frac{1}{|\mathsf{M}_t^k|} \sum_{i \in \mathsf{M}_t^k} s_{x_i}^0.$$

Note that the first term in the above expression is only a function of $k$-history $x_{t-k+2}^t$. Using this,

$$\frac{\partial p_{\boldsymbol{\theta}}}{\partial \mathbf{K}_2} = \sqrt{c} \frac{1}{|\mathsf{M}_t^k|} \sum_{i \in \mathsf{M}_t^k} e_{x_i} \otimes \mathrm{vec} \left( \phi(x_{t-k+2}^t) \left[ s_{x_i}^0 - \bar{s}_t^0 \right]^\top \right) + \mathcal{O}(t\sqrt{c}e^{-c}) \cdot \mathbf{1},$$

$$= c^{1/2} \frac{1}{|\mathsf{M}_t^k|} \sum_{a \in \mathcal{S}} \#\{a \in \mathsf{M}_t^k\} \left( e_a \right) \otimes \mathrm{vec} \left( \phi(x_{t-k+2}^t) \left( s_a^0 - \bar{s}_t^0 \right) \right)^\top + \mathcal{O}(t\sqrt{c}e^{-c}) \cdot \mathbf{1}.$$

Note that the second term $\bar{s}_t^0$ is a function of the $k$-gram MLE $\widehat{p}_k$, precisely,

$$\widehat{p}_k[a] = \frac{\#\{a \in \mathsf{M}_t^k\}}{|\mathsf{M}_t^k|}.$$

Using this $\bar{s}_t^0 = S^0 \widehat{p}_k$ where $S^0$ is the $\mathbb{R}^{nd \times nd}$ matrix where the first block is the embedding matrix and 0's everywhere else. Using this

$$\frac{\partial p_{\boldsymbol{\theta}}}{\partial \mathbf{K}_2} = \sqrt{c} \sum_{a \in \mathcal{S}} \widehat{p}_k[a] \, e_a \otimes \mathrm{vec}\left( \phi(x_{t-k+2}^t) \left[ s_a^0 - S^0 \widehat{p}_k \right]^\top \right) + \mathcal{O}(t\sqrt{c}e^{-c}) \cdot \mathbf{1}. \tag{20}$$

**Parameters of the first layer.** The derivatives with respect to the first layer parameters are given by,

$$\frac{\partial p_{\boldsymbol{\theta}}}{\partial \mathbf{V}_1^{(h)}} = \sum_{i=1}^{t} a_{(i,t)}^{(2)} \left( (e_{x_i} - \bar{e}_t^{(1)})(\bar{r}_i^{(0)})^\top \right) \otimes \left( r_1[t]^\top \mathbf{Q}_2^\top \mathbf{K}_2 \mathcal{W}_o^{(h)} + (r_1[i] - \bar{r}_t^{(1)})^\top \mathbf{K}_2^\top \mathbf{Q}_2 \mathcal{W}_o^{(h)} \right),$$

$$\frac{\partial p_{\boldsymbol{\theta}}}{\partial \mathbf{A}_1^{(h)}[i,j]} = a_{(j,i)}^{(1,h)} a_{(i,t)}^{(2)} \left( e_{x_i} - \bar{e}_t^{(1)} \right) \otimes$$

$$\left( r_1[t]^\top \mathbf{Q}_2^\top \mathbf{K}_2 \mathcal{W}_o^{(h)} \mathbf{V}_1^{(h)} (r_0[j] - \bar{r}_i^{(0,h)}) + (r_1[i] - \bar{r}_t^{(1)})^\top \mathbf{K}_2^\top \mathbf{Q}_2 \mathcal{W}_o^{(h)} \mathbf{V}_1^{(h)} (r_0[j] - \bar{r}_i^{(0,h)}) \right),$$

where the averages of embeddings weighted with attention scores are given by,

$$\bar{e}_t^{(1)} = \sum_{i=1}^{t} a_{(i,t)}^{(2)} e_{x_i},$$

$$\bar{r}_i^{(0,h)} = \sum_{j=1}^{i} a_{(j,i)}^{(1,h)} r_0[j].$$

For the heads that are not activated, the derivatives with $\mathbf{A}_1^{(h)}[i,j]$ are 0, since the $\mathbf{V}_1^{(h)}$ is 0. For the activated heads, the averaged embedding is given by,

$$\bar{r}_i^{(0,h)} = \sum_{j=1}^{i} a_{(j,i)}^{(1,h)} r_0[j] = r_0[i-h] + \mathcal{O}(i \exp\{-c\}) \cdot \mathbf{1}.$$

Now, consider two cases for the derivatives.

- For $j = i - h$, $r_0[j] - \bar{r}_i^{(0,h)} = \mathcal{O}(\exp\{-c\})$, hence the derivative is $\mathcal{O}(ic \exp\{-c\})$.
- For $j \neq i - h$, due to the property of the softmax function, the attention scores are $\mathcal{O}(\exp\{-c\})$, hence the derivative is again $\mathcal{O}(c \exp\{-c\})$.

For the derivative with respect to $\mathbf{V}_1^{(h)}$, we again have two cases,

- For the non-activated heads $h \geqslant k$, $\mathbf{K}_2 \mathcal{W}_o^{(h)} = \sum_{h'=1}^{k-1} \sum_{j=1}^{\mathcal{S}} s_j^{h'}(s_j^{h'})^\top \cdot \sum_{j=1}^{\mathcal{S}} s_j^h s_j^\top = 0$. Similarly for the other term, $\mathbf{Q}_2 \mathcal{W}_o^{(h)} = 0$. Hence the derivative is 0.
- For the activated heads $h \leqslant k-1$, using the previous computations we have,

$$\mathbf{K}_2(r_1[i] - \bar{r}_t^{(1)}) = \mathcal{O}(t\sqrt{c}e^{-c}) \cdot \mathbf{1},$$

$$\mathbf{Q}_2(r_1[t]) = \sqrt{c} \sum_{h=1}^{k-1} s_{x_{t+1-h}}^h + \mathcal{O}(t\sqrt{c}e^{-c}) \cdot \mathbf{1},$$

$$\mathbf{K}_2 \mathcal{W}_o^{(h)} = \sqrt{c} \sum_{h'=1}^{k-1} \sum_{j=1}^{\mathcal{S}} s_j^{h'}(s_j^{h'})^\top \cdot \sum_{j=1}^{\mathcal{S}} s_j^h s_j^\top$$

The product is solely a function of $k$-history $x_{t-k+2}^t$ and does not depend on $i$. Computing the multiplicative factor in the front,

$$\sum_{i=1}^{t} a_{(i,t)}^{(2)} \left( (e_{x_i} - \bar{e}_t^{(1)}) (\bar{r}_i^{(0)})^\top \right) = \sum_{i \in \mathsf{M}_t^k} a_{(i,t)}^{(2)} \left( (e_{x_i} - \bar{e}_t^{(1)}) (\bar{r}_i^{(0)})^\top \right) + \mathcal{O}(\exp\{-c\})$$

Note that $\bar{r}_i^{(0)} = s_{x_{i-h}} + \mathcal{O}(i \exp\{-c\})$ and for $i \in \mathsf{M}_t^k$, we have, $s_{x_{i-h}} = s_{x_{t+1-h}}$. Using this,

$$\sum_{i=1}^{t} a_{(i,t)}^{(2)} \left( (e_{x_i} - \bar{e}_t^{(1)}) (\bar{r}_i^{(0)})^\top \right) = \sum_{i \in \mathsf{M}_t^k} a_{(i,t)}^{(2)} \left( (e_{x_i} - \bar{e}_t^{(1)}) (s_{x_{t+1-h}})^\top \right) + \mathcal{O}(\exp\{-c\}),$$

$$= \left[ \sum_{i \in \mathsf{M}_t^k} a_{(i,t)}^{(2)} (e_{x_i}) - \left( \sum_{i \in \mathsf{M}_t^k} a_{(i,t)}^{(2)} \right) \bar{e}_t^{(1)} \right] (s_{x_{t+1-h}})^\top + \mathcal{O}(\exp\{-c\})$$

The term in the square brackets is $\mathcal{O}(\exp\{-c\})$ (for hard attention it is zero). Hence the derivative with respect to $\mathbf{V}_1^{(h)}$ is $\mathcal{O}(\exp\{-c\})$.

$$\frac{\partial p_{\boldsymbol{\theta}}}{\partial \mathbf{V}_1^{(h)}} = \mathcal{O}(\exp\{-c\}) \left[ \sqrt{c} \sum_{h=1}^{k-1} s_{x_{t+1-h}}^h + \mathcal{O}(t\sqrt{c}e^{-c}) \cdot \mathbf{1} \right]. \tag{21}$$

**Final Gradients**  Bringing things together, from Equations (19), (20), (21),

$$\frac{\partial p_{\boldsymbol{\theta}}}{\partial \mathbf{Q}_2} = \mathcal{O}(t\sqrt{c}e^{-c}) \cdot \mathbf{1} + \frac{\mathcal{O}(t\sqrt{c}e^{-c}) \cdot \mathbf{1}}{|\mathsf{M}_t^k|},$$

$$\frac{\partial p_{\boldsymbol{\theta}}}{\partial \mathbf{V}_1^{(h)}} = \mathcal{O}(\exp\{-c\}) \left[ \sqrt{c} \sum_{h=1}^{k-1} s_{x_{t+1-h}}^h + \mathcal{O}(t\sqrt{c}e^{-c}) \cdot \mathbf{1} \right],$$

$$\frac{\partial p_{\boldsymbol{\theta}}}{\partial \mathbf{K}_2} = \sqrt{c} \sum_{a \in \mathcal{S}} \widehat{p}_k[a] \, e_a \otimes \text{vec} \left( \phi(x_{t-k+2}^t) \left[ s_a^0 - S^0 \widehat{p}_k \right]^\top \right) + \mathcal{O}(t\sqrt{c}e^{-c}) \cdot \mathbf{1}.$$

Note that the only non-vanishing gradient is the gradient with respect to $\mathbf{K}_2$. Now, the gradient can be written as

$$\frac{\partial p_{\boldsymbol{\theta}}}{\partial \mathbf{K}_2} = \sqrt{c} \sum_{a \in \mathcal{S}} \widehat{p}_k[a] \, e_a \otimes \text{vec} \left( \phi(x_{t-k+2}^t) \left[ s_a^0 - S^0 \widehat{p}_k \right]^\top \right) + \mathcal{O}(t\sqrt{c}e^{-c}) \cdot \mathbf{1}.$$

We use $p_\tau$ to denote the vector $p_\tau \left( . | x_{T-k+2}^T \right)$, the gradient using the difference as,

$$\frac{\partial p_{\boldsymbol{\theta}}}{\partial \mathbf{K}_2} = \sqrt{c} \sum_{a \in \mathcal{S}} p_\tau[a] \, e_a \otimes \text{vec} \left( \phi(x_{t-k+2}^t) \left[ s_a^0 - S^0 p_\tau \right]^\top \right) + \mathcal{O}(\|p_\tau - \widehat{p}_k\|^2) + \mathcal{O}(t\sqrt{c}e^{-c}) \cdot \mathbf{1},$$

$$= \psi(x_{t-k+2}^t) + \mathcal{O}(\|p_\tau - \widehat{p}_k\|^2) + \mathcal{O}(t\sqrt{c}e^{-c}) \cdot \mathbf{1},$$

where $\psi$ is appropriately defined. The first term here only depends on the $(k)$-history and we can use Proposition 3.2, to show that it does not contribute to the final gradient. To give the final computation,

$$\frac{\partial \mathcal{L}}{\partial \mathbf{K}_2} = \mathop{\mathbb{E}}_{p_\tau \sim \mathcal{P}} \mathop{\mathbb{E}}_{x^T \sim p_\tau} \left\langle p_\theta(x^T) - p_\tau \left( . | x^T \right), \frac{\partial \log p_\theta(x^T)}{\partial \mathbf{K}_2} \right\rangle,$$

$$= \mathop{\mathbb{E}}_{p_\tau \sim \mathcal{P}} \mathop{\mathbb{E}}_{x^T \sim p_\tau} \left\langle \widehat{p}_k - p_\tau \left( . | x^T \right), \frac{\partial \log p_\theta(x^T)}{\partial \mathbf{K}_2} \right\rangle,$$

$$= \mathop{\mathbb{E}}_{p_\tau \sim \mathcal{P}} \mathop{\mathbb{E}}_{x^T \sim p_\tau} \left\langle p_\tau \left( . | x_{T-k+2}^T \right) - p_\tau \left( . | x^T \right), \frac{\partial \log p_\theta(x^T)}{\partial \mathbf{K}_2} \right\rangle + \mathop{\mathbb{E}}_{p_\tau \sim \mathcal{P}} \mathop{\mathbb{E}}_{x^T \sim p_\tau} \| \widehat{p}_k - p_\tau \left( . | x_{T-k+2}^T \right) \| \| \frac{\partial \log p_\theta(x^T)}{\partial \mathbf{K}_2} \|,$$

$$= \mathop{\mathbb{E}}_{p_\tau \sim \mathcal{P}} \mathop{\mathbb{E}}_{x^T \sim p_\tau} \left\langle p_\tau \left( . | x_{T-k+2}^T \right) - p_\tau \left( . | x^T \right), \psi(x_{t-k+2}^t) \right\rangle + \mathop{\mathbb{E}}_{p_\tau \sim \mathcal{P}} \mathop{\mathbb{E}}_{x^T} \mathcal{O}(\| p_\tau \left( . | x_{T-k+2}^T \right) - \widehat{p}_k \|^2) \cdot \mathbf{1} + \mathcal{O}(t\sqrt{c}e^{-c}) \cdot \mathbf{1},$$

The first term vanishes due to Proposition 3.2 and this finishes the proof.

## C. Supporting Lemmas for The Derivatives of Cross-entropy Loss

In this section, we provide the proofs of the lemmas presented in Section 3 of the main text. In the end, we remark about the extension of Proposition 3.2.

**Lemma 3.1.** *Consider any parametric model $p_\theta(.) : [\mathcal{S}]^T \to \Delta^{\mathcal{S}-1}$ that maps a sequence of states to a probability vector on the states. The derivative of the population cross-entropy loss $\mathcal{L}(\theta)$ with respect to a parameter $\theta_i \in \mathbb{R}$ is*

$$\partial_{\theta_i} \mathcal{L}(\theta) = \mathbb{E}_{p_\tau \sim \mathcal{P}} \mathbb{E}_{x^T \sim p_\tau} \left\langle p_\theta(x^T) - p_\tau\left(.\,|\, x^T\right), \partial_{\theta_i} \log p_\theta(x^T) \right\rangle,$$

*where $\partial_{\theta_i}$ is the partial derivative with respect to $\theta_i$ and $\log(\cdot)$ denotes component-wise logarithm.*

**Proof** Recalling the definition of the population cross-entropy loss from Eq. (1), we have,

$$\mathcal{L}(\theta) = \mathbb{E}_{p_\tau \sim \mathcal{P}} \mathbb{E}_{x^T \sim p_\tau} \ell\left(\theta, x^T\right),$$

$$\ell\left(\theta, x^T\right) = -\sum_{s=1}^{\mathcal{S}} p_\tau(x_{T+1}=s|x^T) \log\left(p_\theta(\theta, x^T)[s]\right),$$

$$\partial_{\theta_i} \ell\left(\theta, x^T\right) = -\sum_{s=1}^{\mathcal{S}} p_\tau(x_{T+1}=s|x^T) \frac{\partial_{\theta_i} p_\theta(x^T)[s]}{p_\theta(x^T)[s]}$$

Using the fact that $p_\theta(x^T) \in \Delta^{\mathcal{S}-1}$,

$$\sum_{s=1}^{\mathcal{S}} p_\theta(x^T)[s] = 1.$$

Taking the derivative of the above expression, we have,

$$\sum_{s=1}^{\mathcal{S}} \partial_{\theta_i} p_\theta(x^T)[s] = 0.$$

Adding this to the derivative of the loss $\ell$ gives,

$$\partial_{\theta_i} \ell\left(\theta, x^T\right) = -\sum_{s=1}^{\mathcal{S}} p_\tau(x_{T+1}=s|x^T) \frac{\partial_{\theta_i} p_\theta(x^T)[s]}{p_\theta(x^T)[s]} + \sum_{s=1}^{\mathcal{S}} \partial_{\theta_i} p_\theta(x^T)[s],$$

$$= \left\langle p_\theta(x^T) - p_\tau\left(.\,|\, x^T\right), \partial_{\theta_i} \log p_\theta(x^T) \right\rangle$$

∎

**Remark C.1.** In general, a parametric model computes a function $f_\theta : [\mathcal{S}]^* \to \mathbb{R}^{\mathcal{S}}$ after which a normalizing function like soft-max is used to project it onto the simplex $\Delta^{\mathcal{S}-1}$. The partial derivative w.r.t to any parameter $\theta_i$ in this case simplifies to

$$\partial_{\theta_i} \mathcal{L}(\theta) = \mathbb{E}_{p_\tau \sim \mathcal{P}} \mathbb{E}_{x^T \sim p_\tau} \left\langle p_\theta(x^T) - p_\tau\left(.\,|\, x^T\right), \partial_{\theta_i} f_\theta(x^T) \right\rangle.$$

**Proposition 3.2.** *For any $\theta_* \in \mathbb{R}^p$ such that $\partial_{\theta=\theta_*} \log p_\theta(x^T) = g\left(p_\tau, x_t^T\right)$, i.e., the score is solely a function of the context $p_\tau$ and the last $T-t+1$ elements of the sequence $x^T$, the gradient of the population loss $\mathcal{L}$ can be written as*

$$\nabla \mathcal{L}(\theta_*) = \mathbb{E}_{p_\tau \sim \mathcal{P}} \mathbb{E}_{x^T \sim p_\tau} \left\langle p_{\theta_*}(x^T) - p_\tau\left(.\,|\, x_t^T\right), g\left(p_\tau, x_t^T\right) \right\rangle.$$

*Furthermore, if for such $\theta_* \in \mathbb{R}^p$, the model estimates the conditional probability of the next token $p_\tau\left(.\,|\, x_t^T\right)$, i.e., $p_{\theta_*}(x^T) = p_\tau\left(.\,|\, x_t^T\right)$ almost surely for $p_\tau \sim \mathcal{P}$, then $\theta_*$ is a stationary point.*

**Proof** From the above lemma, the partial derivative of the population loss is given by,

$$\partial_{\theta_i}\mathcal{L}(\theta) = \mathbb{E}_{p_\tau \sim \mathcal{P}} \mathbb{E}_{x^T \sim p_\tau} \Big\langle p_\theta(x^T) - p_\tau(.|x^T), \partial_{\theta_i} \log p_\theta(x^T) \Big\rangle,$$

Using our assumption on the score from the proposition, the above expression is rewritten as follows:

$$\mathbb{E}_{x^T \sim p_\tau} \Big\langle p_\tau(.|x^T), \partial_{\theta_i} \log p_\theta(x^T) \Big\rangle = \mathbb{E}_{x^T \sim p_\tau} \Big\langle p_\tau(.|x^T), g(p_\tau, x_t^T) \Big\rangle.$$

Now the sequence $x^T$ is split into two random variables $(x^{t-1}, x_t^T)$.

$$\mathbb{E}_{x^T \sim p_\tau} \Big\langle p_\tau(.|x^T), \partial_{\theta_i} \log p_\theta(x^T) \Big\rangle = \mathbb{E}_{(x^{t-1}, x_t^T) \sim p_\tau} \Big\langle p_\tau(.|(x^{t-1}, x_t^T)), g(p_\tau, x_t^T) \Big\rangle, \tag{22}$$

We use the following property on factorization of the expectation, for any two random variables, A, B. Let $\sigma(A), \sigma(B)$ be the support of the respective random variable, we have the following fact,

$$\mathbb{E}_{A,B} Pr\Big(\cdot \Big| (A,B)\Big) \phi(B) = \sum_{b \in \sigma(B)} \sum_{a \in \sigma(A)} Pr(A = a, B = b) Pr\Big(\cdot \Big| (A = a, B = b)\Big) \phi(B),$$

$$= \sum_{b \in \sigma(B)} Pr(B = b) \phi(B) \sum_{a \in \sigma(A)} Pr(A = a | B = b) Pr\Big(\cdot \Big| (A = a, B = b)\Big),$$

$$= \sum_{b \in \sigma(B)} Pr(B = b) \phi(B) Pr\Big(\cdot \Big| B = b\Big),$$

$$= \sum_{b \in \sigma(B)} Pr(B = b) \phi(B) Pr\Big(\cdot \Big| B = b\Big) \sum_{a \in \sigma(A)} Pr(A = a | B = b),$$

$$= \sum_{b \in \sigma(B)} \sum_{a \in \sigma(A)} Pr(A = a, B = b) \phi(B) Pr\Big(\cdot \Big| B = b\Big)$$

$$= \mathbb{E}_{A,B} Pr\Big(\cdot \Big| B\Big) \phi(B).$$

Using the above property, we can rewrite the expression Eq.(22) using $A = x^{t-1}$ and $B = x_t^T$, the per task loss can be written as,

$$\mathbb{E}_{x^T \sim p_\tau} \Big\langle p_\tau(.|x^T), \partial_{\theta_i} \log p_\theta(x^T) \Big\rangle = \mathbb{E}_{x^T \sim p_\tau} \Big\langle p_\tau(.|x_t^T), g(p_\tau, x_t^T) \Big\rangle.$$

This gives us the desired result. The remaining argument is direct: the residue vanishes when the probability estimate matches the conditional probability. ∎

**Remark C.2.** The proof of the proposition hinges on dividing the sequence $x^T$ into a prefix $x^{t-1}$ and a suffix $x_t^T$, as illustrated in Eq. (22). This particular split is chosen for clarity in presentation and discussion. However, the proof remains valid for any partition of the sequence into two disjoint subsequences, which need not be contiguous or follow a prefix–suffix structure. Consequently, the result extends to conditioning on arbitrary non-contiguous subsequences that are not necessarily suffixes.

### C.1. Conditional Probabilities: Definition and Proper Asymptotics

In the main paper, we have used a somewhat informal treatment of conditional probabilities. Here, we provide formal definitions to supplement our discussion. First, we define them for any general sequences of length $T$ for any generic probability distribution on $[\mathcal{S}]^*$.

**Definition C.3.** Given the probability distribution $p_\tau(x_1, \ldots, x_T, x_{T+1})$ of a sequence of random variables $x_1, \ldots, x_T$ taking their values in $[\mathcal{S}]$. We define the following conditional probabilities of the next token given a $(T-t)-$history is:

$$p_\tau\left(x_{T+1} = i_{T+1} \Big| x_{t+1}^T = i_{t+1}^T\right) = \frac{\sum_{i^t} p_\tau(x^{T+1} = i^{T+1})}{\sum_{i^t} p_\tau(x^T = i^T)}.$$

where the marginal distribution for sequence length $t$ is given by

$$p_\tau(x^T = i^T) = \sum_{i_T \in [\mathcal{S}]} p_\tau(x^{T+1} = i^{T+1}).$$

Under the assumption that $p_\tau$ is an $n$-gram language model, it remains to establish that the conditional probabilities are defined in a time-homogeneous manner and that the $k$-gram estimators converge to them asymptotically. These aspects are addressed in detail in Section H.

## D. Derivatives of The Simplified Transformer

In this section, we focus on the derivatives of the simplified two-layer transformer. Using the derivative of the masked self-attention map Lemma G.1, the derivatives for the two layers are computed here.

**Lemma D.1.** *Using Lemma G.1, we define the following quantities, which can be interpreted as attention-weighted embeddings for both layers.*

$$\bar{r}_t^{(1)} = \sum_{i=1}^{t} a_{(i,t)}^{(2)} r_1[i], \tag{23}$$

$$\bar{e}_t^{(1)} = \sum_{i=1}^{t} a_{(i,t)}^{(2)} e_{x_i}, \tag{24}$$

$$\bar{r}_i^{(0,h)} = \sum_{j=1}^{i} a_{(j,i)}^{(1,h)} r_0[j]. \tag{25}$$

*With the above notations the derivatives of the output probabilities after $t$ tokens with respect to the parameters of the model are given by,*

$$\frac{\partial p_{\boldsymbol{\theta}}}{\partial \mathbf{K}_2} = \sum_{i=1}^{t} a_{(i,t)}^{(2)} \left(e_{x_i}\right) \otimes \text{vec}\left(\mathbf{Q}_2 r_1[t](r_1[i] - \bar{r}_t^{(1)})^\top\right)^\top, \tag{26}$$

$$\frac{\partial p_{\boldsymbol{\theta}}}{\partial \mathbf{Q}_2} = \sum_{i=1}^{t} a_{(i,t)}^{(2)} \left(e_{x_i}\right) \otimes \text{vec}\left(\mathbf{K}_2 (r_1[t] - \bar{r}_t^{(1)})(r_1[t])^\top\right)^\top, \tag{27}$$

$$\frac{\partial p_{\boldsymbol{\theta}}}{\partial \mathbf{V}_1^{(h)}} = \sum_{i=1}^{t} a_{(i,t)}^{(2)} \left((e_{x_i} - \bar{e}_t^{(1)})(\bar{r}_i^{(0)})^\top\right) \otimes \left(r_1[t]^\top \mathbf{Q}_2^\top \mathbf{K}_2 \mathcal{W}_o^{(h)} + (r_1[i] - \bar{r}_t^{(1)})^\top \mathbf{K}_2^\top \mathbf{Q}_2 \mathcal{W}_o^{(h)}\right) \tag{28}$$

$$\frac{\partial p_{\boldsymbol{\theta}}}{\partial \mathbf{A}_1^{(h)}[i,j]} = a_{(j,i)}^{(1,h)} a_{(i,t)}^{(2)} \left(e_{x_i} - \bar{e}_t^{(1)}\right) \otimes$$

$$\left(r_1[t]^\top \mathbf{Q}_2^\top \mathbf{K}_2 \mathcal{W}_o^{(h)} \mathbf{V}_1^{(h)} \left(r_0[j] - \bar{r}_i^{(0,h)}\right) + (r_1[i] - \bar{r}_t^{(1)})^\top \mathbf{K}_2^\top \mathbf{Q}_2 \mathcal{W}_o^{(h)} \mathbf{V}_1^{(h)} \left(r_0[j] - \bar{r}_i^{(0,h)}\right)\right) \tag{29}$$

**Proof**

The final output probabilities given by the model are

$$p_{\boldsymbol{\theta}}(x^t) = U r_2[t] = \sum_{i=1}^{t} a_{(i,t)}^{(2)} U r_0[i] = \sum_{i=1}^{t} a_{(i,t)}^{(2)} e_{x_i}.$$

where the attention scores write

$$a_{(i,t)}^{(2)} = \frac{\exp\langle \mathbf{K}_2 r_1[i], \mathbf{Q}_2 r_1[t]\rangle}{\sum_{j=1}^{t} \exp\langle \mathbf{K}_2 r_1[j], \mathbf{Q}_2 r_1[t]\rangle}.$$

Using the Lemma G.1, the derivatives of the output probabilities after $t$ tokens with respect to the parameters in the second layer and the embedding of the first layer are given by,

$$\frac{\partial p_{\boldsymbol{\theta}}}{\partial \mathbf{K}_2} = \sum_{i=1}^{t} a_{(i,t)}^{(2)} \left(e_{x_i}\right) \otimes \operatorname{vec}\left(\mathbf{Q}_2 r_1[t](r_1[i] - \bar{r}_t^{(1)})^\top\right)^\top,$$

$$\frac{\partial p_{\boldsymbol{\theta}}}{\partial \mathbf{Q}_2} = \sum_{i=1}^{t} a_{(i,t)}^{(2)} \left(e_{x_i}\right) \otimes \operatorname{vec}\left(\mathbf{K}_2 (r_1[t] - \bar{r}_t^{(1)})(r_1[t])^\top\right)^\top,$$

For $i \neq t$, $\quad \dfrac{\partial r_2[t]}{\partial r_1[i]} = a_{(i,t)}^{(2)} \left(e_{x_i} - \bar{e}_t^{(1)}\right) \otimes \left(\mathbf{K}_2^\top \mathbf{Q}_2 r_1[t]\right)^\top,$

$$\frac{\partial r_2[t]}{\partial r_1[t]} = a_{(t,t)}^{(2)} \left(e_{x_t} - \bar{e}_t^{(1)}\right) \otimes \left(\mathbf{K}_2^\top \mathbf{Q}_2 r_1[t]\right)^\top + \sum_{i=1}^{t} a_{(i,t)}^{(2)} \left(e_{x_i} - \bar{e}_t^{(1)}\right) \otimes \left(\mathbf{Q}_2^\top \mathbf{K}_2 (r_1[i] - \bar{r}_t^{(1)})\right)^\top.$$

The output embeddings of the first attention layer,

$$r_1[i] = \mathcal{W}_o^{(0)} r_0[0] + \sum_{h=1}^{n-1} \sum_{j=1}^{i} a_{(j,i)}^{(1,h)} \mathcal{W}_o^{(h)} \mathbf{V}_1^{(h)} r_0[j],$$

$$\text{where} \quad a_{(j,i)}^{(1,h)} = \frac{\exp \mathbf{A}_1^{(h)}[i,j]}{\sum_{l=1}^{i} \exp \mathbf{A}_1^{(h)}[i,l]}, \quad \mathcal{W}_o^{(h)} = \sum_{j=1}^{\mathcal{S}-1} s_j^h s_j^\top.$$

The Jacobian of the first layer embeddings $r_1$ w.r.t to attention and value matrix of the head $h$,

$$\frac{\partial r_i[i]}{\partial \mathbf{V}_1^{(h)}} = (\bar{r}_i^{(0)})^\top \otimes \mathcal{W}_o^{(h)}, \quad \frac{\partial r_i[i]}{\partial \mathbf{A}_1^{(h)}[i,j]} = a_{(j,i)}^{(1,h)} \mathcal{W}_o^{(h)} \mathbf{V}_1^{(h)} \left(r_0[j] - \bar{r}_i^{(0,h)}\right).$$

Combining them, we get,

$$\frac{\partial p_{\boldsymbol{\theta}}}{\partial \mathbf{V}_1^{(h)}} = \sum_{i=1}^{t} \frac{\partial p_{\boldsymbol{\theta}}}{\partial r_1[i]} \frac{\partial r_1[i]}{\partial \mathbf{V}_1^{(h)}}$$

$$= \sum_{i=1}^{t-1} a_{(i,t)}^{(2)} \left[\left(e_{x_i} - \bar{e}_t^{(1)}\right) \otimes \left(\mathbf{K}_2^\top \mathbf{Q}_2 r_1[t]\right)^\top\right] (\bar{r}_i^{(0)})^\top \otimes \mathcal{W}_o^{(h)}$$

$$+ \sum_{i=1}^{t} a_{(i,t)}^{(2)} \left(e_{x_i} - \bar{e}_t^{(1)}\right) \otimes \left(\mathbf{Q}_2^\top \mathbf{K}_2 (r_1[i] - \bar{r}_t^{(1)})\right)^\top (\bar{r}_i^{(0)})^\top \otimes \mathcal{W}_o^{(h)},$$

$$= \sum_{i=1}^{t} a_{(i,t)}^{(2)} \left((e_{x_i} - \bar{e}_t^{(1)})(\bar{r}_i^{(0)})^\top\right) \otimes \left(r_1[t]^\top \mathbf{Q}_2^\top \mathbf{K}_2 \mathcal{W}_o^{(h)} + (r_1[i] - \bar{r}_t^{(1)})^\top \mathbf{K}_2^\top \mathbf{Q}_2 \mathcal{W}_o^{(h)}\right),$$

$$\frac{\partial p_{\boldsymbol{\theta}}}{\partial \mathbf{A}_1^{(h)}[i,j]} = \frac{\partial p_{\boldsymbol{\theta}}}{\partial r_1[i]} \frac{\partial r_1[i]}{\partial \mathbf{A}_1^{(h)}[i,j]},$$

$$= a_{(i,t)}^{(2)} a_{(j,i)}^{(1,h)} \left(e_{x_i} - \bar{e}_t^{(1)}\right) \otimes$$

$$\left(r_1[t]^\top \mathbf{Q}_2^\top \mathbf{K}_2 \mathcal{W}_o^{(h)} \mathbf{V}_1^{(h)} (r_0[j] - \bar{r}_i^{(0,h)}) + (r_1[i] - \bar{r}_t^{(1)})^\top \mathbf{K}_2^\top \mathbf{Q}_2 \mathcal{W}_o^{(h)} \mathbf{V}_1^{(h)} (r_0[j] - \bar{r}_i^{(0,h)})\right).$$

In the above simplifications, we have used the following property of Kronecker product,

$$(A \otimes B)(C \otimes D) = (AC) \otimes (BD),$$

for matrices $A, B, C, D$ of appropriate dimensions. $\blacksquare$

# E. Proofs of Representation with Simplified Transformers

**Lemma B.1.** *[First-Layer] With the set of parameter $\theta_k^*$ given in Def. A.2,*

*(a) The attention score of head $h$ in layer 1 of key $i$ and query $j$ denoted by $a_{(j,i)}^{(1,h)}$ is*

$$
a_{(j,i)}^{(1,h)} = \begin{cases} 1 - \mathcal{O}(ie^{-c}) \ when \ i \geqslant h \ and \ j = i - h, \\ 1 - \mathcal{O}(ie^{-c}) \ when \ i < h \ and \ j = 0, \\ \mathcal{O}(e^{-c}) \ o.w.. \end{cases} \tag{14}
$$

*(b) The first layer outputs the embeddings:*

$$
r_1[i] = \begin{cases} s_{x_i}^0 + \sum_{h=1}^{k-1} s_{x_{i-h}}^h + \mathcal{O}(ie^{-c}) \cdot \mathbf{1} & for \ i \geqslant k-1, \\ s_{x_i}^0 + \sum_{h=1}^{i} s_{x_{i-h}}^h + \sum_{h=i+1}^{k-1} s_{x_0}^h + \mathcal{O}(ie^{-c}) \cdot \mathbf{1} & for \ i < k-1. \end{cases} \tag{15}
$$

**Proof** Recall that the attention scores of head $h$ in the first layer for key $i$ and query $j$ are given by

$$
a_{(j,i)}^{(1,h)} = \frac{\exp \mathbf{A}_1^{(h)}[i,j]}{\sum_{l=1}^{i} \exp \mathbf{A}_1^{(h)}[i,l]},
$$

For the **activated head** $h$, the $\mathbf{A}_1^{(h)}$ is given by Eq. (8a),

$$
\mathbf{A}_1^{(h)} = c \sum_{l=h}^{T-1} e_l e_{l-h}^\top + c \sum_{l=0}^{h} e_l e_0^\top
$$

Using the above two equations for $i \geqslant h$, the attention scores can be simplified as,

$$
\mathbf{A}_1^{(h)}[i,j] = e_i^\top \left[ c \sum_{l=h}^{T-1} e_l e_{l-h}^\top + c \sum_{l=0}^{h} e_l e_0^\top \right] e_j = c \, \mathbb{1}\{j = i - h\},
$$

$$
\sum_{j=1}^{i} \exp \mathbf{A}_1^{(h)}[i,j] = (i-1) + e^c,
$$

$$
\text{For } j \neq i - h, \quad a_{(j,i)}^{(1,h)} = \frac{1}{(i-1) + e^c} = \mathcal{O}(\exp\{-c\}),
$$

$$
\text{and, for } j = i - h, \quad a_{(j,i)}^{(1,h)} = \frac{e^c}{(i-1) + e^c} = 1 - \mathcal{O}(i \exp\{-c\}).
$$

Coming to the embeddings after the first layer,

$$
r_1[i] = \mathcal{W}_o^{(0)} r_0[0] + \sum_{h=1}^{n-1} \sum_{j=1}^{i} a_{(j,i)}^{(1,h)} \, \mathcal{W}_o^{(h)} \mathbf{V}_1^{(h)} r_0[j],
$$

where,

$$
\mathcal{W}_o^{(h)} = \sum_{j=1}^{\mathcal{S}} s_j^h s_j^\top.
$$

Note that for $h \geqslant k$ the term $\mathbf{V}_1^{(h)} = 0$, and for $h < k$, the term $\mathbf{V}_1^{(h)}$ is given by Eq. (8b). Using the above two equations and $r_0[i] = s_{x_i}$, we can write the embeddings after the first layer as,

$$
r_1[i] = \mathcal{W}_o^{(0)} s_{x_i} + \sum_{h=1}^{n-1} \sum_{j=1}^{i} a_{(j,i)}^{(1,h)} \, \mathcal{W}_o^{(h)} \mathbf{V}_1^{(h)} s_{x_j}.
$$

Note that $\mathbf{V}_1^{(h)} s_{x_j} = s_{x_j}$ for $h < k$ and $\mathcal{W}_o^{(h)} s_{x_j} = s_{x_j}^h$. Furthermore, for $j \in [\mathcal{S}]$,

$$\|s_j\|_\infty \leqslant 1, \text{ and } \|\mathbf{V}_1^{(h)} s_j\|_\infty \leqslant 1.$$

Using this, the above equation can be written as,

$$r_1[i] = s_{x_i}^0 + \sum_{h=1}^{k-1} s_{x_{i-h}}^h + \mathcal{O}(ie^{-c}) \cdot \mathbf{1}.$$

A similar computation for $i < h$ gives the required result. ∎

**Lemma B.2.** *[Second-Layer]* *With the set of parameter $\boldsymbol{\theta}_k^*$ given in Def. A.2, the attention scores of the second layer are*

$$a_{(i,t)}^{(2)} = \begin{cases} \frac{1}{|\mathsf{M}_t^k|} - \frac{\mathcal{O}(te^{-c})}{|\mathsf{M}_t^k|^2} \text{ for } i \in \mathsf{M}_t, \\ \frac{\mathcal{O}(e^{-c})}{|\mathsf{M}_t^k|} \text{ o.w.} \end{cases} \tag{16}$$

.

**Proof** The product of the query and key in the second layer from Eq. 8c is given by,

$$(\mathbf{Q}_2)^\top \mathbf{K}_2 = c \sum_{h=1}^{k-1} \sum_{j=1}^{\mathcal{S}} s_j^{h-1} (s_j^h)^\top.$$

Using this the attention scores in the second layer for any key $t$ and query $i$ are given by,

$$a_{(i,t)}^{(2)} = \frac{\exp \langle \mathbf{K}_2 r_1[i], \mathbf{Q}_2 r_1[t] \rangle}{\sum\limits_{j=1}^{t} \exp \langle \mathbf{K}_2 r_1[j], \mathbf{Q}_2 r_1[t] \rangle}.$$

To compute the inner product,

$$\langle \mathbf{K}_2 r_1[i], \mathbf{Q}_2 r_1[t] \rangle = \langle r_1[t], \mathbf{Q}_2^\top \mathbf{K}_2 r_1[i] \rangle$$

We know from Lemma B.1 that the embeddings after the first layer for $i \geqslant k$ are given by,

$$r_1[i] = \sum_{h=0}^{k-1} s_{x_{i-h}}^h + \mathcal{O}(ie^{-c}) \cdot \mathbf{1}.$$

Now for $t \geqslant i$,

$$(\mathbf{Q}_2)^\top \mathbf{K}_2 r_1[i] = c \sum_{h=1}^{k-1} \sum_{j=1}^{\mathcal{S}} s_j^{h-1} (s_j^h)^\top \left[ \sum_{h=0}^{k-1} s_{x_{i-h}}^h + \mathcal{O}(ie^{-c}) \cdot \mathbf{1} \right],$$

$$= c \sum_{h=1}^{k-1} s_{x_{i-h}}^{h-1} + \mathcal{O}(ie^{-c}) \cdot \mathbf{1}.$$

$$\langle r_1[t], (\mathbf{Q}_2)^\top \mathbf{K}_2 r_1[i] \rangle = c \left\langle \sum_{h=0}^{k-1} s_{x_{t-h}}^h, \sum_{h=1}^{k-1} s_{x_{i-h}}^{h-1} \right\rangle + \mathcal{O}(te^{-c}),$$

$$= c \left\langle \sum_{h=1}^{k} s_{x_{t+1-h}}^{h-1}, \sum_{h=1}^{k-1} s_{x_{i-h}}^{h-1} \right\rangle + \mathcal{O}(te^{-c}),$$

Now we use the fact that the embeddings $s_i^{h_1}, s_j^{h_2}$ are orthogonal for $i \neq j$ or $h_1 \neq h_2$. Using this, we can write the above expression as,

$$\langle r_1[t], (\mathbf{Q}_2)^\top \mathbf{K}_2 r_1[i] \rangle = c \sum_{h=1}^{k-1} \left\langle s_{x_{t+1-h}}^{h-1}, s_{x_{i-h}}^{h-1} \right\rangle + \mathcal{O}(te^{-c}),$$

$$= c \sum_{h=1}^{k-1} \mathbb{1}\{x_{i-h} = x_{t+1-h}\} + \mathcal{O}(te^{-c}).$$

If the $k$-history of $t + 1$ and $i$ match, i.e., $i \in \mathsf{M}_t^k$, then the summation is maximum at $(k-1)c^2$ otherwise it will be $\leqslant (k-2)c^2$ as there is atleast one mismatch. Using this, we can write the attention scores as,

$$\sum_{j=1}^{t} \exp \langle \mathbf{K}_2 r_1[j], \mathbf{Q}_2 r_1[t] \rangle \leqslant |\mathsf{M}_t| \exp\{(k-1)c\} + (t - |\mathsf{M}_t|) \exp\{(k-2)c\}.$$

Hence, the attention scores are given by neglecting the terms with double exponentiation (i.e., $\exp\{\exp\{-c\}\}$),

For $i \in \mathsf{M}_t^k$, $\quad a_{(i,t)}^{(2)} = \dfrac{\exp{(k-1)c^2}}{\displaystyle\sum_{j=1}^{t} \exp \langle \mathbf{K}_2 r_1[j], \mathbf{Q}_2 r_1[t] \rangle} \geqslant \dfrac{\exp{(k-1)c^2}}{|\mathsf{M}_t| \exp\{(k-1)c\} + (t - |\mathsf{M}_t|) \exp\{(k-2)c\}},$

$$\frac{1}{|\mathsf{M}_t|} - a_{(i,t)}^{(2)} \leqslant \frac{(t - |\mathsf{M}_t|) \exp\{(k-2)c\}}{|\mathsf{M}_t| \left[|\mathsf{M}_t| \exp\{(k-1)c\} + (t - |\mathsf{M}_t|) \exp\{(k-2)c\}\right]} \leqslant \frac{t - |\mathsf{M}_t|}{|\mathsf{M}_t|^2} \exp\{-c\}$$

For $i \notin \mathsf{M}_t^k$, $\quad a_{(i,t)}^{(2)} \leqslant \dfrac{\exp{(k-2)c}}{|\mathsf{M}_t| \exp{(k-1)c}} = \mathcal{O}(\exp\{-c\}) \quad$ o.w..

Hence, the attention scores are given by,

$$\text{For } i \in \mathsf{M}_t^k, \quad a_{(i,t)}^{(2)} = \frac{1}{|\mathsf{M}_t^k|} - \frac{t\mathcal{O}(\exp\{-c\})}{|\mathsf{M}_t^k|^2},$$

$$\text{For } i \notin \mathsf{M}_t^k, \quad a_{(i,t)}^{(2)} = \mathcal{O}(\exp\{-c\}) \quad \text{o.w..}$$

This completes the proof of the lemma. ∎

## F. Possible Extensions of The Results

In this section, we discuss the possible extensions of the results presented in the main text. First we discuss how the results can be extended to a general transformer architecture. Then we discuss the possible constructions for stationary points and beyond suffixes.

**Other stationary points.** Before we begin, we note that there are other possible constructions of the stationary points. In the main text, we focus on the set of parameter configurations where for each $h$ in $[1, k-1]$ a single activated head computes the $(-h)$-token, while the other heads are turned off. However, there can be multiple heads dedicated to computing the $(-h)$-tokens for $h \leqslant k - 1$. Using symmetry, the current approach can be extended to show that the gradients of parameters of these heads vanish, similar to the proof of Theorem 4.1.

For example, consider the bigram MLE estimator; in the construction given in Eq. (10) a head is activated to compute the $(-1)-$token and the other heads are turned off. Alternatively, there exist parameter configurations where all heads are activated and compute the $(-1)-$token similar to Figure 3. Our proof of Theorem 4.1 works in this case too, with the arguments of symmetry, the gradients of these heads will vanish. This can be shown using the same arguments as in the proof of Theorem 4.1. We give an explicit construction for this simple bigram case.

$$(\mathbf{Q}_2)^\top = \sqrt{c} \sum_{j=1}^{\mathcal{S}} \sum_{h=1}^{n-1} s_j^0 (s_j^h)^\top, \tag{30a}$$

$$\mathbf{K}_2 = \sqrt{c} \sum_{j=1}^{\mathcal{S}} \sum_{h=1}^{n-1} s_j^h (s_j^h)^\top, \tag{30b}$$

$$\mathbf{A}_1^{(h)} = c \sum_{l=1}^{T-1} \mathbf{e}_{l-1}^T \left( \mathbf{e}_l^T \right)^\top \text{ for } h \in [n-1], \tag{30c}$$

$$\mathbf{V}_1^{(h)} = \sqrt{c} \sum_{j=1}^{\mathcal{S}} s_j s_j^\top \text{ for } h \in [n-1]. \tag{30d}$$

Here, all heads are switched on and the $(-1)$−token is computed by all the heads. The query is used to compare the $x_t$ token with all the previous heads (notice 0 in blue in the superscript for all $h$ in $\mathbf{Q}_2$). This can be generalized to any $k$-gram, where the $(-h)$-token for $h < k$ is computed by different heads with varying multiplicities.

### F.1. General Transformer Architecture

The results presented in the main text are given for a simplified transformer architecture. In this section, we discuss how the results can be extended to a general transformer architecture. We provide brief proof sketches for two extensions: adding a value matrix to the second layer and replacing the concatenation of head embeddings and the first-layer skip connection.

**Token and position embeddings.** First we define some block embeddings in a dimension $\mathbb{R}^d$ and later lift them into the dimension of the transformers, i.e., $nd$ and sequences upto length $T$.

$$\text{Token embeddings} \to s_0, s_1, \dots, s_{\mathcal{S}-1} \in \mathbb{R}^d$$
$$\text{Position embeddings} \to p_0, p_1, \dots, p_{T-1} \in \mathbb{R}^d.$$

The embeddings are mutually orthogonal, i.e.,

$$s_i \perp s_j, \text{ for all } i \neq j \in [N].$$
$$p_i \perp p_j, \text{ for all } i \neq j \in [T].$$
$$p_i \perp s_j, \text{ for all } i \in [T], \ j \in [N].$$

For $h \in [k+1]$, we denote the $s_i^h, p_i^h \in \mathbb{R}^{nd}$, the lift of the block embeddings which are defined as the following,

$$s_i^h = x \begin{bmatrix} \overbrace{0_d \ 0_d \ \cdots}^{h \text{ blocks}} & s_i & \cdot & \cdot \end{bmatrix}$$
$$p_i^h = \begin{bmatrix} \underbrace{0_d \ 0_d \ \cdots}_{h \text{ blocks}} & p_i & \cdot & \cdot \end{bmatrix}$$

The embedding layer maps to $q_0 \in \mathbb{R}^{T \times nd}$ where each row $i$ is the embedding of $i^{\text{th}}$ element in the sequence along with its positional embedding, i.e., $q_0[i] = s_{x_i}^0 + p_i^0 \in \mathbb{R}^{nd}$.

**The attention layers.** The first layer has $k$ attention heads and also has a skip connection. For a head $h \in [k]$, let $Q_1^{(h)}, K_1^{(h)}, V_1^{(h)}$ denote the query, key and value matrices. The forward pass on $q_0$ writes

$$r_1 = r_0 + \sum_{h=1}^{k} \boldsymbol{\sigma} \left( r_0 \, (\mathbf{Q}_1^{(h)})^\top \mathbf{K}_1^{(h)} \, r_0^\top \right) \, r_0 \, \left( \mathbf{V}_1^{((h))} \right)^\top,$$

Here $\boldsymbol{\sigma}(.)$ is a row-wise softmax operator with a causal masking. Note that here we are adding in comparison to the simplified architecture where we are concatenating.

The second layer has just one head and also no skip connection (the skip connection can be included and the value matrix should be scaled appropriately so that it is ignored after normalization). Let $K_2, Q_2, V_2$ be the key, query and value matrices of the second layer. The forward pass for the second layer writes

$$r_2 = \boldsymbol{\sigma} \left( r_1 \ (\mathbf{Q}_2)^\top \mathbf{K}_2 \ r_1^\top \right) \ r_1 \ (\mathbf{V}_2)^\top .$$

Now that $r_2$ is not row wise normalized and the result does not give an output vector on the simplex, we have to normalize them using a softmax operator, i.e., $\boldsymbol{\sigma}(r_2)$ to get the final output. In this case, the counting algorithm (MLE) implemented by the attention-only transformer previously discussed cannot compute logits of the probability $\log p_\tau$, so that softmax normalization over $\log p_\tau$ gives the true probability distribution $p_\tau$. Hence, an MLP layer is used to compute the logits.

**An MLP layer.** As discussed above, there is an MLP layer that computes the logarithm of its input. Let $M$ be the set of parameters of the MLP layer and $m$ be the function implemented, $m(.) : \mathbb{R}^\mathcal{S} \to \mathbb{R}^\mathcal{S}$ does component wise logarithm $m(x) = \log(|x| + \epsilon)$ for some small epsilon.

Writing it together, the forward pass of the transformer writes,

$$r_1 = r_0 + \sum_{h=1}^k \boldsymbol{\sigma} \left( r_0 \ (\mathbf{Q}_1^{(h)})^\top \mathbf{K}_1^{(h)} \ r_0^\top \right) \ r_0 \ \left( \mathbf{V}_1^{((h))} \right)^\top ,$$
$$r_2 = \boldsymbol{\sigma} \left( r_1 \ (\mathbf{Q}_2)^\top \mathbf{K}_2 \ r_1^\top \right) \ r_1 \ (\mathbf{V}_2)^\top ,$$
$$r_2^+ = m(Ur_2),$$
$$p_{\boldsymbol{\theta}}(x^t) = \boldsymbol{\sigma}(r_2^+)$$

This model generalizes the simplified version by adding a value matrix in the second layer, moving beyond simple concatenation for head embeddings and the first-layer skip connection, and explicitly using positional encoding. While the simplified model's representation results extend to this generalized version, proving that the gradients vanish requires additional work, particularly for the value matrices.

For the $t^{\text{th}}$ token, the output after the second layer of the transformer can be written as,

$$r_2[t] = \sum_{i=1}^t a_{(i,t)}^{(2)} \mathbf{V}_2 r_1[i],$$

where the attention scores in the second layer $a_{(i,t)}^{(2)}$ for key $i$ and query $t$ are given by,

$$a_{(i,t)}^{(2)} = \frac{\exp \langle \mathbf{K}_2 r_1[i], \mathbf{Q}_2 r_1[t] \rangle}{\sum_{j=1}^t \exp \langle \mathbf{K}_2 r_1[j], \mathbf{Q}_2 r_1[t] \rangle} .$$

The final output probabilities are given by

$$p_{\boldsymbol{\theta}}(x^t) = Ur_2[t] = \sum_{i=1}^t a_{(i,t)}^{(2)} Ur_0[i] = \sum_{i=1}^t a_{(i,t)}^{(2)} e_{x_i}.$$

Here, we show that the gradient with respect to $\mathbf{V}_2$ vanishes. The others follow the same pattern as the simplified model and will be not be precisely computed here. The partial derivative with respect to gradient gives us,

$$\frac{\partial r_2[t]}{\partial \mathbf{V}_2} = \sum_{i=1}^t a_{(i,t)}^{(2)} \ I_\mathcal{S} \otimes r_1[i], = I_\mathcal{S} \otimes \bar{r}_t^{(1)}$$

Define $\theta_*^k$ by

$$(Q_1^{(h)})^\top K_1^{(h)} = \begin{cases} c_h \sum_{l=h}^{T-1} p_{l-h} p_l^\top & \text{where} \quad \text{for } h \in [k-1] \\ 0 & \text{o.w.} \end{cases}, \tag{32}$$

$$V_1^{(h)} = \begin{cases} \sum_{j=1}^{\mathcal{S}} s_j^h (s_j^0)^\top, & \text{for } h \in \mathcal{N} \\ 0 & \text{o.w.} \end{cases}, \tag{33}$$

$$(Q_2)^\top K_2 = c \sum_{j=1}^{\mathcal{S}} \sum_{h \in \mathcal{N}} s_j^{h-1} (s_j^h)^\top, \tag{34}$$

$$V_2 = \sum_{j=1}^{\mathcal{S}} e_j (s_j^0)^\top. \tag{35}$$

Intuitively, this is just an expansion on simplified transformer with lifting in the first layer and multiplication with value made explicit. Recalling $\bar{s}_t^0 = \frac{1}{|\mathsf{M}_t^k|} \sum_{i \in \mathsf{M}_t^k} s_{x_t}^0$, using this,

$$\bar{r}_t^{(1)} = \sqrt{c} \bar{s}_t^0 + \sum_{h=1}^{k-1} s_{x_{t+1-h}}^h + \mathcal{O}(\exp\{-c\}).$$

At the limit $T \to \infty$, $\bar{s}_t^0$ depends only on the context and $(k-1)$-history of the token $t+1$ and the other summation only depends on the $(k-1)$-history of the token $t+1$. Hence, the gradient vanishes as $c \to \infty$. This gives a glimpse to how the computation can be extended to a general transformer architecture.

### F.2. Other Constructions for Stationary points and Beyond Contigous History

Note that we only consider estimators conditioned on the contiguous suffices. Consider again the simple example of bigram, which is conditioning on a single element in the $n$-history. There can be many such estimators, for example, conditioning on the first element in the $n$-history $p(x_n = .|x_{n-1})$, the second element in the $n$-history $p(x_n = .|x_{n-2})$, and so on. It is natural to wonder if such an estimators are stationary and we can extend our results to all these possible estimators.

To generalize this, we define $\mathcal{N}-$MLE estimators indexed by a set $\mathcal{N} \subseteq [n-1]$. For any $\mathcal{N}$, we define a subsequence $x_{\mathcal{N}}^t = (x_{t-h})_{h \in \mathcal{N}}$ as $\mathcal{N}$-history and the $\mathcal{N}$-MLE estimator is given by,

$$\frac{\sum_{l=0}^{T} \mathbb{1}\{x_{\mathcal{N}}^l = x_{\mathcal{N}}^{t+1}\} \mathbb{1}\{x_l = i\}}{\sum_{l=0}^{T} \mathbb{1}\{x_{\mathcal{N}}^l = x_{\mathcal{N}}^{t+1}\}}.$$

which compares the $\mathcal{N}$-history of the token $t+1$ with the $\mathcal{N}$-history of all the tokens in the sequence.

There is a small technical challenge for a transformer to implement this. To compare the $\mathcal{N}$-history, it must compare the $(-h)$-token of token $i$ with the $(-h)$-token of the token $(t+1)$ (not $t$) for $h \in \mathcal{N}$. This essentially requires comparing the $(-h)$-token of $i$ with the $(-(h-1))$-token of $t$. As a result, an induction head must attend to the $(-(h-1))$-token in the first layer. When $\mathcal{N}$ forms a suffix, this step isn't explicitly necessary since $h-1$ will also be in $\mathcal{N}$. Otherwise, it must be computed explicitly using additional heads or changing the attention matrix specifically depending on the last token.

When the inputs are of fixed length the later approach could be used. We specify the construction first and then give the intuition on why it would work.

$$(\mathbf{Q}_2)^\top = \sqrt{c} \sum_{j=1}^{\mathcal{S}} \sum_{h \in \mathcal{N}} s_j^h (s_j^h)^\top, \tag{36a}$$

$$\mathbf{K}_2 = \sqrt{c} \sum_{j=1}^{\mathcal{S}} \sum_{h \in \mathcal{N}} s_j^h (s_j^h)^\top, \tag{36b}$$

$$\mathbf{A}_1^{(h)} = c \sum_{l=h}^{T-2} \mathbf{e}_{l-h}^T \left(\mathbf{e}_l^T\right)^\top + \mathbf{e}_{T-h+1}^T \left(\mathbf{e}_{T-1}^T\right)^\top \text{ for } h \in \mathcal{N}, \tag{36c}$$

$$\mathbf{V}_1^{(h)} = \begin{cases} \sqrt{c} \sum_{j=1}^{\mathcal{S}} s_j s_j^\top & \text{for } h \in \mathcal{N}, \\ 0 & \text{o.w.} \end{cases} \tag{36d}$$

The main modification is in the attention matrix of the first layer, i.e., now head $h$ computes $(-h)$ token for $i \neq T$ but $(-(h-1))$ token for $i = T$. The last token acts as a special token (Nichani et al., 2024) and the attention matrix is modified accordingly. The gradients of the heads will vanish similar to the proof of Theorem 4.1. However, the major drawback is that it only works for fixed length inputs and does not even work for arbitrary sequence lengths. This is not a concern for the previous constructions corresponding to contiguous suffix.

## G. Derivatives of the self attention map.

We calculate the derivatives of the masked self-attention map with respect to key, value, query and the input embeddings. Let $q \in \mathbb{R}^{T \times d}$ be the input embeddings, $Q, K, V \in \mathbb{R}^{d \times d}$ be the query, key and value matrices. Let $q_i$ denote the $i^{\text{th}}$ row of $q$. For any $t$, the output embeddings of the $t^{\text{th}}$-token of the transformer layer can be written as

$$q_t^+ = \sum_{i=1}^{t} p_i V q_i,$$

$$p_i = \frac{\exp \langle K q_i, Q q_t \rangle}{\sum\limits_{j=1}^{t} \exp \langle K q_j, Q q_t \rangle}.$$

**Lemma G.1.** *Consider a self attention map defined by*

$$q_t^+ = \sum_{i=1}^{t} p_i V q_i, \quad \text{where} \quad p_i = \frac{\exp \langle K q_i, Q q_t \rangle}{\sum\limits_{j=1}^{t} \exp \langle K q_j, Q q_t \rangle}.$$

*Define $\bar{q} = \sum_{i=1}^{t} p_i q_i$. The partial derivatives are given by,*

$$\frac{\partial q_t^+}{\partial V} = \sum_{i=1}^{t} p_i \left(q_i^\top \otimes I_d\right) = \bar{q}^\top \otimes I_d,$$

$$\frac{\partial q_t^+}{\partial K} = \sum_{j=1}^{t} p_j \left(V q_j\right) \otimes \text{vec} \left(Q q_t (q_j - \bar{q})^\top\right)^\top,$$

$$\frac{\partial q_t^+}{\partial Q} = \sum_{j=1}^{t} p_j \left(V q_j\right) \otimes \text{vec} \left(K (q_j - \bar{q}) q_t^\top\right)^\top,$$

*For $i \neq t$,* $\dfrac{\partial q_t^+}{\partial q_i} = p_i V + p_i \left(V(q_i - \bar{q})\right) \otimes \left(K^\top Q q_t\right)^\top,$

$$\frac{\partial q_t^+}{\partial q_t} = p_t V + p_t \left(V(q_t - \bar{q})\right) \otimes \left(K^\top Q q_t\right)^\top + \sum_{j=1}^{t} p_j \left(V q_j\right) \otimes \left(Q^\top K (q_j - \bar{q})\right)^\top.$$

**Proof** First taking the derivative with respect to the value $V$ gives us,

$$\frac{\partial q_t^+}{\partial V} = \sum_{i=1}^{t} p_i \left(q_i^\top \otimes I_d\right).$$

The derivative wrt to $q_i$ gives us

$$\frac{\partial q_t^+}{\partial q_i} = p_i V + \sum_{j=1}^{t} (Vq_j) \otimes \frac{\partial p_j}{\partial q_i},$$

$$\frac{\partial q_t^+}{\partial K} = \sum_{j=1}^{t} (Vq_j) \otimes \frac{\partial p_j}{\partial K},$$

$$\frac{\partial q_t^+}{\partial Q} = \sum_{j=1}^{t} (Vq_j) \otimes \frac{\partial p_j}{\partial Q}.$$

To compute the derivative of $p$ wrt $q, Q, K$, we begin with definition of intermediate functions,

$$g : \mathbb{R}^t \to \mathbb{R}^t \quad \text{where} \quad g(x) = \frac{\exp x_i}{\sum\limits_{j=1}^{t} \exp x_j},$$

$$h : \mathbb{R}^{d \times d} \times \mathbb{R}^{d \times d} \times \mathbb{R}^{t \times d} \to \mathbb{R}^t, \quad \text{where} \quad h(K, Q, q) = (\langle Kq_i, Qq_t \rangle)_{i=1}^{t}.$$

Using the above definitions, $p$ can be written as

$$p = g(h(K, Q, q)).$$

The partial derivative of $g$ writes

$$\frac{\partial g_j}{\partial x_k} = g_j \mathbb{1}\,(k = j) - g_k g_j,$$

$$\frac{\partial h_k}{\partial q_i} = \begin{cases} \mathbb{1}\{k = i\} \left(K^\top Q q_t\right)^\top, & \text{for } i \neq t, \\ \left(Q^\top K q_k\right)^\top + \mathbb{1}\{k = t\} \left(K^\top Q q_t\right)^\top, & \text{for } i = t. \end{cases},$$

$$= \mathbb{1}\{k = i\} \left(K^\top Q q_t\right)^\top + \mathbb{1}\{i = t\}(Q^\top K q_k)^\top.$$

$$\frac{\partial h_k}{\partial K} = \text{vec} \left(Q\, q_t q_k^\top\right)^\top,$$

$$\frac{\partial h_k}{\partial Q} = \text{vec} \left(K\, q_k q_t^\top\right)^\top.$$

Using the chain rule to take the derivative gives us,

$$\frac{\partial p_j}{\partial q_i} = \sum_{k=1}^{t} \frac{\partial g_j}{\partial h_k} \frac{\partial h_k}{\partial q_i},$$

$$= \sum_{k=1}^{t} \left[ p_j \mathbb{1}(k=j) - p_k p_j \right] \left[ \mathbb{1}\{k=i\} \left( K^\top Q q_t \right)^\top + \mathbb{1}\{i=t\}(Q^\top K q_k)^\top \right],$$

$$= p_j \left[ \mathbb{1}\{j=i\} \left( K^\top Q q_t \right)^\top + \mathbb{1}\{i=t\}(Q^\top K q_j)^\top \right]$$

$$- p_i p_j \left( K^\top Q q_t \right)^\top - \mathbb{1}\{i=t\} p_j \sum_{k=1}^{t} p_k (Q^\top K q_k)^\top.$$

$$\frac{\partial p_j}{\partial K} = \sum_{k=1}^{t} \frac{\partial p_j}{\partial h_k} \frac{\partial h_k}{\partial K} = \sum_{k=1}^{t} \left[ p_j \mathbb{1}(k=j) - p_k p_j \right] \text{vec} \left( Q\, q_t q_k^\top \right)^\top,$$

$$= p_j \text{vec} \left( Q\, q_t q_j^\top \right)^\top - p_j \sum_{k=1}^{t} p_k \text{vec} \left( Q\, q_t q_k^\top \right)^\top,$$

Define $\widehat{q} = \sum_{k=1}^{t} p_k q_k$,

$$\frac{\partial p_j}{\partial K} = p_j \text{vec} \left( Q q_t (q_j - \widehat{q})^\top \right)^\top.$$

$$\frac{\partial p_j}{\partial Q} = \sum_{k=1}^{t} \frac{\partial p_j}{\partial h_k} \frac{\partial h_k}{\partial Q} = \sum_{k=1}^{t} \left[ p_j \mathbb{1}(k=j) - p_k p_j \right] \text{vec} \left( K\, q_k q_t^\top \right)^\top,$$

$$= p_j \text{vec} \left( K\, q_j q_t^\top \right)^\top - p_j \sum_{k=1}^{t} p_k \text{vec} \left( Q\, q_k q_t^\top \right)^\top,$$

$$= p_j \text{vec} \left( K(q_j - \widehat{q}) q_t^\top \right)^\top.$$

Substituting the following derivates,

$$\frac{\partial p_j}{\partial K} = p_j \text{vec} \left( Q q_t (q_j - \widehat{q})^\top \right)^\top,$$

$$\frac{\partial p_j}{\partial Q} = p_j \text{vec} \left( K(q_j - \widehat{q}) q_t^\top \right)^\top.$$

$$\frac{\partial q_t^+}{\partial K} = \sum_{j=1}^{t} (V q_j) \otimes \frac{\partial K}{\partial p_j} = \sum_{j=1}^{t} p_j (V q_j) \otimes \text{vec} \left( Q q_t (q_j - \widehat{q})^\top \right)^\top,$$

$$\frac{\partial q_t^+}{\partial Q} = \sum_{j=1}^{t} (V q_j) \otimes \frac{\partial p_j}{\partial Q} = \sum_{j=1}^{t} p_j (V q_j) \otimes \text{vec} \left( K(q_j - \widehat{q}) q_t^\top \right)^\top,$$

Coming to the derivatives wrt $q_i$, we have,

$$\frac{\partial p_j}{\partial q_i} = p_j \left[ \mathbb{1}\{j=i\} \left( K^\top Q q_t \right)^\top + \mathbb{1}\{i=t\}(Q^\top K q_j)^\top \right]$$

$$- p_i p_j \left( K^\top Q q_t \right)^\top - \mathbb{1}\{i=t\} p_j \sum_{k=1}^{t} p_k (Q^\top K q_k)^\top.$$

Writing the above expression on case by case basis, we get,

$$\frac{\partial p_j}{\partial q_i} = \begin{cases} p_t(1 - p_t)\left(K^\top Q q_t\right)^\top + p_j\left(Q^\top K(q_j - \widehat{q})\right)^\top, & \text{when } i = j = t, \\ p_j(1 - p_j)\left(K^\top Q q_t\right)^\top, & \text{when } i = j \neq t, \\ -p_i p_j\left(K^\top Q q_t\right)^\top, & \text{when } i \neq j \quad \text{and} \quad i \neq t, \\ -p_t p_j\left(K^\top Q q_t\right)^\top + p_j\left(Q^\top K(q_j - \widehat{q})\right)^\top, & \text{when } i = t \quad \text{and} \quad i \neq j, \end{cases}$$

First computing the derivative wrt to $q_i$ for $i \neq t$, we get,

$$\frac{\partial q_t^+}{\partial q_i} = p_i\, V + \sum_{j=1}^{t} (V q_j) \otimes \frac{\partial p_j}{\partial q_i},$$

$$= p_i\, V + p_i\,(V q_i) \otimes \left(K^\top Q q_t\right)^\top - p_i \sum_{j=1}^{t} p_j\,(V q_j) \otimes \left(K^\top Q q_t\right)^\top,$$

$$= p_i\, V + p_i\,(V q_i) \otimes \left(K^\top Q q_t\right)^\top - p_i\,(V\widehat{q}) \otimes \left(K^\top Q q_t\right)^\top,$$

$$= p_i\, V + p_i\,(V(q_i - \widehat{q})) \otimes \left(K^\top Q q_t\right)^\top.$$

$$\frac{\partial q_t^+}{\partial q_i} = p_t\, V + p_t\,(V q_t) \otimes \left(K^\top Q q_t\right)^\top - p_t \sum_{j=1}^{t} p_j\,(V q_j)\left(K^\top Q q_t\right)^\top + \sum_{j=1}^{t} p_j\,(V q_j) \otimes \left(Q^\top K(q_j - \widehat{q})\right)^\top,$$

$$= p_t\, V + p_t\,(V(q_t - \widehat{q})) \otimes \left(K^\top Q q_t\right)^\top + \sum_{j=1}^{t} p_j\,(V q_j) \otimes \left(Q^\top K(q_j - \widehat{q})\right)^\top.$$

This proves the lemma. ∎

## H. Higher Order Markov Chain

This section first recalls key properties of higher-order Markov chains and their stationary distributions. We then leverage these properties to establish that the $k$-gram estimator has a limit as the sequence length, $T$, approaches infinity.

**Definition H.1** (Higher Order Markov Process). A Markov process $p_\tau$ of order $k$ generates a sequence of random variables $x_1, x_2, \ldots \in [\mathcal{S}]$ such that the conditional distribution of $x_{t+1}$ given $x^t$ depends only on $x_{t-k+1}^t$, i.e., for any $t \geqslant k$,

$$p_\tau\left(x_{t+1} \,\middle|\, x^t\right) = p_\tau\left(x_{t+1} \,\middle|\, x_{t-k+1}^t\right).$$

Now, we define the transition tensor with is given by $p_\tau$.

**Definition H.2** (Transition Tensor). The transition tensor of a Markov process of order $k$ is a $k$-dimensional tensor $\mathcal{P}$ such that

$$\mathcal{P}_{i^{k+1}} := p_\tau\left(x_{k+1} = i_{k+1} \,\middle|\, x^k = i^k\right).$$

Now, we lift the process into the higher dimension and write it as a simple Markov process of order 1. We create a Markov chain $\Xi$ on the space $[S]^k$ of order 1 Markov chain with transition matrix $\mathcal{T}$. Index the states by $i^k \in [S]^k$. Now the transition probabilities for this are defined as

$$\mathcal{T}\left(Y^k \,\middle|\, i^k\right) = \begin{cases} \neq 0 & \text{if } Y^k = (i_2^k, i_{k+1}), \text{ for some } i_{k+1} \in [S], \\ = 0 & \text{otherwise} \end{cases}.$$

The transition probabilities are explicitly given by

$$\mathcal{T}\left(i_2^{k+1} \,\middle|\, i^k\right) = \mathcal{P}_{i^{k+1}}.$$

**Stationary distribution.** Let $\pi \in \mathbb{R}^{|\mathcal{S}|^k}$ denote the stationary distribution of the chain $\Xi$, i.e.,

$$\pi^\top \mathcal{T} = \pi^\top.$$

In the summation form , for the coordinate indexed by $i_2^{k+1}$, we get that,

$$\sum_{j^k} \pi_{j^k} \cdot \mathcal{T}\left( i_2^{k+1} \,\middle|\, j^k \right) = \pi_{i_2^{k+1}}.$$

Note that $\mathcal{T}\left( i_2^{k+1} \,\middle|\, j^k \right) = 0$ whenever $j_2^k \neq i_2^k$. Using this, we get,

$$\sum_{j^k} \pi_{j^k} \cdot \mathcal{T}\left( i_2^{k+1} \,\middle|\, j^k \right) = \sum_{j^k} \pi_{j^k} \cdot \mathcal{T}\left( i_2^{k+1} \,\middle|\, j^k \right) \left( \mathbb{1}\left\{ j_2^k \neq i_2^k \right\} + \mathbb{1}\left\{ j_2^k = i_2^k \right\} \right),$$

$$= \sum_{j^k} \pi_{j^k} \cdot \mathcal{T}\left( i_2^{k+1} \,\middle|\, j^k \right) \mathbb{1}\left\{ j_2^k = i_2^k \right\},$$

$$= \sum_{j_1} \pi_{(j_1, i_2^k)} \mathcal{T}\left( i_2^{k+1} \,\middle|\, (j_1, i_2^k) \right),$$

$$= \sum_{i_1} \pi_{i^k} \mathcal{T}\left( i_2^{k+1} \,\middle|\, i^k \right).$$

Using this expansion, we get the following stationarity condition,

$$\pi_{i_2^{k+1}} = \sum_{i_1} \pi_{i^k} \mathcal{T}\left( i_2^{k+1} \,\middle|\, i^k \right). \tag{37}$$

### H.1. Generating Sequences with The Markov Chain $\Xi$

In this subsection, we show the subsequences of length $k$ follow the stationary distribution when generated from the Markov Chain $\Xi$. To generate sequences from the Markov Chain $\Xi$, we do the following generation,

a)  Generate a sequence of length $k$ sampling from the stationary distribution $\pi$ (say $i^k$)

b)  Sample the next token from the distribution $\mathcal{P}_{i^k, (.)}$ say $i_{k+1}$

c)  After generating a sequence $i^t$ of length $t > k$, the next token is generated by the sampling from the distribution $\mathcal{P}_{i_{t-k+1}^t, (.)}$

**Lemma H.3** (Stationarity of $k$-tuples). *Let $x^t$ be the random variable representing the sequences generated from $\Xi$, the distribution of subsequence $x_{l-k+1}^l$ of length $k$ is given by $\pi$.*

**Proof** We will prove this using induction. For the base case, observe that for $l = k$, the value is sampled from $\pi$ by construction. Use the induction hypothesis, any $k$ length subsequence of $x^l$ obeys the law given by $\pi$. It remains to show that $x_{l-k+2}^{l+1}$ are distributed according to $\pi$.

$$p_\tau\left( x_{l-k+2}^{l+1} = i_{l-k+2}^{l+1} \right) = \sum_{i^{l-k+1}} p_\tau\left( x^{l+1} = i^{l+1} \right),$$

$$= \sum_{i^{l-k+1}} p_\tau\left( x^l = i^l \right) p_\tau\left( x_{l+1} = i_{l+1} \,\middle|\, x^l = i^l \right)$$

Using the Markov property of the sequence generation, we have,

$$p_\tau\left( x_{l+1} = i_{l+1} \,\middle|\, x^l = i^l \right) = p_\tau\left( x_{l+1} = i_{l+1} \,\middle|\, x_{l-k+1}^l = i_{l-k+1}^l \right).$$

Substuting this expression, we have,

$$p_\tau\left(x_{l-k+2}^{l+1}=i_{l-k+2}^{l+1}\right)=\sum_{i^{l-k+1}}p_\tau\left(x^l=i^l\right)\mathcal{P}\left(x_{l+1}=i_{l+1}\ \middle|\ x_{l-k+1}^l=i_{l-k+1}^l\right),$$

$$=\sum_{i_{l-k+1}}\sum_{i^{l-k}}p_\tau\left(x^l=i^l\right)p_\tau\left(x_{l+1}=i_{l+1}\ \middle|\ x_{l-k+1}^l=i_{l-k+1}^l\right),$$

$$=\sum_{i_{l-k+1}}\left[\sum_{i^{l-k}}p_\tau\left(x^l=i^l\right)\right]p_\tau\left(x_{l+1}=i_{l+1}\ \middle|\ x_{l-k+1}^l=i_{l-k+1}^l\right).$$

Using the fact that $\sum_{i^{l-k}}p_\tau\left(x^l=i^l\right)=p_\tau\left(x_{l-k+1}^l=i_{l-k+1}^l\right)$. Using the induction hypothesis, we have that $p_\tau\left(x_{l-k+1}^l=i_{l-k+1}^l\right)=\pi_{i_{l-k+1}^l}$.

$$p_\tau\left(x_{l-k+2}^{l+1}=i_{l-k+2}^{l+1}\right)=\sum_{i_{l-k+1}}\pi_{i_{l-k+1}^l}p_\tau\left(x_{l+1}=i_{l+1}\ \middle|\ x_{l-k+1}^l=i_{l-k+1}^l\right),$$

$$=\sum_{i_{l-k+1}}\pi_{i_{l-k+1}^l}\mathcal{T}\left(i_{l-k+2}^{l+1}\ \middle|\ i_{l-k+1}^l\right).$$

Using the Eq. (37),

$$p_\tau\left(x_{l-k+2}^{l+1}=i_{l-k+2}^{l+1}\right)=\sum_{i_{l-k+1}}\pi_{i_{l-k+1}^l}\mathcal{T}\left(i_{l-k+2}^{l+1}\ \middle|\ i_{l-k+1}^l\right),$$

$$=\pi_{i_{l-k+2}^{l+1}}.$$

This proves the hypothesis for length $l+1$. Hence, by induction, the hypothesis holds for any length. ∎

**Lemma H.4** (Stationarity of sub-$k$-tuples)**.** *For any $t\geqslant 0$ and $l\leqslant k$, we have the following shift invariant property for the marginals,*

$$p_\tau\left(x^l=i^l\right)=p_\tau\left(x_t^{t+l-1}=i^l\right)$$

**Proof** Using Lemma H.3, we have that the distribution of the subsequence $x_t^{t+k-1}$ is given by $\pi$ for any $t\geqslant 0$, i.e.,

$$p_\tau\left(x^k=i^k\right)=p_\tau\left(x_t^{t+k-1}=i^k\right)=\pi_{i^k}.$$

Summing the above expression over all possible values of $i_{l+1}^k$, we get,

$$\sum_{i_{l+1}^k}p_\tau\left(x^k=i^k\right)=\sum_{i_{l+1}^k}p_\tau\left(x_t^{t+k-1}=i^k\right)=\sum_{i_{l+1}^k}\pi_{i^k}.$$

We know that

$$\sum_{i_{l+1}^k}p_\tau\left(x^k=i^k\right)=p_\tau\left(x^l=i^l\right),$$

$$\sum_{i_{l+1}^k}p_\tau\left(x_t^{t+k-1}=i^k\right)=p_\tau\left(x_t^{t+l-1}=i^l\right)$$

This proves the lemma. ∎

## H.2. Lower-order Conditional Probabilities

The following lemma show that the lower order conditional probabilities are time homogenous and are given by the stationary distribution $\pi$.

**Lemma H.5.** *For the sequences generated by the markov chain $\Xi$, for $l \leqslant k$, we have,*

$$p_\tau\left(x_{T+1} = . \;\middle|\; x_{T-l+1}^T = i^l\right) = p_\tau\left(x_{l+1} = . \;\middle|\; x^l = i^l\right) = \frac{\sum\limits_{i_2^{k-l}} \pi_{i_2^{k+1}}}{\sum\limits_{i^{k-l}} \pi_{i^k}}$$

**Proof** For $l = k$, this holds due the Markov property of order k and

$$p_\tau\left(x_{T+1} = . \;\middle|\; x_{T-k+1}^T = i^k\right) = p_\tau\left(x_{k+1} = . \;\middle|\; x^k = i^k\right).$$

Now, for $l < k$, we have,

$$p_\tau\left(x_{T+1} = i_{l+1} \;\middle|\; x_{T-l+1}^T = i^l\right) = \frac{p_\tau\left(x_{T-l+1}^{T+1} = i^{l+1}\right)}{p_\tau\left(x_{T-l+1}^T = i^l\right)},$$

Using the shift-invariant property of the marginals, we have that,

$$p_\tau\left(x_{T-l+1}^{T+1} = i^{l+1}\right) = p_\tau\left(x^{l+1} = i^{l+1}\right) = \sum_{i_{l+2}^k} \pi_{i^k},$$

$$p_\tau\left(x_{T-l+1}^T = i^l\right) = p_\tau\left(x^l = i^l\right) = \sum_{i_{l+1}^k} \pi_{i^k}.$$

This proves the lemma. ∎

**Conditional Probability for Any Subset of Sequence.** Previously, we defined it for contiguous history; now, we can extend the definition to any subset of sequence as well. Consider any set $\mathbf{K} \subseteq [0, \mathcal{S} - 1]$, $[\mathcal{S}] - \mathbf{K}$ is an ordered set defined as $\{k - i : i \in \mathbf{K}\}$. We define the sequence $x^{[\mathcal{S}] - \mathbf{K}}$ as the ordered set $\{x_i : i \in [\mathcal{S}] - \mathbf{K}\}$. We define the conditional probability of the next token given the sequence $x^{[\mathcal{S}] - \mathbf{K}}$ as

$$\mathcal{P}_{[\mathcal{S}] - \mathbf{K}}\left(i_{[\mathcal{S}] - \mathbf{K}}, i_{k+1}\right) := p_\tau\left(x_{k+1} = i_{k+1} \;\middle|\; x^{[\mathcal{S}] - \mathbf{K}} = i^{[\mathcal{S}] - \mathbf{K}}\right),$$

$$= \frac{p_\tau\left(x_{k+1} = i_{k+1}, x^{[\mathcal{S}] - \mathbf{K}} = i^{[\mathcal{S}] - \mathbf{K}}\right)}{p_\tau\left(x^{[\mathcal{S}] - \mathbf{K}} = i^{[\mathcal{S}] - \mathbf{K}}\right)},$$

$$= \frac{\sum\limits_{i^{([\mathcal{S}] - \mathbf{K})^c}} p_\tau\left(x^{k+1} = i^{k+1}\right)}{\sum\limits_{i^{([\mathcal{S}] - \mathbf{K})^c}} p_\tau\left(x^k = i^k\right)}.$$

**Consistency of the sub-k counting estimators.** Here, we use the ergodic properties of the chain $\Xi$ to show that the lower order estimators convergence as $T \to \infty$.

**Lemma H.6.** *Consider the $l+1$-gram estimator $\widehat{p}_l$ for $0 < l \leqslant k - 1$, then as $T \to \infty$, $\widehat{p}_l(x^T) \to p_\tau(. | x^l)$.*

**Proof** Using the ergodicity of the chain $\Xi$ (Penev, 1991), the $k+1-$gram estimator converges to the stationary distribution as it equivalent to counting the empirical frequency on the chain of higher order. Now we write the $l-$gram estimator in the following way:

$$\widehat{p}_l(i_{k+1} | i_{k-l+1}^k) = \frac{\sum_{j=1}^T \mathbb{1}\{x_{j-l+1}^{j+1} = i_{k-l+1}^{k+1}\}}{\sum_{j=1}^T \mathbb{1}\{x_{j-l+1}^j = i_{k-l+1}^k\}}$$

Now we can write the following expression as

$$\sum_{j=1}^{T} \mathbb{1}\{x_{j-l+1}^{j+1} = i_{k-l+1}^{k+1}\} = \sum_{i_2^l} \sum_{j=1}^{T} \mathbb{1}\{x_{j-k+1}^{j+1} = i_2^{k+1}\},$$

$$\sum_{j=1}^{T} \mathbb{1}\{x_{j-k+1}^{j} = i_{k-l+1}^{k+1}\} = \sum_{i^l} \sum_{j=1}^{T} \mathbb{1}\{x_{j-k+1}^{j} = i^{k}\}.$$

Now, using the ergodicity of the lifted markov chain, we can say that

$$\frac{\sum_{j=1}^{T} \mathbb{1}\{x_{j-k+1}^{j+1} = i_2^{k+1}\}}{T} \overset{t\to\infty}{\to} \pi_{i_2^{k+1}},$$

$$\sum_{i_2^l} \frac{\sum_{j=1}^{T} \mathbb{1}\{x_{j-k+1}^{j+1} = i_2^{k+1}\}}{T} \overset{t\to\infty}{\to} \sum_{i_2^l} \pi_{i_2^{k+1}}$$

Using the similar argument, the ratio can now be written as

$$\frac{\sum_{j=1}^{T} \mathbb{1}\{x_{j-l+1}^{j+1} = i_{k-l+1}^{k+1}\}}{\sum_{j=1}^{T} \mathbb{1}\{x_{j-l+1}^{j} = i_{k-l+1}^{k}\}} = \frac{\sum_{i_2^l} \pi_{i_2^{k+1}}}{\sum_{i^l} \pi_{i^k}} = p_\tau(x_{l+1} = i_{k+1} | x^l = i_{k-l+1}^{k})$$

The last equality flows from the lemma H.5. ∎

# I. More Experiments

## I.1. Experiments with Attention Only Transformer

In this section, we repeat the main experiment in the paper with the common non-disentangled transformer architecture, demonstrating the generality of our results. Figure 5, shows the evolution of attention matrices during training with both one-hot and learned embeddings for the general architecture.

For the transformer experiments, we use vocabulary size $\mathcal{S} = 3$. The length of the input sequences is $T = 32$, and the sequences are sampled from an in-context tri-gram language model, i.e., $n = 3$. The transition matrices for a fixed context are sampled from a Dirichlet prior with $\alpha = 1$. The embedding dimension is set to $d = \mathcal{S} + T$, to be consistent in both setups, and we use one-hot or learned embeddings for both positional and semantic embeddings in different experiments. The transformer is trained with Adam with a weight decay of $0.0001$ for $4096$ iterations, with a constant learning rate of $0.005$ and a batch size of $128$. The test loss is evaluated over $2^{16}$ test sequences.

## I.2. The Contiguous Solutions are Preferred during Training

In Figure 6, we repeat the procedure in Figure 3 for different seeds and plot the attention heads in the second plateau. Through all the experiments transformer more often attends to the $(-1)$-token first, rather than the $(-2)$-token. Recall that the underlying mechanism behind the sub-$n$-gram estimators is checking if the history of the token $x_{T+1}$ and $x_j$ matches and adding $x_j$ if they do. We conjecture that, since the token $x_T$ is always provided through the skip connection, it is easier to learn to match $x_T$ with $x_{j-1}$ than, for example, $x_{T-1}$ and $x_{j-2}$.

## I.3. Plot of norms of the gradients along the trajectory

In Figures 7, 8, we plot the norm of the gradient during training along the similar lines as (Odonnat et al., 2025). The plots demonstrate how the norm of gradients stays low during the plateau stages and spikes during the jump between the plateaus.

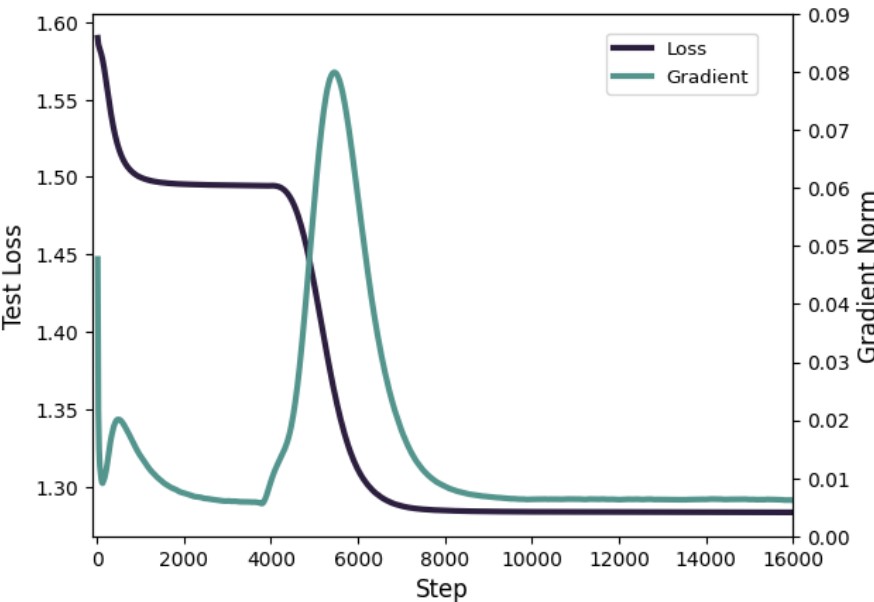

Figure 7: **Norms of gradients for simplified transformer.** A two layer simplified transformer for $S = 5$ and a sequence length of $L = 128$ from a 3-gram language model. The plots show that the norm of gradients stays low during the plateau stages and spikes during the jump between the plateaus.

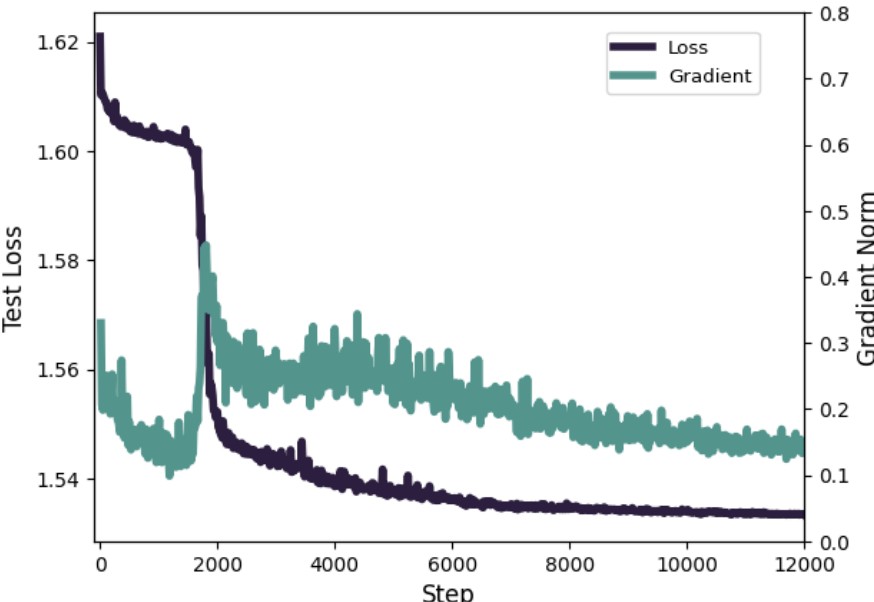

Figure 8: **Norms of gradients for transformers with MLP layers.** A two layer simplified transformer for $S = 5$ and a sequence length of $L = 64$ from a 3-gram language model. The plots show that the norm of gradients stays low during the plateau stages and spikes during the jump between the plateaus. However, there is only a single intermediate plateau, unlike attention-only transformers, as a single head attends to both the $(-1, -2)$-tokens, and it emerges after the plateau. The plateau corresponds to the unigram.

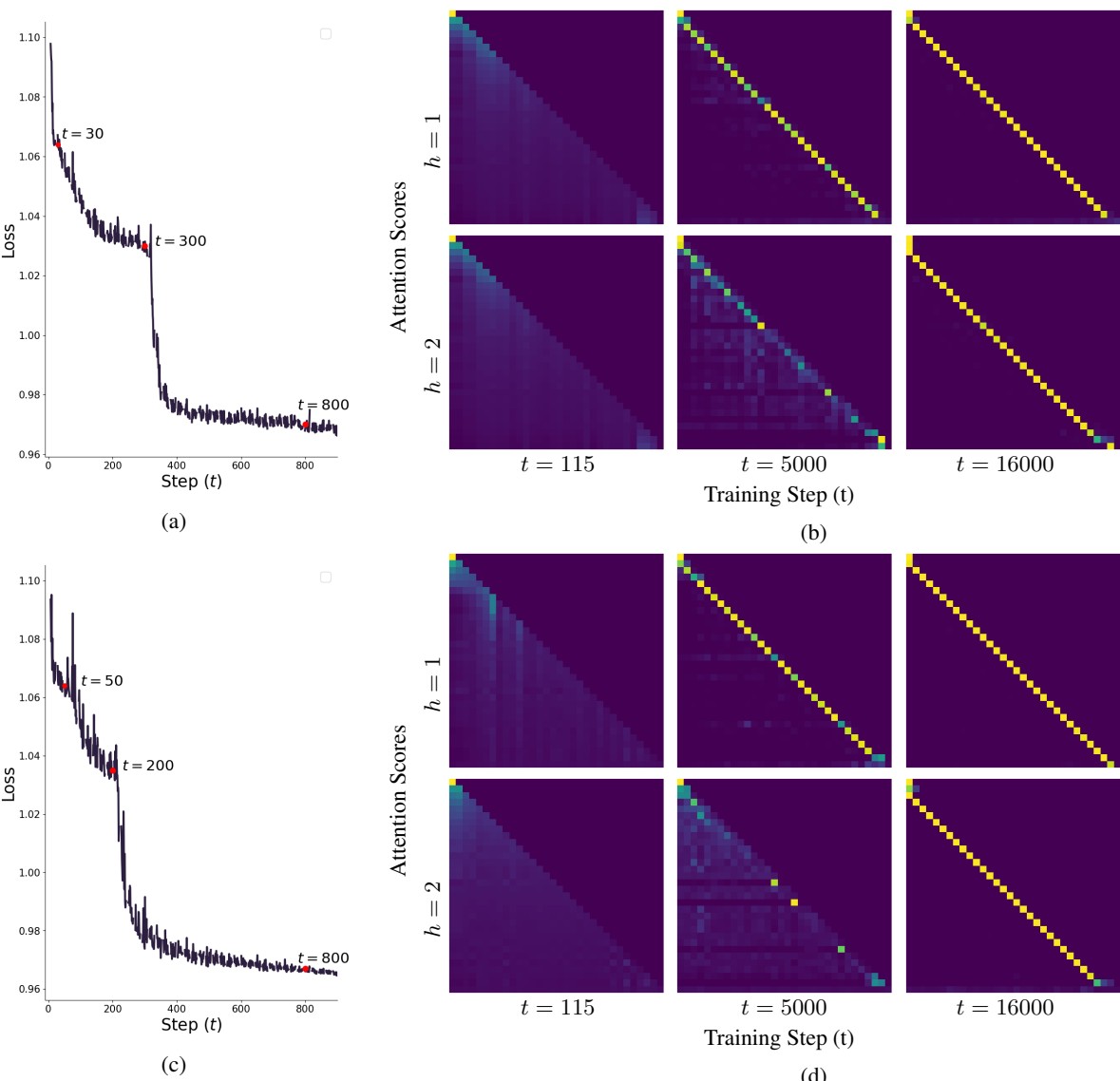

Figure 5: The evolution of the attention heads in the first layer during training of an attention-only transformer with one-hot (a-b) and learned embeddings(c-d). (a-c) Progression of the test loss during training. The highlighted points are the iterations on the plateaus for which we demonstrate the attention matrices. (b-d) The evolution of attention scores of the heads of the transformer during training representing the tokens it is attending. First, both of the attention heads attend to all the previous tokens uniformly, i.e., the induction heads are not formed. At the second plateau, they both attend to the previous token, or one head is not formed yet while the other attends to the previous token. Finally, as the model escapes this plateau, the second attention head learns to attend to $(-2)$-token at the end of training.

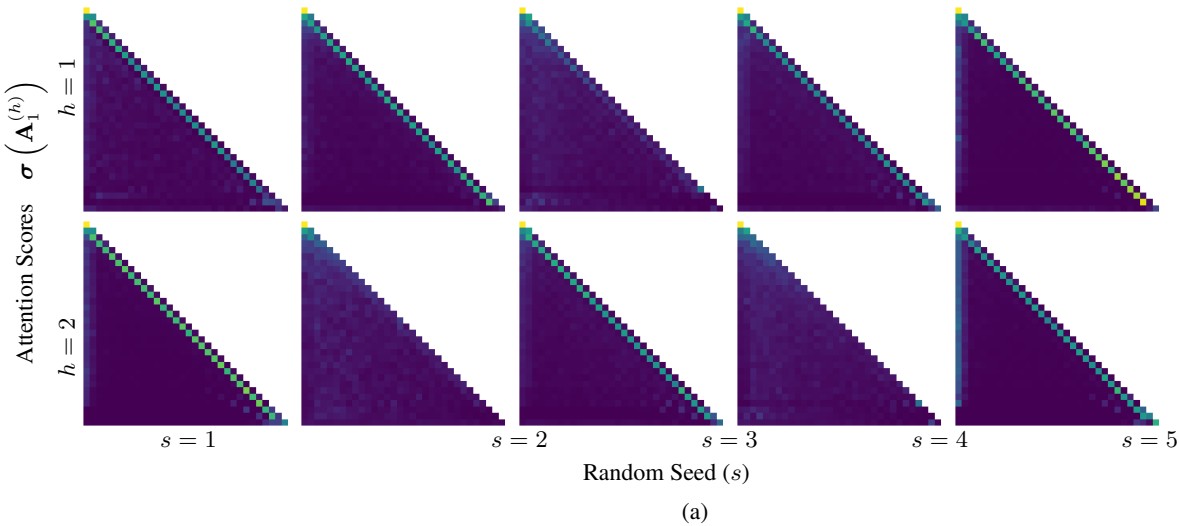

(a)

Figure 6: Attention maps of the two heads in the second plateau for different random seeds denoted by $s$. It shows how the transformers attends to $(-1)$-token first and never attends the $(-2)$-token before attending $(-1)$-token across 5 random seeds.

