# OpenReview forum: "Learning In-context $n$-grams with Transformers: Sub-$n$-grams Are Near-Stationary Points"
_ICML.cc/2025/Conference — ICML 2025 poster_

### Official Review · Reviewer_wvkz · 2025-03-10

**Overall Recommendation:** 3

**Summary:**

In this work, the authors study the loss landscape of transformers in the task of next-token prediction. To that end, they consider a simplified two-layer transformer and focus on learning n-gram language models in context with the cross-entropy loss. They rely on a constructive approach of transformer that mimics k-grams with $k \leq n$ such that the gradients of the population loss tend to zero when the sequence length goes to infinity. The authors deduce from this result that k-grams are stationary points of the population cross-entropy loss, which provides insights into the stage-wise learning dynamics and phase transition in transformers. Finally, they conduct empirical validation of their theoretical insights that illustrate the dynamics of learning n-grams that lead to jumps between stationary points.

**Claims And Evidence:**

The theoretical claims are well supported by clear and detailed proofs, and the authors also provide experimental validation of their theory in a controlled setting.

**Essential References Not Discussed:**

To the best of my knowledge, there were no essential references not discussed in the current submission. I mention below some works that I think are relevant to the current submissions and might be discussed.

- [1] demonstrated that any transformer-based LLM has an equivalent Markov chain formulation, which enabled them to study LLMs' generalization capabilities in pre-training and in-context learning on Markov chains.

- [2] studies how well n-gram statistics approximate the next-token transformer prediction, which enables the author to understand better learning dynamics and overfitting.

I acknowledge that both [1] and [2] could be considered as concurrent work, hence, I do not require the authors to discuss them. But I think that they can provide valuable context on the current submission's contributions and help argue the benefits of studying n-grams and Markov chains data to better understand transformers and LLMs.

*References*

[1] Zekri et al. Large Language Models as Markov Chains, arXiv 2024 (appeared on arXiv on 3 October 2024, according to https://arxiv.org/abs/2410.02724)

[2] Timothy Nguyen. Understanding Transformers via N-gram Statistics, NeurIPS 2024 (appeared on arXiv on 30 June 2024, according to https://arxiv.org/abs/2407.12034)

**Experimental Designs Or Analyses:**

I checked the soundess of the experiments. I believe that, while most of the contributions are theoretical, the experiments are too lightweight with a focus on small-scale, synthetic data and simplified models. I think the submission would benefit from experiments in more practical settings and/or discussions on how the current contributions extend to such scenarios.

**Methods And Evaluation Criteria:**

The authors provide theoretical results to better understand the sequential learning dynamics of transformers. They conduct toy experiments to validate their findings. Hence, the methods make sense for the problem at hand.

**Other Comments Or Suggestions:**

None

**Other Strengths And Weaknesses:**

**Strengths**

- The paper is well written with detailed prior work, clear notations, and a technical background.
- The problem tackled is interesting
- The theoretical findings provide a better understanding of the sequential learning of transformers in in-context learning of n-grams

**Weakness**

I list below what I think are weaknesses, but I would be happy to be corrected if I misunderstood some important aspects of the authors' contributions.

- I have trouble seeing how the insights provided by the authors could help in more practical scenarios since strong assumptions are made on both the task and the model. This work rather seems to provide a better understanding of the sequential learning observed in [1], which already considers a specific case of in-context learning of n-grams.

- I believe the asymptotic nature of the results on the stationary points is not quite satisfactory since, in practice, $T$ is finite and $c$ should be too (see questions section). As such, it would have been more interesting, in my opinion, to control the gradient norms to know how far from $0$ they are (see questions section)

- The experiments seem limited to a simplified model with synthetic data and vocabulary size $S=3$ and sequence length $T=32$ (far from the asymptotic regime). It would have been interesting to see how well the provided theory can explain more practical scenarios. An idea to improve the submission (inspired by [2]) could be to consider a small but realistic model (e.g., 2-layer transformer with feed-forward layers) and observe learning curves in parallel with gradient norms (easily recovered with automatic differentiation), hoping to see stage-wise learning and near-stationary points (low gradient norms) appear in tandem.

Overall, I find the paper interesting and the theory well derived. I appreciate controlled settings to conduct rigorous analysis, but I believe that the setting is too oversimplified, which impacts the submission's contributions. This is the reason for my current score but I remain open to modifying my score, provided the authors clarify the points mentioned in the weaknesses section.

**Update after rebuttal**: I increased my score from 2 to 3 based on the authors' answers to my rebuttal comments (I thank them for the additional clarifications and experiments; despite the simple setting, the authors motivate well how it can be generalized to more practical ones).

*References*

[1] Edelman et al. The Evolution of Statistical Induction Heads: In-Context Learning Markov Chains, NeurIPS 2024

[2] Odonnat et al. Clusering Head: A Visual Case Study of Training Dynamics in Transformers. arxiv 2024

**Questions For Authors:**

1) Could the authors describe practical scenarios in which the assumption of Proposition 3.2 holds (when the derivative of the model’s logits depends solely on a subsequence of the input history)?

2) The authors introduce $c$ as a constant scaling factor in Eq. (7) and (8), but I don't understand where it comes from or how it is defined. Moreover, the authors make it vary up to $\infty$ in the following theoretical results; however, in my understanding, a constant scaling factor should be fixed or depend on the problem's parameters. Could the authors elaborate on that?

3) The authors state l. 343, second column, that "the same techniques and methodology extend naturally to a general transformer architecture". I believe this is somewhat of a bold statement as it would lead to significantly more involved computations, and because of that, it is not clear, at least to me, how much of the current insights can be extended to this more general setting. Could the author elaborate on that?

4) What the authors call "near-stationary points" can be seen as to approximate stationary points, also called $\epsilon$-stationary points [1], i.e., their gradient is bounded by some $\epsilon$. In this submission, $\epsilon$ depends on how large the sequence length is since the convergence to zero is asymptotic in $T$ (and $c$). Could the author elaborate on how to control this $\epsilon$ value?

5) A naive bound on Theorem 4.1 (b) could, for instance involve Cauchy-Swartz and Jensen (since the absolute value is convex) and make three terms appear: the average difference in probabilities, the average norm of the derivative at $\theta^*$, and the term in $e^{-c}$. How do those terms depend on $T, c$ and the data? Answering this question could help to control $\epsilon$. If the authors have other ideas to bound the gradient to obtain some control over $\epsilon$, I would be happy to hear them.

6) Could the authors elaborate on how much considering the feed-forward layers would impact the current analysis?

7) In practice, language does not verify the Markov property, nor do other data modalities (e.g., tokenized code, images, and time series). How well do the insights provided by the current work translate to real-world data commonly processed with transformers?

*References*

[1] Hollender et al. The computational complexity of finding stationary points in non-convex optimization, In Mathematical Programming  2024

**Relation To Broader Scientific Literature:**

I find that related work and prior works are well introduced and compared. The submission's contributions are interesting and are part of a growing interest in the literature on the theoretical understanding of transformers. As such, the authors borrow a lot from prior works (data framework [1, 2], disantangled transformer model [2, 3] sub-gram representation [2, 4]), and some theoretical results are, as the authors acknowledge, extensions of existing results (e.g., Lemma 3.1 generalizing [5, 6]).

*References*

[1] Edelman et al. The Evolution of Statistical Induction Heads: In-Context Learning Markov Chains, NeurIPS 2024

[2] Nichani et al. How transformers learn causal structure with gradient descent, ICML 2024.

[3] Friedman et al. Learning transformer programs, NeurIPS 2023.

[4] Rajaraman et al. Transformers on Markov data: Constant depth suffices, NeurIPS 2024

[5] Bietti et al. Birth of a Transformer: A Memory Viewpoint, NeurIPS 2023

[6] Makkuva et al. Attention with markov: A framework for principled analysis of transformers via Markov chains. ICLR 2025

**Theoretical Claims:**

The theoretical findings are supported by detailed and clear proofs.

---

> ### Author Rebuttal · Authors · 2025-04-01
>
> We sincerely appreciate the detailed feedback and thoughtful suggestions provided by the reviewer.
>
> **References:**
> We thank the reviewer for bringing concurrent works Zekri et al. (2024) and Nguyen (2024) to our attention. We acknowledge their relevance to our work, especially in motivating studying learning $n$-grams with transformers and we will cite them in the final version.
>
> **Practicality of Proposition 3.2:**
> Although we formally prove that the assumptions of Proposition 3.2 hold for sub-$n$-gram constructions and in the context of learning $n$-gram language models, we conjecture that the conditions are satisfied more generally and this is the reason why there are long plateaus corresponding to **marginal estimates** of the conditional target distribution in many in-context learning setups with cross-entropy loss. One such pattern can be observed in two-hop reasoning tasks (see [1]), where there is a long plateau corresponding to an estimate of $p(end)$ (marginal estimate), where the goal is the predict $p(end|start, bridge)$ (true conditioned estimate). Another example is factuall-recall setting of [2], where the plateau is $p(.|\text{subject})$ (conditioned only on one token), where the goal is to predict  $p(.|\text{subject,relation})$ (true estimated conditioned on two tokens).
>
> **Experimental Setting:** please see reply to R. 6GFt
>
> **Scaling Factor $c$:** The scaling factor $c\in\mathbb{R}^+$, determines the norm of the parameters in the construction. Please see the response to Reviewer xuc4 that provides a formal lemma for the construction.
>
> **$\epsilon$-stationary points and error control**.  To precisely give the error bounds and delineate the dependence of the gradient on $c,T$, we give the following computation. To illustrate, we give the computation of gradient w.r.t the  key matrix of the second layer $K_2$ in $\mathbb{R}^{nd \times nd}$ .
>
> $$ \begin{align*}
> \frac{\partial p_{\theta}}{\partial K_2} = \mathbb{E}\_{\tau} \mathbb{E}\_{x^T}   \sqrt{c} \sum\_{a \in S} ( \hat{p}\_k[a] - p_{\tau} (x_{T+1} = a \big| x^T_{ \color{red} T - n + 1}) )  \phi(x\_{\color{green} T-k+2}^T) (s^0_{a} - S^{0} \hat{p}\_k )^{\top}  + { \color{blue} O( \sqrt{c} T e^{-c} )  },
> \end{align*} $$
>
> where $\hat{p}_k$ is the $k-$gram estimator defined in Eq.(2) of the main paper, $\phi = \sum\_{h=1}^{k-1} s^h\_{x\_{T+1-h}}$ and $S^0$ is a block matrix in $\mathbb{R}^{nd \times nd}$ where the first block is given by the embedding matrix S and $0$'s everywhere else. Now we can split the gradient further,
>
> $$ \begin{align*}
> \frac{\partial p_{\theta}}{\partial K_2} = \mathbb{E}\_{\tau} \mathbb{E}\_{x^T}   \sqrt{c} \sum\_{a \in S} ( p_{\tau}(x_{T+1}{=}a \big| x^T_{ \color{red} T - k + 2}) - p_{\tau} (x_{T+1}{=}a \big| x^T_{ \color{red} T - n + 1}) ) ~  \phi(x\_{\color{red} T-k+2}^T) (s^0_{a} - S^{0} p_{\tau}(x\_{T+1}{=} \cdot \big| x^T_{ \color{red} T - k + 2})  )^{\top}  + \sqrt{c}  {\color{green} \mathbb{E}\_{\tau} \mathbb{E}\_{x^T}   \\| \hat{p}\_k - p\_{\tau}(. \big| x^{T}\_{T-k+2})  \\|^2 } + { \color{blue} O( \sqrt{c} T e^{-c} )  },
> \end{align*} $$
>
> The term in blue represents the impact of finite $c$ and the term in green the impact of finite $T$.  The first term goes to zero as a result of Proposition 3.2.
>
> **Feed forward layers:** With feedforward networks, the representation question itself remains unclear—specifically, what solutions transformers learn.  One key difference observed in our experiments is that feedforward networks can solve the task with fewer heads, and each head can attend to multiple previous positions. In contrast, in attention-only transformers, each head attends to a specific previous token. Consequently, the number of possible intermediate stages is uncertain if feed-forward layers are involved. For example, in our experiments with an order-2 chain, we observe only one intermediate stage.
>
> **Generalization of our result to broader architectures:** Our proof sketch provides intuition for determining stationarity. Roughly, if the first layer computes a function $\psi(x^i_{i-k+1})$ of the $k$-history and the second layer aligns with it, the core argument—that gradients are supported by $M_t^{k}$ and that the gradient depends only on the $k$-history—can be carefully extended. In this way, we believe we can extend the analysis to general transformers. However, we tone down the claim in l.343 to say general *attention only transformers*.
>
> **Markov property,  other data modalities**
> Although we formally prove that the assumptions of Proposition 3.2 hold for sub-$n$-gram constructions and in the context of learning $n$-gram language models, the proposition itself does not require any Markov assumption and holds for all the next token prediction tasks using cross-entropy loss, e.g. tokenized code, time series or other in-context learning settings discussed above.
>
> References:
>
> [1] Guo et al. https://arxiv.org/pdf/2502.13913
>
> [2] Nischani et. al. https://arxiv.org/pdf/2412.06538

---

> > ### Comment · Reviewer_wvkz · 2025-04-03
> >
> > I thank the authors for the answers and additional experiments (please don't forget to mention [2] in the revised version as inspiration of the gradients computation (see weakness in my initial review)).  I find the experiments interesting and helpful to connect theory and practice. I list below my remaining issues:
> > 1) I still find the justification of assumptions in Proposition 3.2 unclear, although the authors demonstrate it in the n-grams case, which is rarely achieved in practice. The fact that it can be understood in other simplified settings (Nichani et al.) does not make it more valid, although I admit that it provides some context to understand it better.
> > 2) I appreciate the additional gradient computation, which might enable control over the error to achieve a perfect stationary point. However, the dependence in c, T (negative exponential for c and linear for T) makes it hard to achieve a small gradient when $T$ grows, except for very high values of c which, as the authors said, controls the norm of the parameters. I believe that such a situation is not desirable if one wants to achieve low gradients and near-stationary points. In its current form, the control over the error is not very tight, but I acknowledge the additional efforts made by the authors. I believe this shows that there remains additional work to be able to have strong claims over the near-stationarity when we are not in an asymptotic regime, which, again, does not seem viable given that $c$ is a constant scaling. If the authors wish to elaborate on that point, I will take their answer into account in my final recommendation.
> >
> > I thank the authors again for their detailed answer and will consider the additional experiments and efforts made by the authors to address my concerns, along with the other reviews (and their responses), to settle on a final recommendation and decide whether I increase my score towards 3.
> >
> > [2] Odonnat et al. Clusering Head: A Visual Case Study of Training Dynamics in Transformers. arxiv 2024

---

> > > ### Author Response · Authors · 2025-04-09
> > >
> > > We sincerely thank the reviewer for their continued and thoughtful engagement. We also appreciate the reviewer’s acknowledgment of our additional experiments and gradient computations, and we will make sure to cite Odonnat et al.(2024) in the revised version, as suggested. Below we address the remaining concerns:
> > >
> > > **On Proposition 3.2 and Its Assumptions:** We understand the reviewer’s concern that the assumptions behind Proposition 3.2 may appear restrictive in practice. While we formally verify the proposition for $n$-grams, we want to emphasize that the core insight—namely, that the gradients vanish as a result of consistent prefix marginalization—applies more broadly across multiple simplified settings observed in literature (e.g., Nichani et al., 2024; Guo et al., 2025). To this end, we would like to emphasize Reviewer xuc4’s perspective on the value of the $n$-gram case as a useful and interpretable case study. As they noted, studying such simplified constructions allows us to isolate and better understand structural progression phenomena in learning dynamics. Moreover, they acknowledge the potential for generalization, suggesting that the intuition behind our results may extend to broader classes of sequence models (e.g., HMMs, DFAs, CFGs), even if formal proofs are not yet available. This aligns with our position: while the current theory is grounded in the $n$-gram case, we believe the observed empirical patterns (e.g., persistent plateaus aligned with marginal estimates) in tasks such as two-hop reasoning and factual recall are consistent with our conjecture that similar mechanisms may be at play in more complex settings.
> > >
> > > **Revised Finite Bound:** Given the exponential decay of the last term in $c$, we had not carefully optimized for the linear factor $T$ in front of the exponential term. In response to the reviewer’s concerns, we have revisited the bound with the aim of tightening its dependence on the factor  $T$ appearing in front of $e^{-c}$. We now clarify that this factor is in fact $T / |M_k^T|$, where $|M_k^T|$ denotes the size of the set of tokens whose $k$-history matches that of ${(T+1)}^{th}$ token. In the limit as $T \to \infty$, this ratio can be upper bounded by the reciprocal of the minimum transition probability, i.e., $1 / \min \pi_{i^k}$[a]. Under the additional assumption of a uniform lower bound $\gamma$ on transition probabilities (as in Assumption 1 of [3], which holds with high probability with a Dirichlet $Dir(\alpha \mathbf{1})$ prior on the rows of the transtion matrices), the last term becomes $O(\gamma^{-1} \sqrt{c} e^{-c})$. This addresses the concern raised by that the error term grows with $T$.
> > >
> > > [a] $\pi$ is the stationary distribution of the higher-order chain and is defined in App.H.
> > >
> > > **Scale of $c$ and validity of near-stationarity:** In the attached Figure 2 (available at https://anonymous.4open.science/r/icml_rebuttal_2025-C11E/rebuttal.pdf), we plot the norms of the parameters that involve the scaling $c$ divided by the square root of their dimension. This quantity roughly captures the value of $c$ in our construction. The plots indicate that the $c$ is already large at the intermediate stationary point. The gradient is vanishing at a rate of $max\{ e^{-c},  e^{-T}\}$. High values of $c$ along with the fact that the gap with the asymptotic regime is vanishing exponentially ensure that the claim of near-stationary is valid.
> > >
> > > We hope that these clarifications, along with the previous revisions, address the remaining concerns.
> > >
> > > [3] Nichani, Eshaan, Alex Damian, and Jason D. Lee. "How transformers learn causal structure with gradient descent." arXiv preprint arXiv:2402.14735 (2024).

---

### Official Review · Reviewer_6GFt · 2025-03-12

**Overall Recommendation:** 3

**Summary:**

This paper explores the loss landscape of next-token prediction in a synthetic setup, where the model is trained to learn the transition probabilities of a Markov chain of order $n$ in-context. The authors use a simplified two-layer transformer (disentangled transformer) and theoretically analyze the population cross-entropy loss, showing that 1) there exists a parameter configuration of the model that is equivalent to a sub $k$-gram model ($k<n$), 2) any sub-$k$-gram is a stationary point of the loss in the asymptotic regime of context window $ T \to \infty $. They empirically verify these findings, showing that the plateaus in the test loss progression reported in Eldeman et al. (2024) correspond to sub-$k$-grams based on the attention maps.

**Claims And Evidence:**

The theoretical results are based on a simplified attention-only transformer where the residual connection is replaced with concatenation — a simplification used in previous works to make the analysis tractable. The experiments verify the results on both the simplified and standard attention-only transformer, suggesting that the simplification does not distort the optimization landscape much from this perspective.

To verify the claims empirically, they show that the loss plateaus during training correspond to sub-$k$-grams by confirming that the attention maps follow the construction discussed in the theory. However, verifying that the gradient of the loss around these points is near zero would further strengthen the empirical validation.

**Essential References Not Discussed:**

See above

**Experimental Designs Or Analyses:**

see above

**Methods And Evaluation Criteria:**

See above

**Other Comments Or Suggestions:**

N/A

**Other Strengths And Weaknesses:**

N/A

**Questions For Authors:**

I do find the result about the stationarity of $k$-grams a useful addition to the body of work on Markov chains for in-context learning. However, as mentioned earlier, a more detailed discussion on how your results compare and fit into the broader context of related works would help clarify the scope of the contribution. Can you comment on the results in  Rajaraman et al. (2024) and [1]?


[1] Chen S, Sheen H, Wang T, Yang Z. Unveiling induction heads: Provable training dynamics and feature learning in transformers.

**Relation To Broader Scientific Literature:**

Several recent works have investigated the optimization landscape of transformers by considering Markov chains as data models. The empirical observation of multiple loss plateaus during training on in-context Markov chains was first reported in Edelman et al. (2024), as noted in the paper. Edelman et al. (2024) also showed empirically that each plateau corresponds to a model that is close (in terms of KL divergence) to a sub-$k$-gram, but their theoretical analysis was limited.

In this sense, showing concretely that sub-$k$-grams are stationary points is a valuable follow-up to this result.

There are two related works that could benefit from further discussion on results/differnces in the paper:
- The idea of the construction of the $k$-gram solutions is very close to the construction in Rajaraman et al. (2024).
- [1] studies the gradient flow dynamics in the same Markov chain setup.

[1] Chen S, Sheen H, Wang T, Yang Z. Unveiling induction heads: Provable training dynamics and feature learning in transformers.

**Theoretical Claims:**

I followed the proof sketch in the main body and selectively reviewed parts of the supplementary material without checking all the details.

---

> ### Author Rebuttal · Authors · 2025-04-01
>
> We appreciate the reviewer for bringing Chen et al. (2024) to our attention, which we unfortunately overlooked in our discussion of related work.
>
> **Comparison with Chen et. al. 2024.** The paper examines the same task using a disentangled transformer, with the primary differences being a **three-stage training procedure** and **the inclusion of a specific FFN layer**, whereas we focus on an attention-only transformer. The key differences are as follows:
>
> - **Selection of History:** In their approach, the selection of previous $n-1$ tokens for $n$-grams is performed by an FFN layer. In contrast, in our model, this selection is handled by the attention heads in the first layer.
>
> - **Stage-Wise Dynamics:** A crucial difference is the absence of stage-wise dynamics in the analysis of Chen et al. This can be attributed to their three-stage training process and, in particular, their initialization scheme (Assumption 3.3), which ensures that different heads attend to different tokens from the start. Such an initialization eliminates stage-wise learning dynamics. In contrast, as seen in our experiments, for general initializations attention heads progressively learn to attend to new tokens at each stage. However, the initialization used by Chen et al. almost pre-configures the heads to attend to the required tokens from the outset, effectively removing the stage-wise learning process. However, Chen et. al. studies the training dynamics where our paper just studies the properties of the landscape.
>
> We will add this detailed comparison to the paper.
>
> **Comparison with Rajaraman et. al. 2024.** The method for constructing $n$-grams aligns with Rajaraman et al. (2024). However, the construction of $k$-grams (sub-$n$-grams) introduces a subtle but important modification to their  $n$-gram formulation. This difference stems from the **activation state of attention heads**—specifically, whether heads are active or inactive.  This nuance is critical for interpreting the **stage-wise dynamics** observed in experiments: unlike $n$-grams (which rely on all heads), $k$-gram formation depends on *which heads are active at intermediate stages*, as heads progressively activate during training.
>
> **Additional Experiments and General Comments on the Setting.**  We thank the reviewers for this suggestion and we provide additional experiments with the gradients for simplified and regular architectures at https://anonymous.4open.science/r/icml_rebuttal_2025-C11E/main.pdf. The reason for the choice of sequence length $T=32$ for the plots is due to the impracticality of showing larger attention maps on paper. Please see the attached figures, where we provide the loss curve and the gradient norms not only for the simplified model with longer sequence length and larger vocabulary size but also for a 2-layer transformer architecture **involving mlps**.
>
> >Reviewer xuc4
>
> **$k$-gram Construction:**
> To provide an intuitive understanding, we initially presented it in an informal manner. However, we recognize that this approach introduced several minor technical inaccuracies. To ensure rigor, we complement with the additional lemmas for $k$-gram construction which are partially stated in the App..
>
> $$
> \begin{align*}
>     A_{1}^{h} &= c \sum_{l=h+1}^{T} e_{l} e_{l-h}^{\top} + c \sum_{l = 1}^{h} e_l e_1^{\top},
>     \quad V_{1}^{h} = \sum_{j=1}^{S} s_{j} s_{j}^{\top}, \text{ for } h \leq k-1, \text{ o.w. } 0  \\\\
>     K_{2} &= \sqrt{c} \sum_{j=1}^{S}  \sum_{h = 1}^{k-1} s_{j}^{h} \left(s_{j}^{h}\right)^{\top}, \quad
>     Q_{2} = \sqrt{c} \sum_{j=1}^{S}   \sum_{h = 1}^{k-1}  s_{j}^{h} \left(s_{j}^{h-1}\right)^{\top}.
> \end{align*}
> $$
>
> **Lemma 1** For the parameters defined,
> (a) The attention score $a\_{(i,j)}^{(1,h)}$ between key and query elements $(i,j)$ of head $h$ of layer 1 is
>     $$
>     \begin{align}
>          a\_{(i,j)}^{(1,h)} = \begin{cases} 1 - O(i e^{-c}) \text{ when } i > h \text{ and } j = i-h, \\\\
>          1 - O(i e^{-c}) \text{ when } i \leq h \text{ and } j = 1, \\\\
>          O(e^{-c}) \text{ o.w.}.
>          \end{cases}.
>     \end{align} $$
>
> (b) Considering $M_k^t$ defined in the paper, the attention score $a^2\_{(t,i)}$ of layer 2 between elements $i,t$ is
> $$\begin{align*}
> a^2\_{(t,i)} = \begin{cases}
>     \frac{1}{|M^k\_t|} - \frac{O( t e^{-c})}{|M^k\_t|} \text{ for } i \in M\_t^k, \\\\
>     O(e^{-c}) \text{ o.w. }
> \end{cases}
> \end{align*}
> $$
>
> This fixes the inaccuracies identified, i.e.,
>   - The attention matrix and the attention scores after softmax are lower triangular which fixed our previous oversight where we defined it as the transpose of the current matrices. We will ensure consistency across the paper.
>    - We choose ${e\_1,\ldots, e\_T}$ as the elementary basis and fixing the notation in the summation.
>    - We also fixed the bordercase, now the token out of the range will just attend the first token $(x\_1)$ that acts as a sink.
>    - Eq (9) now holds correctly in the limit $c \to \infty$ and for tokens $t>n$.

---

> > ### Comment · Reviewer_6GFt · 2025-04-04
> >
> > Thank you for your clarification. I believe the discussion during the review process has helped clarify the paper’s contributions, both with respect to existing work and the theoretical concerns raised by other reviewers. These discussions should be added to the paper.
> >
> > Despite the limitations, I raise my score to 3 with the following perspective: Recent works have used Markov chains as a controlled testbed for analyzing the optimization dynamics of language models. While this is a simplified setup and over-relying on it may not always yield theoretical insights for real-world scenarios, I still find explanations for phenomena like stage-wise training to be a useful addition to this line of work. The theoretical limitations in the finite regime raised by other reviewers are valid. That said, the result in the asymptotic regime is sound and can be useful, as long as the concerns raised by other reviewers and the claims are properly clarified in the abstract and main body.

---

> > > ### Author Response · Authors · 2025-04-09
> > >
> > > We thank the reviewer for their follow-up and for increasing their score. We appreciate the reviewer's recognition of the value and the usefulness of the asymptotic result. We are glad that the discussion clarified our contributions and successfully addressed the concerns. We will revise the paper to reflect these discussions and ensure that distinctions and limitations are clearly communicated.

---

### Official Review · Reviewer_xuc4 · 2025-03-15

**Overall Recommendation:** 4

**Summary:**

This paper contributes the following.

1. A sufficient condition for the population cross entropy loss to vanish in
   the setting of $n^{\text{th}}$-order Markovian sequence modelling.
   The condition is based on expressing the derivative of the model's logits
   independently of  $n-1$ tokens in the input sequence, plus some other
   conditions.

2. A construction of a transformer architecture and parameter that, in the hard
   attention limit (i.e. infinite weight norm limit), implements an in-context
   $k$-gram estimator.
   The architecture is a two-layer attention-only transformer with a particular
   distribution of heads and some frozen weights.

3. (Main theorem) Putting these together, a proof that the constructed transformer parameters
   from (1) meet the sufficient condition from (2) in the infinite weight norm
   limit and the infinite sequence length limit.
   They establish that:

   1. such infinite (size and norm) parameters are first-order stationary
      points of the population cross entropy; and, moreover,

   2. in the finite regime, the sufficient condition is approximately
      satisfied, such that the constructed parameters are 'near-stationary'
      (gradient close to zero).

4. A brief discussion in the main text about how the construction and proof
   could potentially be generalised to a slightly more standard transformer
   architecture (with one-hot positional encodings and without frozen
   parameters). There is a longer discussion in appendix F.

5. A brief discussion in the main text about how the sufficient condition
   generalises to non-contiguous sub-$n$-grams
   and how the main theorem could potentially be generalised to these cases.
   There is a longer discussion in the appendices.

6. A brief numerical experiment shows that training a transformer with the
   constructed architecture exhibits several 'plateaus' separated by 'phase
   transitions' in the loss over training.

   The loss of the plateaus roughly lines up with the loss of idealised
   $k$-gram estimators for increasing $k$, and during the plateaus the
   first-layer attention patterns as predicted in the construction for these
   values of $k$.
   Therefore, the experiments show that the transformer's training trajectory
   proceeds through these 'near-stationary' points.

I wrote a pretty long review. Therefore, I include a summary of my review as
follows.

* I think this paper has the potential to make a substantial contribution to
  the field because it represents a clear analysis of a simplified example of a
  phenomenon of current interest.

* I didn't check all of the technical details of the proofs, but the results
  seem plausible to me based on my limited study (so far).

* However, I identified two major issues with the paper on the basis of which I
  have initially recommended 'weak reject:'

  1. The paper is insufficiently clear about the distinction between
     contributions 3.1 and 3.2 in my list above.

  2. The presentation of construction (contribution 3 above) contains numerous
     technical errors.

  I think these issues limit the accessibility of the contributions of the
  paper and make it not yet ready for publication.

* I detail some other minor issues and recommendations in the remainder of my
  review. If the authors show they can address the above clarity concerns, I'd
  be willing to raise my score.

**Edit to add:** Rebuttal addressed my two major concerns. Raising score to 3 for now.

**Edit to add:** After further discussion with authors and wvkz, wvkz raised score to 3. I see no remaining reasons to reject the paper (assuming the revisions proposed by the authors are correct and included in the final version). Therefore, I am raising my score to 4.

**Claims And Evidence:**

**Sufficient condition.**

I checked carefully the proofs for Lemma 3.1 and Proposition 3.2. They appear
to be sound.

1. In several places, the paper refers to this condition as a "characterization
   of the stationary points of the cross-entropy loss" or similar. This
   includes the contribution list in the introduction. It seems to me that this
   terminology risks creating confusion, since a "characterization" in
   mathematics to me suggests a necessary *and* sufficient condition, whereas
   the paper gives only a sufficient condition. Terms such as "sufficient
   condition" or "criterion" might be appropriate, to avoid giving the false
   impression that a necessary condition has been obtained.

2. Line 045R: "a powerful tool for analyzing the structure of the loss
   landscape for various tasks including but not limited to $n$-grams." It's
   not clear to me to what this refers. Is it just non-contiguous
   sub-$n$-grams? Or is something more general meant?

**$k$-gram simple transformer construction.**

I have not fully verified that the construction implements the $k$-gram
estimators, but it appears to be sound based on what I have checked. However, I
think the clarity of the construction requires attention.

3. I attempted to follow the description construction in the main text, and
   uncovered numerous technical errors. Each seems relatively minor and I
   believe a correct construction could be recovered. I list specific issues I
   noticed here.

   1. Equation (7) defining
        $A_1^{(h)} = c \sum_{l=h}^T e_{l-h} e_l^\top$:
      This definition yields an upper-triangular matrix, but I think you want a
      lower-triangular matrix. Also, $A_1^{(h)} \in \mathbb{R}^{T \times T}$,
      suggesting we want $e_i \in \mathbb{R}^{T}$, in which case $e_T$ is not
      defined (since we start with $e_0$.

      Best I can tell, you probably meant
        $A_1^{(h)} = c \sum_{l=h}^{T-1} e_{l} e_{l-h}^\top$.

      This same issue appears to arise in other constructions too.

   2. Equation (7) defining
        $V_1^{(h)} = \sqrt{c} \sum_{j=0}^{\mathcal{S}-1} s_j s_j^\top$:
      Since $s_j$ was defined for
        $j \in [\mathcal{S}] = \{1,\ldots,\mathcal{S}\}$,
      I think this is meant to be a sum from $j = 1$ to $\mathcal{S}$.

      This same issue appears to arise in almost all other sums over $j$,
      except in equation (15), where it's $\sum_{j=1}^{\mathcal{S}-1}$.

   3. Equation (9) neglects the role of the softmax. Perhaps some discussion is
      needed to explain that this equation is meant to hold in the hardmax
      limit. Though, see the next two issues.

   4. Equation (9) appears to be out by a factor of $\sqrt{c}$ (the coefficient
      of the value matrix).

   5. Equation (9) involves terms that are undefined, namely
        $s^\top_{x_{t-n+1}}$ for $t < n$,
      or
        $s^\top_{x_{t-1}}$ for $t=1$.
      In place of these terms, depending on the corrected definition of
      $A_1^{(h)}$, there should be something like a uniform mixture of the
      initial few tokens, or simply $s^\top_1$.

4. In my opinion, the construction should be defined in full, correct detail in
   the main text, and its correctness should be stated formally as a numbered
   theorem/proposition. This would support easier verification of the
   construction's correctness.

Appendix B.1 presents a more detailed description of the construction, with a
partial formalisation of the correctness claim (lemmas B.1 and B.2). This
version seems to be free from some of the technical gaps in the main text
version, but not all of them. I haven't verified the correctness in full detail
(but, once again, it appears to be sound).


**$k$-gram simple transformer satisfies sufficient conditions.**

I have not attempted to verify the soundness of theorem 4.1. I didn't notice
any obvious problems in the proof sketch in the main text. I haven't read
appendices B.2, D, or E containing the full proof.

In my opinion, the implications of this theorem are not clearly articulated in
the paper. In my summary, I distinguish between two claims made by the paper:

* **Limit claim:** Infinite (size and norm) parameters are first-order
  stationary points of the population cross entropy.

* **Finite claim:** In the finite regime, the sufficient condition is
  approximately satisfied, such that the constructed parameters are
  'near-stationary' (gradient close to zero).

This distinction is essential to understanding the contribution of this paper.
Theorem 4.1, as stated, establishes the limit claim, but the titular
contribution of the paper is the finite claim. However, the distinction is not
made clear enough in the paper in my judgement.

5. The finite claim is only ever made explicit informally (in the paragraph
   below theorem 4.1---line 301R---"The result holds asymptotically as
   $T,c\to\infty$," where I take 'asymptotically' here to mean that the
   gradient of constructed parameters approaches zero as the parameters
   approach the infinite norm and infinite length limit). As I understand, the
   proof is implicit in the proof of theorem 4.1. I believe this additional
   conclusion and its proof should be made explicit in formal detail.

6. The limit claim and the finite claim are conflated in the rest of the paper.
   The passage quoted in the previous point continues, "leading us to term the
   $k$-gram estimator as *near*-stationary." However, up to and after this
   point, the paper normally just describes the constructed parameters as
   "stationary" in a technical context without appropriate qualification that
   they are exactly stationary only in the limit:

   1. Section 4 opening (line 181R): "... showing that the sub-$n$-grams are
      stationary." Probably this should say they are "near-stationary."

   2. Section 4.3 opening (line 277R): "We show that the $k$-grams constructed
      ... are first-order stationary points ..." It should probably say that
      are "near-stationary points."

   3. Theorem 4.1 statement (line 289R): "i.e., it is solely a function of the
      $k$-history and context as $T\to\infty$." That should be "almost solely",
      if I understand correctly.

   4. Line 365L: "the derivative is supported only on tokens in $M_t$." If I
      understand correctly, it would be more correct to say the derivative is
      "approximately" only supported on such tokens, or "most of its support"
      is on these tokens, etc.

   5. Line 362R: "As it is a stationary point," should probably read
      "near-stationary."

   6. Conclusion (line 415R): "... the stationary points encountered along the
      training trajectory." It should probably read "near-stationary."

   There are possibly more instances in the main text or appendices.

[CONTINUED IN NEXT SECTION "METHODS AND EVALUATION CRITERIA"]

**Essential References Not Discussed:**

N/A

**Experimental Designs Or Analyses:**

See above.

**Methods And Evaluation Criteria:**

[CONTINUED FROM PREVIOUS SECTION "CLAIMS AND EVIDENCE"]

7. Generally when the role of limits are discussed early in the paper, the
   discussion could be more precise. A non-exhaustive list of specific
   examples:

   1. The claim in the paper's title, "Sub-$n$-grams Are Near Stationary
      Points," is not an accurate description of what is proven. Due to the infinite weight norm required, the
      constructed transformers are actually infinitely far from the identifies
      stationary points (even in the infinite length limit). "Near-Stationary"
      or "Nearly Stationary" would be more appropriate than "Near Stationary".

   2. There is no mention of the infinite norm limit in the abstract (arguably
      acceptable if the discussion were more precise elsewhere).

   3. Contribution list item 2 (line 050R): "solutions representing
      sub-$n$-gram estimators are stationary points," needs qualification.

   4. Later that same paragraph, "... satisfy the conditions sufficient for the
      gradient of the population loss to converge to zero presented in section
      3." In my understanding, the condition in section 3 does not discuss
      convergence to zero gradient, but characterises the gradient and shows
      conditions under which it equals zero.

   In my opinion, it would be better to spell out the limit claim and the
   finite claim on page 1 (perhaps this would be a good opportunity to make it
   clear what "near-stationary" means).

Note that my concern about this contribution is not that the finite parameters
are not exact stationary points per se. Approximate stationarity is sufficient
for the discussion concerning plateaus. It's just important that the paper
avoids giving the false impression that the finite parameters are exactly
stationary, for example.

**Numerical experiments.**

8. The experiments clearly show that the loss of the plateaus roughly lines up
   with the loss of idealised $k$-gram estimators for increasing $k$, and
   during the plateaus the first-layer attention patterns as predicted in the
   construction for these values of $k$. I think this is certainly suggestive
   that the transformer's training trajectory proceeds through these
   'near-stationary' points.

9. I also followed the claims and figures in appendix I and found them
   reasonable.

---

I also discuss some claims that were of some interest to me but were mainly
discussed in the appendices, as follows.

**Discussion of general transformers.**
In general, as architectures change (e.g. weights are unfrozen) the gradient
may cease to vanish because of the new dimensions. Section 4.5 and appendix F.1
pertain to how the construction and proof of near-stationarity can be extended
to a more realistic transformer architecture.

10. The paper claims the construction itself can be easily extended. This claim
    seems unproblematic to me.

11. It is unclear to me whether the paper claims that theorem 4.1 would
    continue to hold in the new construction, without diving into the technical
    details. Appendix F.1 says "we discuss how the results can be extended,"
    but there is no precise statement of the extension that has been achieved
    (i.e., numbered theorem that makes the claim explicit, with proof that
    could be verified).


**Discussion of general non-contiguous sub-histories.**

12. If I understand correctly, the appendices claim that the sufficient
    condition generalised to non-contiguous sub-histories (remark C.2, though
    there is no precise statement); that the construction can be generalised
    (appendix F.2, again no precise statement); and that some elements of the
    proof of theorem 4.1 can be ported to the new construction, but a full
    generalisation to this context is left as an open question. I haven't
    verified the mathematical details, but this seems plausible to me.

**Other Comments Or Suggestions:**

I encountered several minor grammar issues and punctuation/notational/technical
inconsistencies while reading the main text. I think the manuscript would
benefit from careful proof reading to suit a more polished conference paper. A
list of the issues I noticed is as follows.

1. Line 089R: I suppose $R^k$ should be $\mathbb{R}^k$.

2. Throughout there are several terms involving hyphens (such as "$n$-gram,"
   "$(-l)$-history," "$(-l)$-token") for which the hyphens seem to be, in some
   cases, formatted as en-dashes or minus signs. The authors can of course
   choose their preferred punctuation but I invite them to choose one
   preference and use it consistently.

   Relatedly, in rare cases, "$n$-grams" etc. are improperly formatted as text.

3. Line 110R: I suppose "the $x_T^{th}$ token" should be either "$x_T$" or "the
   $T^{\text{th}}$ token."

4. Line 110R: I suppose "$n$-gram estimator" should be "The $n$-gram
   estimator."

5. Line 145R: $Q$ and $K$ here are transposed in comparison to definition 2.2.
   The same is true for $Q_2$ and $K_2$ defined implicitly through equations
   (5).

6. Proposition 3.2 statement, part (a): In what sense is $p_\tau$ a "context"?
   I wasn't able to make sense of this term in this setting.

7. Proposition 3.2 statement, part (b): Since $p_\tau$ is a random variable, I
   believe the condition should specify that
     $p_{\theta_\ast}(x^T) = p_\tau(\cdot | x_t^T)$
   holds almost surely for $p_\tau \sim \mathcal{P}$.

8. Line 364R: Missing period.

9. Remark C.2, line 1018: "conditioning on the any subsequences" typo.

I also take minor issue with the phrasing at several points in the paper's
description of the motivation and results, and I invite the authors to
reconsider these parts or add further justification.

10. Line 013R: the motivating question is "why must training go through plateaus
    to develop such abilities?" It seems to me that this question does not have
    that much relation to the contributions of the paper. The paper establishes
    that approximate stationary points exist, but their existence does not imply
    that training "must" spend any time near such stationary points. A more
    pertinent question, in my opinion, would be "why might training linger at
    plateaus on the way to its eventual solution" (answer: there are partial
    solutions that are stationary points); or perhaps "what parameters comprise
    these plateaus?" (answer: this paper suggests they are sub-$n$-gram
    estimators).

    Similarly, line 026R: "a theoretical foundation for why plateaus
    emerge"---is this really what the paper contributes? It seems to be that the
    existence of plateaus and the fact that, once in a plateau, training
    stagnates, is separate from why training ends up attracted to these
    solutions on the way to the improved solution in the first place.

11. Lines 010R and 081L: References to "circuits, such as induction heads." In
    my understanding, the induction head is only one part of the induction
    circuit studied by Olsson et al. (2022). The "circuit" also involves
    previous-token heads and their composition with induction heads, and is not
    synonymous with the induction heads themselves.

12. The abstract refers to these numerical experiments as "comprehensive" which
    appears to oversell the magnitude of the experimental sections.

Finally I collect some miscellaneous comments.

13. I found the comment at the end of section 4.3 regarding the average loss
    over sequences of varying length interesting. However, I think it
    highlights how results in the infinite $T$ limit are limited in their
    ability to offer insight into training in this setting. For any sequence,
    early in training, any $n$-gram or sub-$n$-gram estimator won't have
    converged to the stationary distribution, and so, predictably, the
    conditional distribution won't align with the estimator's predictions and
    the sufficient condition for zero gradient will not be satisfied.

14. I found the comments in section I.3 interesting. Do you have any
    theoretical insight into why this might be the case? If so, it could be
    valuable to add to the appendix.

15. I invite you to reconsider the inclusion of definition 2.2 in the main
    text. Since the focus is on the simplified architecture, it might make more
    sense to describe the simplified architecture in more formal detail in
    place of this.

16. For what it's worth, I don't feel like there is adequate simplicity payoff
    in leaving some details to the appendix such as positional encoding,
    summation vs. concatenation, and value matrices. The construction and proof
    in the main text is only marginally simpler than the full construction.

    If you want to prioritise simplicity, it might make sense to devise an even
    simpler architecture with infinite sequence length and hard max attention
    built into the architecture, and present the construction and proof in full
    detail in the main text for this architecture, then state the construction
    and the asymptotic version of theorem 4.1 for the full architecture with
    softmax, value matrix, MLP, etc. in the main text and provide the detailed
    proof in the appendix.

Of course, all of these suggestions are offered to the authors based on my
impressions, could be based on misunderstandings, and are only meant for their
consideration.

**Other Strengths And Weaknesses:**

**Strengths.**

As I said above, I think this paper has the potential to be quite a valuable
contribution to the field of "science of deep learning."

**Weaknesses.**

As I said, the paper's contributions are somewhat marred by technical issues
and, in places, a lack of precision and clarity regarding the exact
contributions.

**Questions For Authors:**

As I said, I think this paper makes a valuable contribution, yet I have
recommended weak reject. The reason is because I think there are a number of
issues that get in the way of the paper being accessible. The most serious ones
are the apparent technical errors in presentation of the construction in the
main text (claims, section, item number 3 above) and the distinction between
the "limit case" and the "finite case" (claims section, items number 5 to 7
above). These issues are at the crux of my recommendation. I invite the authors
to focus their response on these issues. If they can address my concerns, one
way or another, I would be pleased to increase my rating.

**Relation To Broader Scientific Literature:**

A current priority for this field of "science of deep learning" is to
understand the dynamics of emergent structure in transformers. Before we can
make progress understanding large-scale models with hundreds of billions of
parameters, we can develop hypotheses in smaller-scale systems. This paper
outlines a pretty neat setting for principled empirical and theoretical work in
this direction.

1. It helps a lot to isolate clean, small-scale examples of phenomena of
   interest, such as stage-wise learning. This paper shows a particularly crisp
   example of stage-wise learning in higher-order Markovian sequence
   prediction. Such was already established by Edelman et al. (2024), however
   this work further simplifies the setting through their experiments with the
   disentangled transformer architecture, which is even simpler than the
   transformers studied by Edelman et al. (2024), but still exhibits stage-wise
   learning.

2. It also helps to have a detailed, mechanistic understanding of how
   particular parameters implement particular in-context learning algorithms.
   The paper contributes a pretty simple construction for a range of
   interesting solutions (sub-$n$-grams). If I understand correctly, this
   construction is original, though I am not familiar with the prior work on
   the disentangled transformer.

3. The main result itself, that these sub-$n$-gram transformers have
   approximately zero gradient, is a pretty interesting observation that fills
   an important piece of the puzzle of the emergence of structure in this
   setting, especially if it does turn out to be true for more general
   transformers.

In summary, I think this paper's contributions are valuable and of interest to
the science of deep learning literature, and could be a step towards a deeper
understanding of the phenomenon of emergent structure in transformers.

**Theoretical Claims:**

See above.

---

> ### Author Rebuttal · Authors · 2025-04-01
>
> Thank you for your detailed and careful review of our paper. We sincerely appreciate the time and effort you have put into reviewing our work and providing constructive feedback. Your comments have helped us identify areas where we can improve the clarity and precision of our presentation. Below, we address the main concerns you have raised and outline the changes we have made in response.
>
> **Distinction between the Finite and Limit Claim**  We give an updated version of Theorem 4.1 that captures the non-asympotatics and make the theorem inline with the title and referenced contributions of the paper.
>
> **Theorem**: For the distangled transformer $p_{\theta}$, the gradients at $\theta^k_*$ are given by
>
> $$ \partial\_{\theta = \theta_*^k}  L(\theta) = \sqrt{c}   {  \mathbb{E}\_{p\_{\tau} \sim \mathcal{P}}   \mathbb{E}\_{x^T \sim p\_{\tau}}  \color{green}{   \| \hat{p}\_k - p\_{\tau}(. \big| x^{T}\_{T-k+2})  \|^2} } + { \color{blue} O( \sqrt{c} T e^{-c} )  }.$$
>
> This error contribution for finite $c,T$ is given by the term above. The blue term represents the error due to finite $c$, while the green term accounts for the finite length and non-stationarity of the chain and vanishes as $e^{-\Theta(T)}$ ( Penev 1991). The limit claim now holds when $c = \Theta(T) \to \infty$.  We hope this clarifies the distinction between the finite and limit cases.
>
> We apologize for being lenient with the distinction between near-stationary and stationary in the text. We agree with the reviewer's concern and will revise the paper to clarify this nuance, explicitly indicating wherever referenced that **stationarity holds only in the limit**. Similarly, we make sure to refer to the points as "near-stationary" instead of "near stationary" throughout the paper.
>
> **k-gram construction.** Please see the response to reviewer 6GFt for the formal construction.
>
> **A tool to analyse landscape.**  Analyzing the stationary points of cross-entropy loss is inherently challenging, particularly compared to regression settings, where the loss landscape is often analytically tractable. Prior work has largely focused on regression due to the tractability, leaving a gap in understanding for sequence-based tasks that rely on cross-entropy loss.
>
> In this proposition, we leverage the factorization of sequence probabilities (e.g., via chain rule or autoregressive decomposition) to derive stationary points. Specifically, we demonstrate how models can converge to stationary distributions by estimating only one factor of the joint probability at a time. This provides a tractable pathway to analyze optimization dynamics in settings where closed-form solutions are otherwise infeasible, see examples given in "practicality of proposition 3.2" section for Reviewer wvkz.
>
> **Other concerns.**
> 10-11.  The construction for general attention-only transformer is not given as it mostly follows on the similar lines as the simplified transformer, hence the neccesary changes and proof sketches were informally given in the Appendix F. We will give a precise formal statement for this case with a proof in the appendix.
>
> **Other Comments or Suggestions.**
>
> 1-5,8-9: We have corrected the typos you identified and carefully proofread to make sure we fixed the inconsistencies in the paper.
>
> 6: We used the term context with a slight abuse of terminology because ${p_{\tau}}$ is the underlying generative process governing the task instance, and determines the distribution of the context $x_1^T$. For clarity we refer to it now as context/task distribution and clarify it in the previous section.
>
> 7: We have fixed the condition to almost sure equality.
>
> 10: We have changed the main question to "Why does training linger at plateaus before developing such abilities?" and editted the introduction in alignment with this point.
>
> 14: Recall that the underlying mechanism behind the sub-$n$-gram estimators is checking if the histroy of the token $x_{T+1}$ and $x_j$ matches and adding $x_j$ if they do. Our conjecture is that, since the token $x_T$ is always provided throught the skip connection, it is easier to learn to match $x_T$ with $x_{j-1}$ than for example $x_{T-1}$ and $x_{j-2}$.
>
> We believe that the revised version effectively addresses your concerns and presents our contributions more clearly. Given these improvements, we hope that you will consider raising your evaluation of our work.

---

> > ### Comment · Reviewer_xuc4 · 2025-04-05
> >
> > Thank you for your detailed rebuttal. I am satisfied that the proposed revisions should address my main concerns, especially the quantification of the gradient bound in the finite case and the commitment to more carefully distinguish these cases through terminology.
> >
> > I have followed the discussion with other reviewers. I appreciate the addition of tracking the gradient norm in experiments, though it would be helpful to have some baseline to compare the value of the gradient norm. I am sympathetic to reviewer wvkz's remaining concerns. Assuming I understood the concerns correctly, I have the following comments.
> >
> > 1. Regarding the limitation of studying n-grams, I'm not so concerned about this, because (1) the n-grams setting like a useful case study, it allows us to study at least one instance of stagewise progression of structure in a model, and (2) while reading the paper I felt encouraged to speculate that future work could relax this assumption to more general sequence structures, e.g. from Markov models to HMMs or DFAs or CFGs or etc., with the idea that generalised sub-structures will also turn out to be near-stationary points. I'm not sure how likely this direction is be to pan out theoretically, and I am sure it would raise its own challenges, but its a thought-provoking direction.
> >
> > 2. Regarding the effectiveness of the bounds in the low-T, low-c regime. I agree this is a concern. On the other hand, we see clear plateaus in the experiments, suggesting there is some meaningful slowdown. I guess the question is whether or not the slowdown is attributable to these solutions. The present evidence based on comparing loss and attention patterns is suggestive. I wonder if there is some way to more effectively establish that the bounds are meaningfully related to the plateau dynamic. Is it possible to compare the observed gradient norms to the predicted gradient bounds? If not, what are the expected sources of deviation?
> >
> > I am currently raising my score from 2 to 3. I look forward to following further discussion.

---

> > > ### Author Response · Authors · 2025-04-09
> > >
> > > We thank the reviewer for their thoughtful follow-up and taking the time to engage deeply with our work. We are very happy that the proposed revisions and clarifications successfully addressed their concerns.
> > >
> > > **Comments on n-grams and future work**.  We appreciate the reviewer's encouraging remarks regarding the research direction of our paper. We found the suggestions for future work to be both insightful and well-aligned with our ongoing efforts. In particular, we have been exploring extensions to regular languages beyond Markov sequences, such as those discussed in [1,2]. We believe that Proposition 3.2 can be a useful tool for showing the near-stationarity of the sub-structures.
> > >
> > >
> > > **Baselines, discrepany and validity of the bounds.**
> > > To interpret the gradient norm experiments meaningfully, we identify two baselines:
> > > - High-gradient baseline: The gradient norm at random initialization provides a reasonable baseline for a relatively high value of the gradient.
> > > - Low-gradient baseline:  The gradient norm at the end of training can be considered a good baseline for a low value of the gradient as the training is expected to converge to a minima.
> > >
> > >
> > > In order to reduce discrepancies and improve the reliability of gradient norm measurements—especially those arising from stochastic noise due to mini-batch sampling—we made the following adjustments to the experimental setup:
> > > - **Larger test batch:** We increased the number of test samples used to compute the gradient norm from $10^3$ to $10^4$, allowing for a better approximation of the population loss gradient.
> > > - **Smoothing the trajectory:** Stochastic noise during training can perturb the trajectory and deviate it from the stationary points, which in turn affects gradient measurements. To mitigate this, we report the gradient norm using an exponential moving average (EMA) of the training trajectory. The variance reduction provided by EMA produces a trajectory that more closely reflects training under the population loss.
> > >
> > >
> > > Figure 1 (available at https://anonymous.4open.science/r/icml_rebuttal_2025-C11E/rebuttal.pdf) presents the updated gradient norm measurements. With these changes, the baselines are clearer and the peaks are closer to the random initializations and the intermediate stages the norm is closer to the norm at the end. Note that in addition to our metrics of loss comparison and attention patterns, Edelman et. al. 2024 also provides the KL-divergence with sub-n-gram solutions during training. Beyond these strategies, we cannot find a rigorous way to measure the validity of our theoritical bounds, as it is hard to estimate the right constants and the distance between probabilites for $k$-gram over the entire population.
> > >
> > >
> > > [1] Akyürek, Ekin, et al. "In-context language learning: Architectures and algorithms." arXiv preprint arXiv:2401.12973 (2024).
> > >
> > > [2] Cagnetta, Francesco, and Matthieu Wyart. "Towards a theory of how the structure of language is acquired by deep neural networks." arXiv preprint arXiv:2406.00048 (2024).

---

### Decision · Program_Chairs · 2025-05-01

**Decision:**

Accept (poster)

**Comment:**

This paper aims to study the loss landscape of next-token prediction for transformers where cross-entropy defines the underlying loss function. More precisely, assuming that the underlying data follows an n-gram language model, the paper analyzes the loss landscape of a simplified attention-only transformer architecture, namely disentangled transformers. The paper shows that k-gram estimators for k <= n form asymptotic stationary points or non-asymptotic near-stationary points. The paper connects these stationary points (for k < n) to the stage-wise learning phenomenon where the training goes through plateaus. The paper also presents synthetic experiments to corroborate the theoretical findings.

The reviewer raised multiple valid questions/concerns about the initial submission. The reviewers pointed out various issues with the presentation of the results and the claimed extension in the initial submission. The reviewers asked for further empirical results to strengthen the main takeaways of the initial submission. The reviewer also asked for further justification about various assumptions in the paper. The authors were able to adequately address most of these concerns during the author-reviewer discussion phase. At the end of the discussion phase, all the reviewers agreed on the importance of the findings in the paper and recommended acceptance provided the final paper takes all the key points discussed during the discussion phase into account.

The authors are strongly recommended to make all the promised changes. In addition, it would be useful to bring up disentangled transformers much earlier (e.g., in the introduction) as the results in the paper are mostly restricted to those.